# Cell-autonomous Hedgehog signaling controls Th17 polarization and pathogenicity

Joachim Hanna [1], Flavio Beke [1], Louise M. O'Brien [1], Chrysa Kapeni [1], Hung-Chang Chen [1], Valentina Carbonaro[1], Alexander B. Kim [1], Kamal Kishore [1], Timon E. Adolph [2], Mikkel-Ole Skjoedt[3,4], Karsten Skjoedt[5], Marc de la Roche[6] & Maike de la Roche [1✉]

Th17 cells are key drivers of autoimmune disease. However, the signaling pathways regulating Th17 polarization are poorly understood. Hedgehog signaling regulates cell fate decisions during embryogenesis and adult tissue patterning. Here we find that cell-autonomous Hedgehog signaling, independent of exogenous ligands, selectively drives the polarization of Th17 cells but not other T helper cell subsets. We show that endogenous Hedgehog ligand, Ihh, signals to activate both canonical and non-canonical Hedgehog pathways through Gli3 and AMPK. We demonstrate that Hedgehog pathway inhibition with either the clinically-approved small molecule inhibitor vismodegib or genetic ablation of *Ihh* in CD4+ T cells greatly diminishes disease severity in two mouse models of intestinal inflammation. We confirm that Hedgehog pathway expression is upregulated in tissue from human ulcerative colitis patients and correlates with Th17 marker expression. This work implicates Hedgehog signaling in Th17 polarization and intestinal immunopathology and indicates the potential therapeutic use of Hedgehog inhibitors in the treatment of inflammatory bowel disease.

[1] Cancer Research UK Cambridge Institute, University of Cambridge, Robinson Way, Cambridge CB2 0RE, UK. [2] Department of Internal Medicine I, Gastroenterology, Hepatology & Endocrinology, Medical University Innsbruck, Innsbruck, Austria. [3] Rigshospitalet - University Hospital Copenhagen, Blegdamsvej 9, 2100 Copenhagen, Denmark. [4] Institute of Immunology and Microbiology, University of Copenhagen, Blegdamsvej 3B, 2200 Copenhagen, Denmark. [5] University of Southern Denmark, J.B.Winslows Vej, 5000 Odense C, Denmark. [6] Department of Biochemistry, University of Cambridge, 80 Tennis Court Road, Cambridge CB2 1GA, UK. ✉email: maike.delaroche@cruk.cam.ac.uk

The adaptive immune system is able to initiate highly tailored immune responses. After antigenic stimulation, naïve CD4[+] T cells can polarize into specialized T helper (Th) cell lineages (Th1, Th2, Th17, and iTregs), characterized by the production of key effector cytokines that dominate subsequent immune responses or into T follicular helper cells that provide B cell help and are essential for germinal center formation. This process of Th lineage polarization is critical for the quiescent naïve CD4[+] T cell to acquire the functional ability to secrete Th effector cytokines and by corollary to exert Th effector functions. Distinct Th lineages are each able to respond to different classes of immune challenges. Th1 cells secrete IFNγ to promote immunity against intracellular pathogens, while Th2 cells secrete IL-4, IL-5, and IL-13 to respond to helminth infections. Th17 cells produce IL-17 and IL-22 required for clearance of extracellular pathogens, while Tregs suppress immune responses (reviewed in ref. [1]).

This lineage-specific fate decision is hence of great importance and is governed by environmental signals received by the CD4[+] T cell. These include the mechanism through which the T cell is activated by antigen presenting cells (APCs) and the strength of T cell receptor (TCR) signaling. Perhaps most critical are signals from the extracellular cytokine milieu which initiate intracellular STAT and SMAD signaling. These factors drive the induction of lineage-specific and lineage-associated transcription factors to promote and reinforce Th lineage fate choice[2].

Th17 polarization is initiated by TGFβ and inflammatory cytokines, which signal through SMAD2 and STAT3, respectively, to induce the Th17 polarization program; this is guided by a number of transcription factors including the 'master' transcription factor RORγt[3]. However, although the transcription factor network governing Th17 identity has been well described[4], the intracellular signaling pathways regulating this complex polarization program are less clear.

Th17 cells are critical for maintaining the integrity of intestinal barrier surfaces and coordinating the immune response against pathogenic extracellular bacteria and fungi. However, Th17 cells are also key drivers of autoimmune diseases including inflammatory bowel disease (IBD), rheumatoid arthritis, and multiple sclerosis[5]. There is therefore strong clinical interest to specifically inhibit polarization and pathogenicity of these cells.

The Hedgehog (Hh) pathway has key functions in regulating cell fate choices throughout metazoan development and adult tissue homeostasis by regulating tissue patterning, cell proliferation, and differentiation[6]. Vertebrate Hh signaling critically depends on the primary cilium, where the pathway is initiated upon binding of extracellular Sonic, Desert or Indian Hh ligands (Shh, Dhh, Ihh) to the transmembrane receptor Patched (Ptch) on a Hh-responsive cell. Ligand binding to Ptch allows the transmembrane protein Smoothened (Smo) to translocate into the primary cilium, where it activates the glioma-associated oncogene (Gli) transcription factors, Gli2 and Gli3. These translocate to the nucleus and initiate the transcription of Hh target genes including Gli1[6,7].

In the immune system, the Hh signaling pathway has been implicated in T cell development in the thymus[8]. Given the structural and morphological similarities between the primary cilium and immune synapses[9,10], we were prompted to study Hh signaling in mature T cells[11]. TCR signaling at the immune synapse upregulates Hh components in CD8[+] T cells independently of exogenous Hh ligands, which we have shown to be important for CD8[+] T cell killing. Our data raised the possibility that Hh signaling in CD8[+] T cells may occur intracellularly[11]. By contrast, less is known about Hh signaling in mature CD4[+] T cells. One study has shown that transgenic expression of Gli2 activator and repressor forms in CD4[+] T cells can influence Th2

polarization[12], but the role of endogenous Hh signaling in Th polarization is not known.

Here, we find that Hh signaling is functionally important for Th17 but not Th1, Th2, and iTreg CD4[+] T cell polarization. While conventional Hh signaling involves paracrine signaling—in which Hh ligands are secreted from one cell and signal to another cell that expresses the Hh receptor—we show that in CD4[+] T cells Hh signaling is cell-autonomous/autocrine and is mediated by Ihh, Smo, and Gli3. Mechanistically, Smo activates Gli3 which is needed for Th17 polarization and effector function, and also regulates metabolic fitness in Th17 cells in a non-canonical fashion through AMPK phosphorylation. Functionally, we show that blocking Hh signaling in vivo genetically or with small molecule Hh inhibitors ameliorates disease in two models of Th17-driven IBD.

## Results

**Key Hh signaling components are selectively upregulated in Th17 cells.** Previous work suggested that Hh components Smo and Ptch are expressed in CD4[+] T cells[13]. We extended these analyses by comprehensively profiling Hh ligands, receptors, signal transducer, and transcription factors (Fig. 1a) throughout CD4[+] Th polarization in an established in vitro polarization protocol (Supplementary Fig. 1) by western blot, immunofluorescence, and quantitative RT-PCR.

The Hh ligand Dhh was not detected in any of the Th lineages and only the Th2 subset expressed and upregulated Shh mRNA post day 3 (Supplementary Fig. 2a, b). By contrast, Ihh was the only Hh ligand already present in naïve CD4[+] T cells and was expressed throughout Th17 polarization (Fig. 1b and Supplementary Fig. 2a–c). Hh receptors Ptch1 and Ptch2 were expressed in all lineages with Ptch1 being predominantly expressed (Supplementary Fig. 2a and Fig. 1c). Smo mRNA was also detected in all Th lineages but showed most pronounced upregulation in Th17 cells on day 3 of culture (Supplementary Fig. 2a). Since Smo is the key signal transducer of the Hh pathway, we wanted to follow up on this observation with the analysis of Smo protein. As no functional anti-Smo antibodies were commercially available, we generated monoclonal antibodies directed against the C-terminus of Smo (Supplementary Fig. 3) and show that only the Th17 lineage expresses strikingly high levels of Smo protein on day 3 and day 5 (Fig. 1c). Using our monoclonal antibody, we find that Smo resides on the plasma membrane and on intracellular vesicles as has been described for CD8[+] T cells previously[11] (Fig. 1d).

Vertebrate Hh signaling activity can be evaluated by the expression of the downstream Gli transcription factors[14]. All lineages upregulated Gli1 at low levels upon Th polarization. Given its potential to regulate Th2 polarization[12], we were interested in assessing Gli2 expression. However, we were unable to detect Gli2 transcripts in any of the conditions tested (Supplementary Fig. 2a). Strikingly, Gli3 was the only transcription factor showing lineage-specific expression in Th17 and iTreg T cells peaking at day 3 of culture (Fig. 1e). Interestingly, expression levels of Ihh, Ptch1, and Smo mRNA were comparable to levels observed in Mouse Embryonic Fibroblasts (MEFs), bona fide Hh signaling-competent cells (Fig. 1b and Supplementary Fig. 2a, b).

Taken together, while all lineages expressed core Hh components, only Th17 cells express high levels of Smo, as well as Gli3 transcription factors by day 3 of culture when lineage choice is specified. Furthermore, Ihh is the only Hh ligand expressed in Th17 cells.

**Th17-polarizing cytokines induce the expression of key Hh components.** Th17 lineage polarization is initiated when naïve

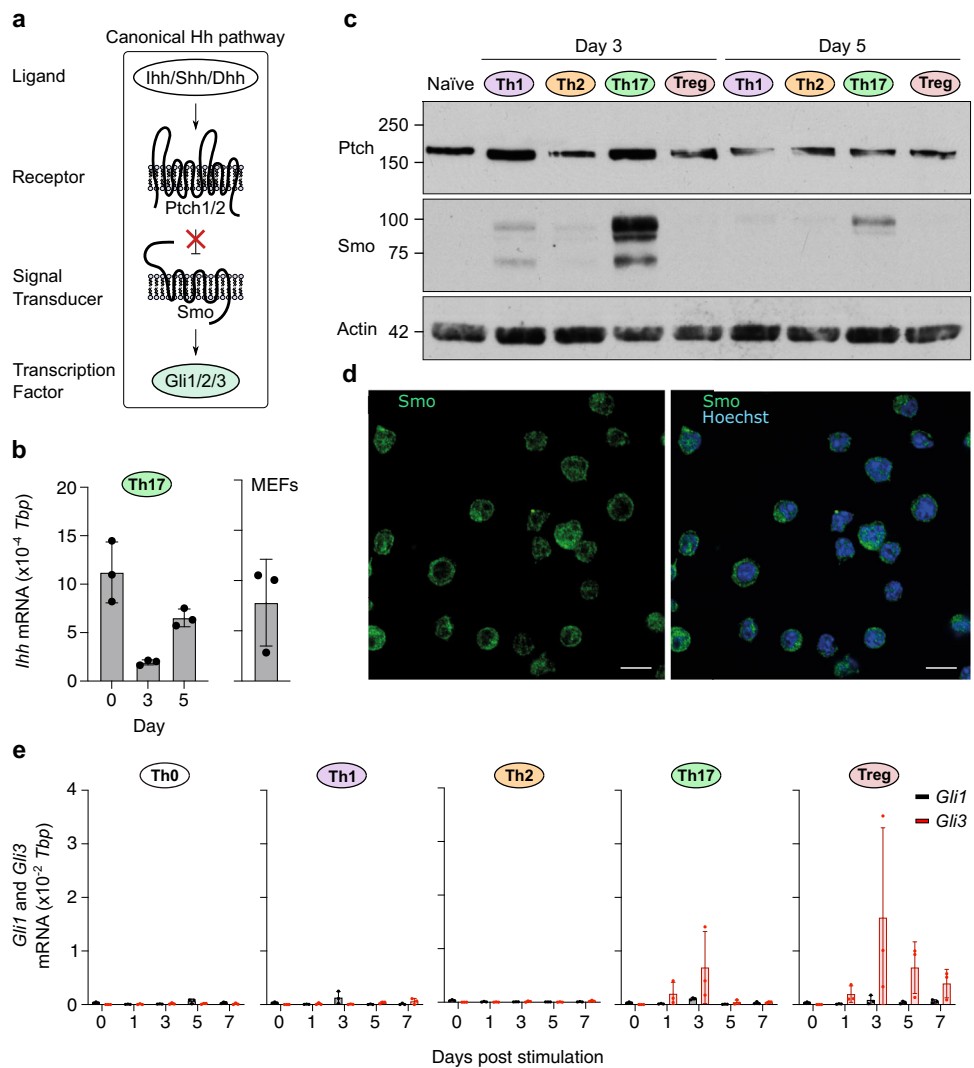

**Fig. 1 Key Hedgehog signaling components are induced in Th17 cells. a** Overview of canonical Hh signaling. **b–e** Naïve CD4+ T cells were purified from spleen and peripheral lymph nodes of C57BL/6 mice and stimulated with plate-bound anti-CD3ε/CD28 antibodies in the presence of polarizing cytokines to generate Th0, Th1, Th2, Th17, and iTreg subsets. **b** Expression of *Ihh* was assessed by qRT-PCR in naïve CD4+ T cells and Th17 cells at the indicated timepoints after TCR stimulation (left) and in mouse embryonic fibroblasts (right). Data is normalized to *Tbp* as a reference gene. n = 3 independent experiments. **c** Immunoblot analysis of Ptch and Smo in naïve CD4+ T cells and T helper (Th) subsets at indicated timepoints post stimulation. n = 2 (naïve) or 3 (Th subsets) independent experiments. **d** Immuno-fluorescence imaging (single *x-y* confocal section) of Th17 cells at day 3 labeled with antibodies against Smo (green). Nuclei were stained with Hoechst (blue). Scale bars: 10 µm. **e** Expression of *Gli1, Gli2,* and *Gli3* were assessed by qRT-PCR in Th subsets at indicated timepoints post stimulation in the presence of polarizing cytokines. Data is normalized to *Tbp* as a reference gene with similar results obtained when using *CD3ε* as a reference gene. n = 3 independent experiments. *Gli2* mRNA was undetectable in all conditions tested (see also Supplementary Fig. 2a, b). Source data are provided in the source data file.

CD4+ T cells recognize cognate antigen on APCs in the presence of a cytokine milieu dominated by IL-6, TGFβ, IL-1β, and IL-23. Given the Th17-specific upregulation of Smo and *Gli3* (Fig.1), we asked whether Th17-inducing cytokines play a role in the induction of these Hh signaling components. To address this question, we analyzed Smo and *Gli3* expression in the presence of different combinations of cytokines involved in Th17 polarization. The ability of the given cytokine cocktails to drive IL-17a production (and hence the strength of Th17 polarization signal) is shown in Supplementary Fig. 4a.

First, we titrated TGFβ in the presence of fixed concentrations of IL-6, IL-23, and IL-1β and assessed the expression of *Gli3* as well as *Gli1* by qRT-PCR and Smo by western blot. Strikingly, *Gli3* but not *Gli1* expression was upregulated by TGFβ in a dose-dependent manner (Fig. 2a, upper panel), which is critically dependent on the presence of IL-6. 1 ng/ml TGFβ led to optimal

IL-17a expression (Supplementary Fig. 4a) and highest Smo levels (Fig. 2a, lower panel, Supplementary Fig. 4b). Interestingly, IL-6 alone was necessary and sufficient to induce Smo protein and stepwise addition of TGFβ, IL-23, and IL-1β increased not only IL-17a production (Supplementary Fig. 4a) but also *Gli3* RNA expression levels (Fig. 2b, upper panel) as well as Smo protein (Fig. 2b, lower panel, Supplementary Fig. 4b). Taken together, this data suggests that while IL-6 is sufficient to induce expression of the signal transducer of the Hh pathway (Smo), TGFβ is also needed to drive expression of the lineage-specific transcription factor (Gli3). Hence the presence of both of these key cytokines is needed for optimal Th17 polarization (Supplementary Fig. 4a) as well as for optimal Hh signaling induction.

In order to validate that TGFβ and IL-6 signaling induce *Gli3* and Smo expression, respectively, we treated Th17 cultures with potent small molecule inhibitors of TGFβ signaling (SB 505124)

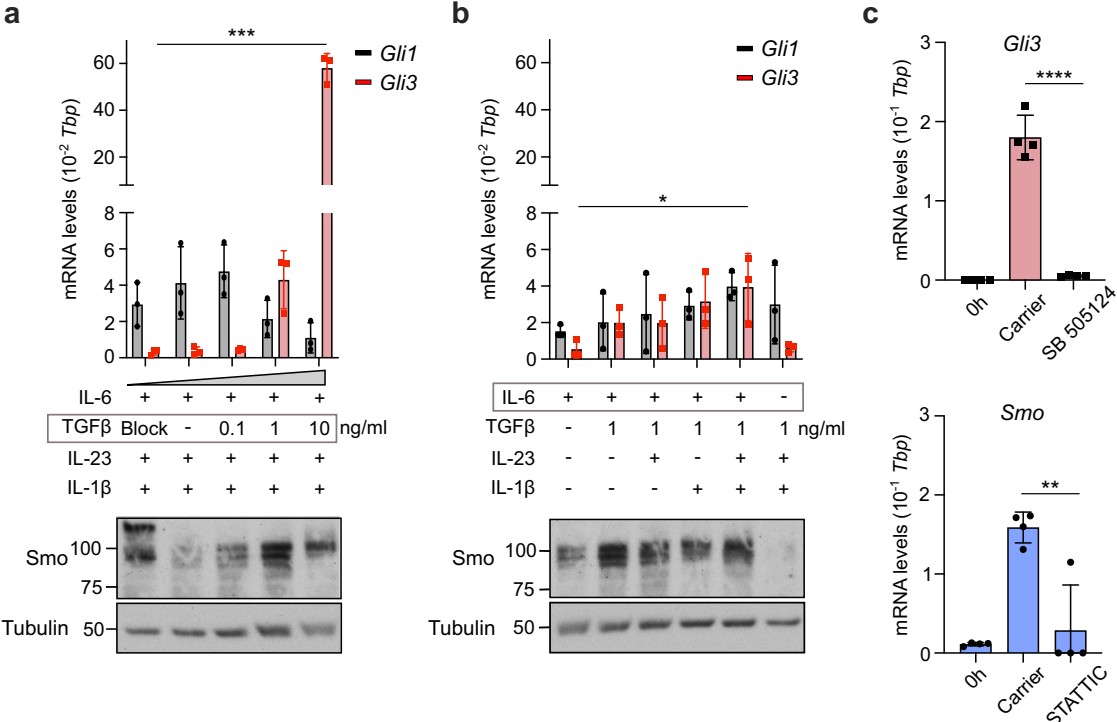

**Fig. 2 IL-6 and TGFβ are the primary inducers of Hh signaling in Th17 cells. a–c** Naïve CD4+ T cells were purified from spleen and peripheral lymph nodes of C57BL/6 mice and polarized with the indicated polarizing cytokines. **a**, **b** Cells were harvested at day 3 post stimulation for immunoblot analysis of Smo and qRT-PCR analysis of *Gli1* and *Gli3*. TGFβ blocking antibody (clone: 1D11) was added in the indicated condition for the duration of polarization at 10 μg/ml. Data is normalized to *Tbp* as a reference gene. Similar results were obtained when *CD3ε* was used as a reference gene. $n = 2$–3 independent experiments. **c** Cells were polarized in vitro under full Th17-polarizing conditions and harvested at day 3 post stimulation for qRT-PCR of *Gli3* and *Smo*. Cells were treated for three days with either 2 μM STATTIC, 1 μM SB 505124, or carrier control; naïve CD4 + T cell RNA levels ("0 h") are shown as controls. Data is normalized to *Tbp* as a reference gene. Similar results were obtained when *CD3ε* was used as a reference gene. $n = 4$. Data are means +/− SD. *p*-values were calculated using a two-way ANOVA (**a**, **b**) or one-way ANOVA (**c**) with Tukey's multiple comparisons test. *$p < 0.05$, **$p < 0.01$, ***$p < 0.001$, ****$p < 0.0001$. Source data are provided in the source data file.

and STAT3, the key transcription factor downstream of IL-6 receptor signaling (STATTIC), respectively. Inhibition of TGFβ signaling led to complete ablation of *Gli3* induction while inhibition of IL-6 signaling via STAT3 abolished the induction of *Smo* (Fig. 2c).

Thus, two distinct Th17 lineage-polarizing cytokines, TGFβ and IL-6, promote the expression of the central Hh components *Gli3* and *Smo*, respectively.

**Exogenous Ihh does not affect Th17 polarization.** In the canonical pathway, Hh signaling is initiated upon binding of the exogenous Hh ligand, generated by Hh-producing cells, to the receptor Ptch on the plasma membrane of Hh-responsive cells[7]. We first asked whether exogenous Hh ligands could promote Th17 polarization either in the absence of any polarizing cytokines (Th0 condition), in the presence of IL-6 alone or in the presence of intermediate (IL-6 and TGFβ) or strong (IL-6, TGFβ, IL-23, and IL-1β) Th17 polarizing conditions (Fig. 3a and Supplementary Fig. 4c). Surprisingly, even high concentrations of active Ihh N-peptide were unable to promote Th17 polarization in any condition tested. To confirm our hypothesis that exogenous Hh ligands do not drive Th17 polarization, we sequestered all extracellular Hedgehog ligands with the 5E1 anti-Hh ligand blocking antibody (Supplementary Fig. 4d) and show that IL-17a production is not affected across any of the "polarization" conditions tested (Fig. 3b). This suggests that exogenous Hh ligands may not be key drivers of Th17 polarization, and that any

role played by Hh signaling in Th17 polarization would be independent of these extracellular ligands.

**Hh inhibitors selectively impair Th17 polarization in vitro.** Hh signaling can be efficiently blocked by small molecule inhibitors of the key signal transducer Smo. To investigate the functional role of Hh signaling in CD4+ Th lineage polarization, we treated T cell cultures with Hh inhibitor cyclopamine for the first 3 days (when lineage identity is determined in vitro) and assessed lineage identity on day 5 (Fig. 4a). Strikingly, only Th17 (IL-17a, top row) but not iTreg (FoxP3, bottom row) polarization was compromised in a dose-dependent manner (Fig. 4b). This suggested that Hh signaling is required at the time of lineage polarization. Viability and cell number of Th17 cells were not significantly affected at the inhibitor concentrations used (Fig. 4b, last two columns). Similar results were obtained with clinically approved Smo inhibitor vismodegib[15] (Supplementary Fig. 5a–c). Cyclopamine treatment also downregulated surface CCR6 expression, a hallmark protein of Th17 cells, in a dose-dependent manner (Fig. 4c) but did not affect Th1 or Th2 polarization (Fig. 4d and Supplementary Fig. 5d). The same dose-dependent downregulation of CCR6 was seen in vismodegib-treated Th17 cells (Suppl. Figure 5c). Inhibition of Smo also led to a dose-dependent downregulation of IL17f, another key Th17 effector cytokine (Supplementary Fig. 5b). Taken together, the data demonstrates that Th17 polarization is greatly diminished upon inhibition of Hh signaling.

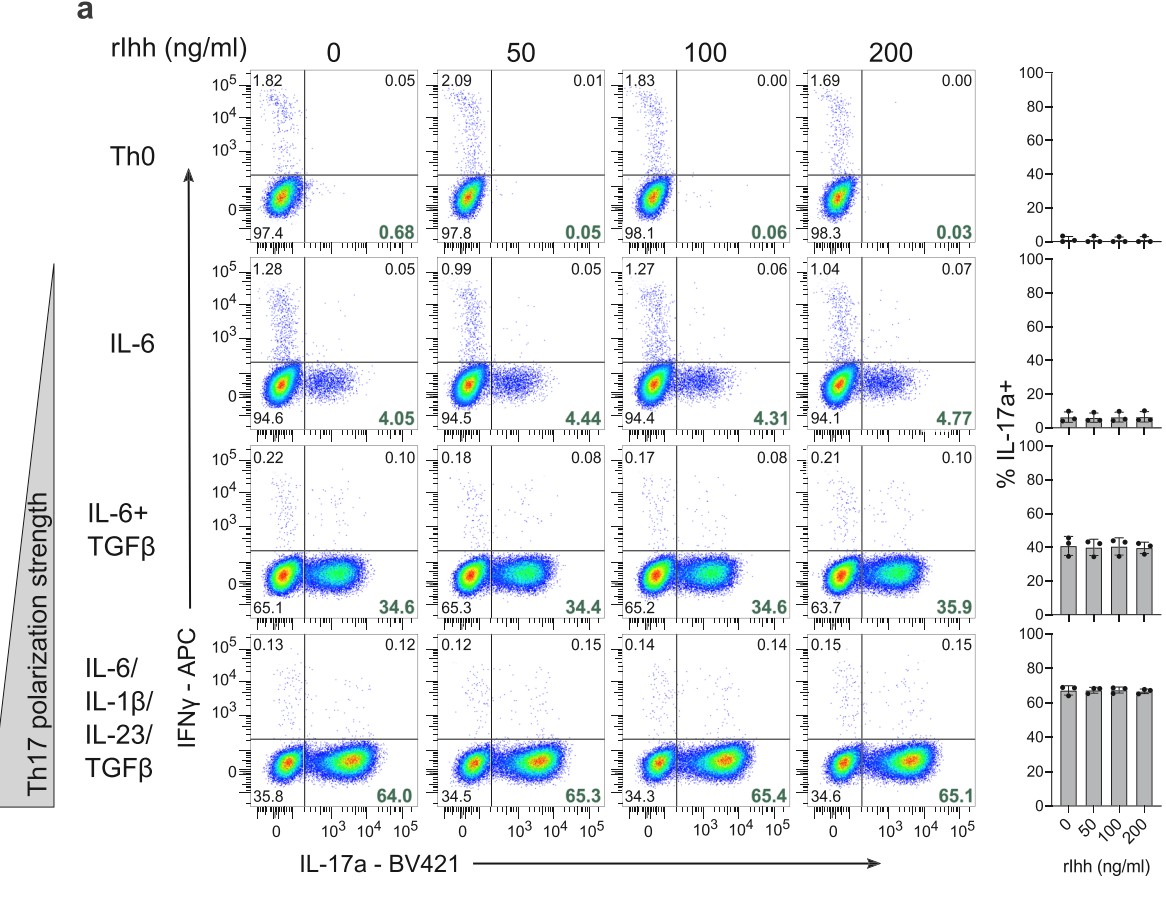

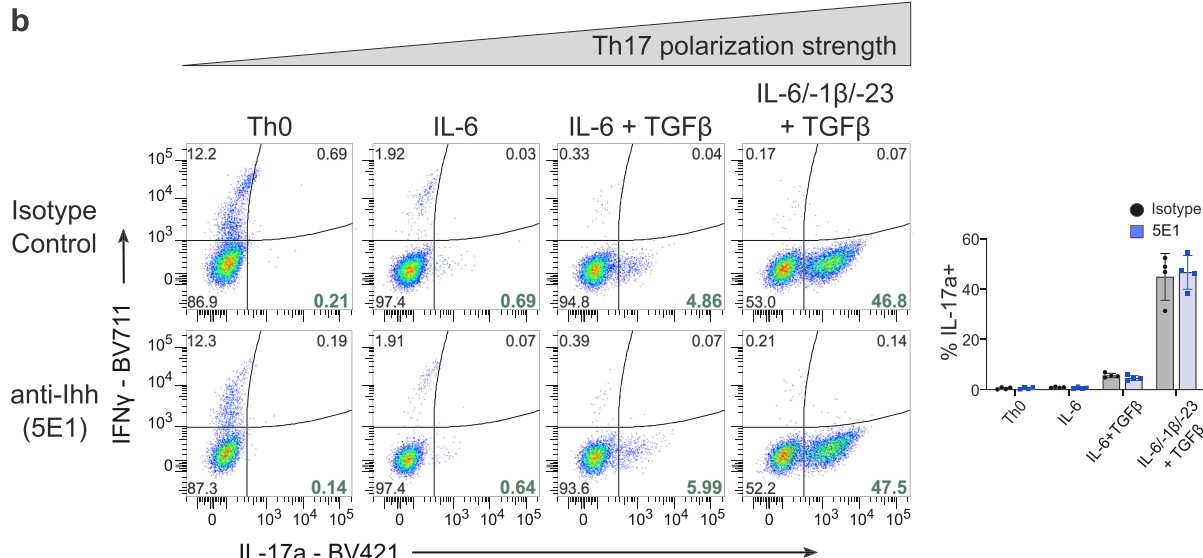

**Fig. 3 Exogenous Hedgehog ligands do not affect Th17 polarization. a** Naïve CD4+ T cells were stimulated with the indicated polarizing cytokines in the presence/absence of recombinant N-terminal murine Ihh fragment at the indicated concentrations. All cells were polarized in the presence of anti-IFNγ and anti-IL4 blocking antibodies and were harvested for analysis by flow cytometry on day 5. Representative flow cytometry plots of $n = 3$ independent experiments are shown with a summary on the right. **b** Naïve CD4+ T cells were stimulated with the indicated polarizing cytokines in the presence or absence of Hh ligand blocking antibody 5E1 or isotype control antibody at 10 µg/ml. $n = 4$. **a**, **b** Data are means $+/-$ SD. p-values were calculated using one-way ANOVA with Tukey's multiple comparisons test (**a**) or two-way ANOVA with Sidak correction (**b**) No statistically significant differences were observed. Source data are provided in the source data file.

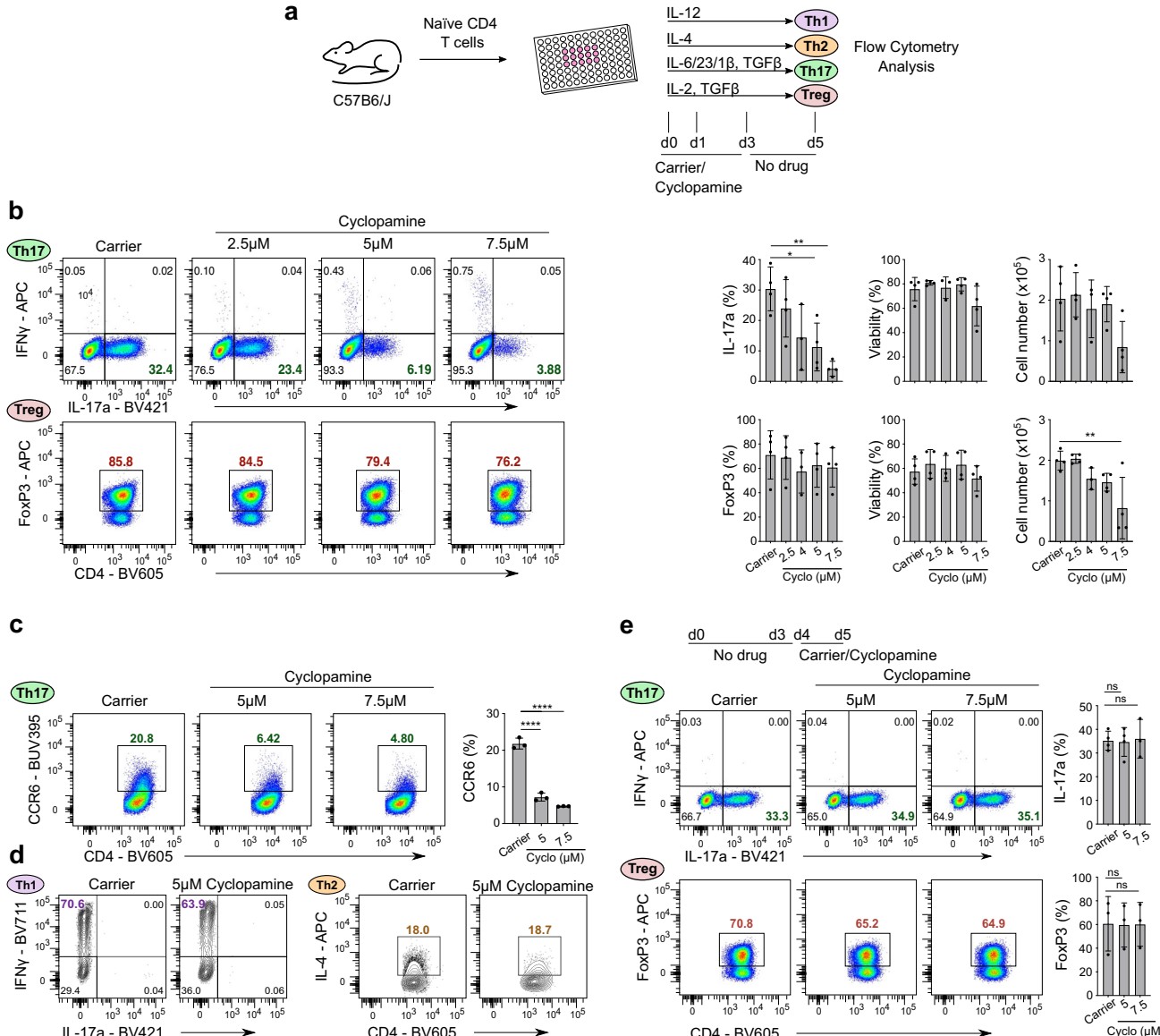

**Fig. 4 Small molecule Hh inhibitors selectively block Th17 polarization in vitro. a** Schematic overview of the Hh inhibitor cyclopamine administration schedule using in panels **b**–**d**. **b** Naïve CD4+ T cells were stimulated under Th17- (top row) or iTreg- (bottom row) polarizing conditions in the presence of the indicated doses of cyclopamine ("cyclo") or carrier control for three days. Cells were harvested for analysis by flow cytometry on day 5. Quantitation of IL-17a and FoxP3 expression for Th17 cells and iTregs respectively, viability measured by the absence of live/dead staining, and cell numbers are shown on the right. Th17 data = top row. iTreg data = bottom row. n = 3–4 independent experiments. **c** Naïve CD4+ T cells were stimulated under Th17 polarizing conditions in the presence of the indicated doses of cyclopamine or carrier control for three days. Cells were harvested for analysis by flow cytometry on day 5. Quantitation of cell-surface CCR6 is shown on the right. n = 3 independent experiments. **d** Naïve CD4+ T cells were stimulated under Th1 or Th2 polarizing conditions in the presence of 5 µM cyclopamine or carrier control for three days. Cells were harvested for analysis by flow cytometry on day 5. n = 3 independent experiments. **e** Naïve CD4+ T cells were stimulated under Th17 or iTreg polarizing conditions in the presence of the indicated dose of cyclopamine or carrier control for the final 24 h of polarization. Cells were harvested for analysis by flow cytometry on day 5. n = 3–4 independent experiments. Data are means +/− SD. p-values were calculated using one-way ANOVA with Tukey's multiple comparisons test. *p < 0.05, **p < 0.01, ***p < 0.001, ****p < 0.0001 .

Next, we investigated the window of active Hh signaling during in vitro CD4+ T cell polarization. For this, we limited the Hh inhibitor treatment to the last 24 h of the 5-day Th17 and iTreg polarization culture before assaying them by flow cytometry on day 5 (Fig. 4e). Interestingly, we found that inhibition of Hh signaling post-day-4 did not affect IL-17a production or FoxP3 levels indicating that Hh signaling is active in the first 3–4 days of culture. These results are in line with the high expression of Hh signaling components during this time. Thus, our data is

consistent with Hh signaling having important roles during Th17 polarization but not Th1, Th2, or iTreg polarization. Furthermore, this data supports the notion that Hh signaling controls Th17 polarization itself (Fig. 4b, e) rather than merely Th17 effector function.

**CD4+ T cell-specific ablation of *Ihh* and *Smo* impairs Th17 polarization.** To confirm our results genetically, we generated

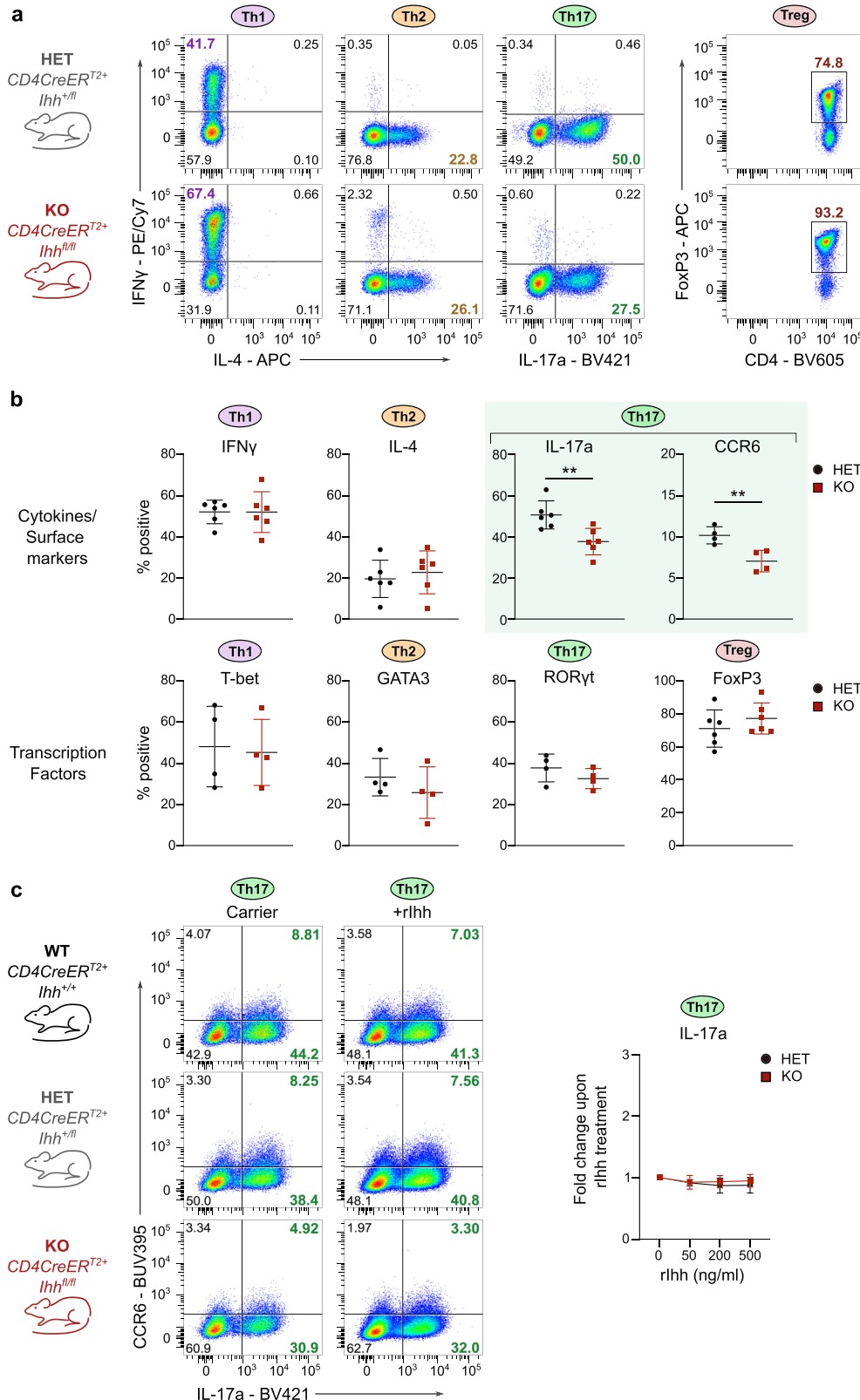

two independent conditional Hh knockout models: one using the *CD4CreER^{T2}* where the Hh ligand Indian Hh (*Ihh*) can be inducibly deleted in naïve CD4+ T cells upon administration of tamoxifen and the other making use of the *dLckCre* which leads to ablation of *Smo* after completion of T cell development in the thymus. The *CD4CreER^{T2} Ihh^{flox}* model is very important to prove that Hh signaling is truly cell-autonomous in CD4+ T cells

and not dependent on exogenous ligands. For all experiments with this mouse model, tamoxifen administration was limited to four days to ensure that only mature naïve CD4+ T cells with *Ihh* knocked out would be present in the periphery at the time of sacrifice[16], which was critical to ensure there were no confounding effects of *Ihh* loss during thymic development for our in vitro (Fig. 5) and in vivo (Fig. 6) studies. A *Rosa26 loxP-STOP-*

**Fig. 5 Conditional knockout of *Ihh* in CD4[+] T cells leads to diminished Th17 polarization but does not affect other Th lineages.** Naïve CD4[+] T cells were purified from spleen and peripheral lymph nodes of either tamoxifen-treated *CD4CreER[T2+] Ihh[+/fl]* (HET) or *Ihh[fl/fl]* (KO) C57BL/6 mice (**a**, **b**) Cells were stimulated under Th1, Th2, Th17 or iTreg polarizing conditions and harvested for analysis by flow cytometry on day 5. n = 4–6 mice per genotype. Flow cytometry plots are shown in **a** and quantification is shown in **b**. Data are means +/− SD. Th17 cell data are highlighted with green shading. **c** Th17 cells were polarized from WT, HET, and KO mice in the presence/absence of 200 ng/ml recombinant N-terminal murine Ihh fragment. Cells were harvested for analysis by flow cytometry on day 5. Representative flow cytometry plots are shown on the left and a summary on the right. n = 3–4 mice per genotype. Data are means +/− SD. **b** p-values were calculated using an unpaired two-tailed Student's *t* test or a two-way ANOVA with Tukey's multiple comparisons test (**c**). *p < 0.05, **p < 0.01, ***p < 0.001. Source data are provided in the source data file.

loxP tdTom cassette allowed for tdTom labelling of cells with active Cre-excision.

Importantly, we were able to conclusively validate the data obtained from the inhibitor studies: loss of *Ihh* (Fig. 5a, b) or *Smo* in CD4[+] T cells (Supplementary Fig. 6) resulted in diminished Th17 lineage polarization compared to control cells, while Th1, Th2, and iTreg polarization remained unaffected. Knockout mice from both models had phenotypically normal peripheral T cell compartments (Supplementary Fig. 7a, b, d).

The data from the *CD4CreER[T2] Ihh[flox]* mice (Fig. 5a, b) demonstrates that CD4[+] T cell-intrinsic Ihh ligand is required for full Th17 polarization. To confirm our hypothesis that this cell-autonomous mode of Hh signaling operates using T cell-intrinsic Hh ligand only, we asked whether the Th17-polarization defect in *Ihh* knockout cells could be rescued by the addition of exogenous Ihh. The addition of recombinant active Ihh did not affect IL-17a production in *Ihh* KO cells (Fig. 5c). Hence exogenous Ihh is not able to rescue the Th17-polarization defect observed in the *Ihh* knockout cells.

Thus, we firmly establish a selective requirement for a cell-autonomous mode of Hh signaling in Th17 polarization but not Th1, Th2, and iTreg polarization.

**Hh signaling controls Th17 polarization in vivo.** Th17 cells are critical drivers of IBD including pathological inflammation of the small intestine. A model of small intestinal inflammation driven by Th17 cells was developed whereby the intraperitoneal injection of anti-CD3 antibodies leads to polarization and accumulation of Th17 cells in the small intestine within 3 days[17]. Using this model system, we treated wildtype C57BL/6 mice by oral gavage with either carrier or Hh inhibitor vismodegib, which has been clinically approved for oral use (Fig. 6a). Strikingly, intestinal inflammation was ameliorated in vismodegib-treated mice. Mice, in which Hh signaling had been inhibited, showed reduced weight loss (Fig. 6b) and less thickening and shortening (lower Weight/Length ratio) of the small intestine (Fig. 6c). Importantly, vismodegib-treated mice had significantly fewer Th17 intraepithelial lymphocytes (IELs), while IFNγ[+] IL17[neg] IELs were non-significantly decreased in the numbers (Fig. 6d and Supplementary Fig. 8). This was accompanied by a significant reduction of IL-17a levels in the serum (Fig. 6e).

In order to demonstrate genetically that Hh signaling is instrumental to induce Th17-mediated colitis we used cells from our inducible *Ihh* KO mice in a T cell adoptive transfer colitis model[18]. This model has been extensively used to study Th17 function in vivo. The strength of the model is that intestinal inflammation is driven by de novo Th17 polarization of the adoptively transferred naïve CD4[+] T cells. In addition, this model recapitulates a key function of Th17 cells in vivo, namely the plasticity to transdifferentiate to IL-17a[−]/IFNγ[+] "ex-Th17" cells. We treated *CD4CreER[T2] Ihh[fl/fl] tdTom[+]* (KO) and *CD4CreER[T2] Ihh[fl/+]tdTom[+]* (HET) mice with tamoxifen for four days to induce the deletion of *Ihh* in CD4[+] T cells. Next, we flow sorted naïve, tdTom[+] CD45RB[hi] CD25[−] CD4[+] T cells from HET and KO mice, adoptively transferred them into T-cell deficient

*Rag2[−/−]* recipient mice and assessed inflammatory disease activity after 6 weeks (Fig. 6f).

While the transfer of *Ihh* HET CD4[+] T cells potently induced colitis, transfer of *Ihh* KO CD4[+] T cells showed strongly diminished colitis indicated by reduced weight loss (Fig. 6g), reduced colon thickening/shortening (Fig. 6h) and reduced mucosal infiltration of T cells expressing IL-17a, IL-22 and IFNγ in the colon (Fig. 6i and Supplementary Fig. 9). In line with this, histological analysis revealed that *Ihh* HET CD4[+] T cells potently induced colitis in *Rag2[−/−]* recipient mice which was characterized by mononuclear and polymorpho-nuclear mucosal to submucosal infiltration of inflammatory cells, crypt hyperplasia, and epithelial injury. By contrast, transferred *Ihh* KO CD4[+] T cells were strongly impaired in their ability to induce colitis and mice showed minimal signs of inflammatory disease in the colon (Fig. 6j). Given that all recipient mice were wildtype for Ihh and able to secrete Ihh ligands, this finding affirms our previous finding in vitro that only the T-cell-intrinsic Ihh pool can promote Th17 polarization even in the presence of exogenous Hh ligands. No differences were observed between *Ihh* HET and KO CD4[+] T cells in mesenteric lymph nodes and spleens (Supplementary Fig. 10a, b), enumeration of IL-17a[−] IFNγ[−] showed no significant difference across all organs (Supplementary Fig. 10c), and Treg numbers were not affected (Supplementary Fig. 10d), demonstrating that the effects shown in Fig. 6 are specific to Th17 polarization in the intestine and not a general failure of *Ihh* KO CD4[+] T cells to home and expand in the lymphopenic host.

Taken together, we demonstrate that cell-autonomous Hh signaling is a critical driver of Th17 polarization in chronic intestinal inflammation mouse models of human IBD. Importantly, we show that the small molecule Hh inhibitor vismodegib, which is clinically approved by the EMA/FDA, effectively protected against intestinal inflammation in a model of IBD.

**Hh signaling does not act via known TFs or proximal cytokine signaling.** Having established that Hh signaling is functionally important for Th17 polarization, we sought to determine the mechanism for this effect. Downstream of the IL-6 and IL-23 cytokine receptors, Th17 polarization is initiated by STAT3 phosphorylation which in turn translocates to the nucleus and initiates the transcription of key Th17 polarizing transcription factors (TFs). We found that phosphoSTAT3 (pSTAT3) levels are unaltered upon IL-6 stimulation when we inhibit the Hh pathway with cyclopamine in Th17 cells, indicating that Th17-initiating cytokine signaling is not controlled downstream of the Hh pathway (Fig. 7a).

Next, we profiled the expression of well-known Th17 TFs[19] in Th17 cells treated with Hh inhibitor cyclopamine or carrier control (Fig. 7b). While *Il17a* mRNA was markedly reduced in our treatment conditions, Th17-regulating TFs *Rorct*, *Rora*, *Irf4*, *Runx1*, and *Batf* were not majorly affected and neither was a recently-identified Th17-associated TF *Vax2*[20]. This suggested that the Hh signaling may be exerting its effect on Th17 polarization through an unstudied regulator of Th17 function rather than through the previously described Th17-regulating transcription factors.

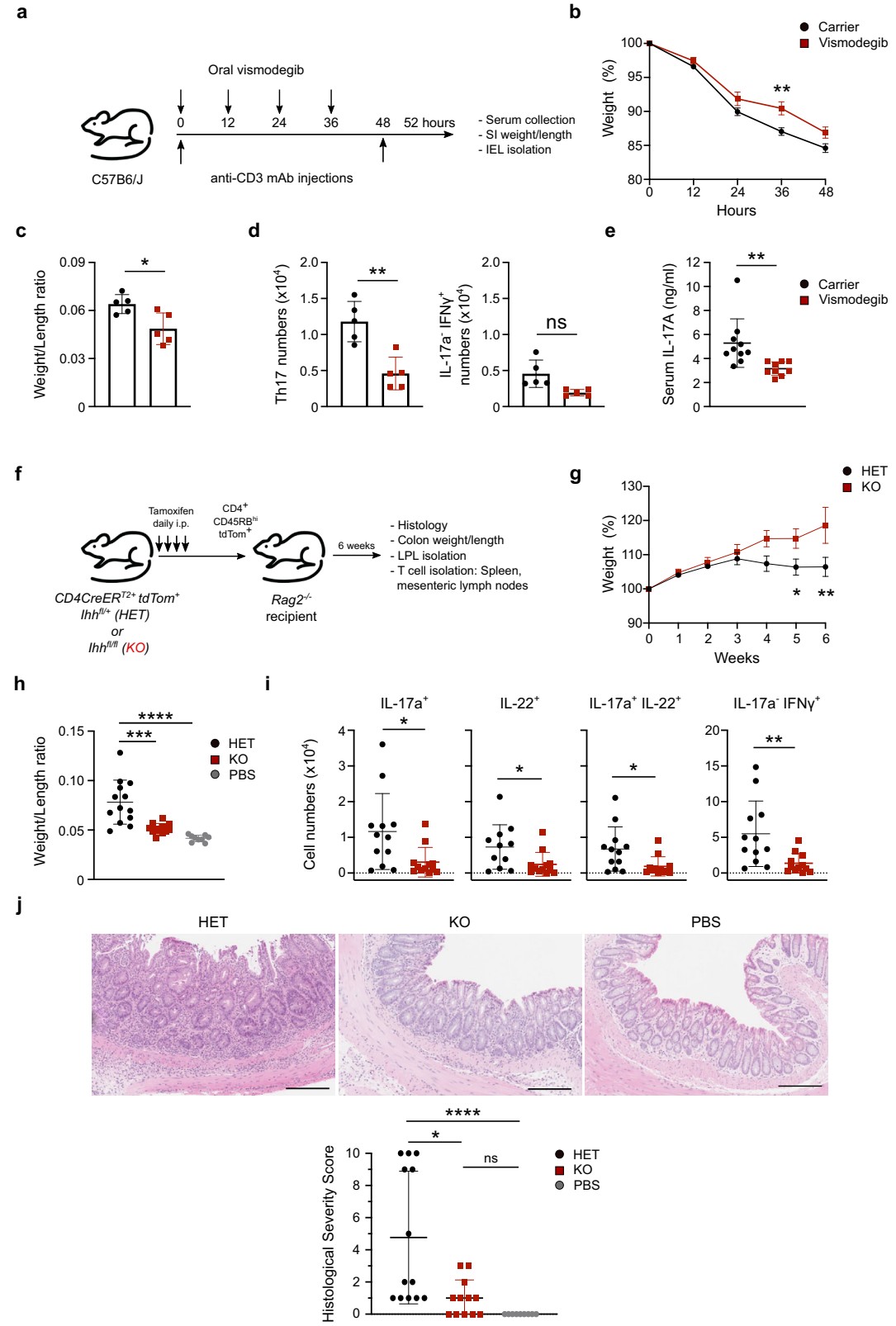

To interrogate the mechanism of Hh-mediated Th17 polarization more globally we performed RNA-Seq analysis of Th17 cells polarized in the presence of carrier control or Hh inhibitor cyclopamine. As expected, expression of known Hh target genes[21] like *Smo*, *Jag2*, *Prdm1*, and *Fst* was significantly reduced in Hh inhibitor-treated cells (Fig. 7c). The expression level of key well-known Th17 TFs in Hh inhibitor-treated Th17 cells was ambiguous: while for example *Rora* and *Batf* were significantly reduced upon inhibitor treatment, *Junb* and *Runx1* were increased (Fig. 7c), which is again consistent with Hh signaling enacting its effect on Th17 polarization through a previously unstudied regulator. Th17 cells have stem-cell like features: plasticity and self-renewal[22]. Interestingly, we find that *Tcf7*, the key stem cell-associated gene in Th17 cells[23], as well as other Wnt

**Fig. 6 Hh signaling is critical for Th17 responses in vivo. a** C57BL/6 mice were injected *i.p.* with 20 µg anti-CD3 monoclonal antibody (Clone: 145-2C11) at 0 and 48 h and dosed every 12 h by oral gavage with 100 mg/kg vismodegib or carrier control. Mice were harvested at 52 h. **b** Weight loss during the course of the experiment, relative to starting weight on day 0 (=100%). Pooled data from three independent experiments. **c** Small Intestine weight/length ratios. Representative data shown from one of three independent experiments. **d** IEL Th17 and IFNγ+ IL-17a− CD4+ T cell numbers. Representative data shown from one of three independent experiments. Gating strategy is shown in Supplementary Fig. 8. **e** Serum IL-17a concentrations at 52 h. Pooled data from three independent experiments. **f** *Rag2*−/− mice were injected *i.p.* with 4 × 10⁵ CD45RB^hi CD25^neg tdTom+ CD4+ T cells isolated from the spleens and peripheral lymph nodes of tamoxifen-treated *CD4CreER^T2Cre Ihh^fl/+ tdTom+* (HET) or *CD4CreER ^T2 Ihh^fl/fl tdTom+* (KO) mice. Mice were harvested at 6 weeks. **g–j** Pooled data from three independent experiments shown. **g** Weight loss during the course of the experiment , relative to starting weight on day 0 (=100%). **h** Colon weight/length ratios. (**i**) Numbers of IL-17a+, IL-22+, IL-17a+ IL-22+, IFNγ+ IL-17a^neg T cells isolated from colonic lamina propria. Gating strategy is shown in Supplementary Fig. 9. **j** Panels on the top show representative H&E staining of the recipient mouse colons. The panel on the bottom shows the quantification of histological severity scored blindly by a gastroenterologist. Scale bars: 200 µm. Data are means +/− SEM (**b**, **g**). Rest are means +/− SD. *p*-values were calculated using an unpaired two-tailed Student's *t* test. *p < 0.05, **p < 0.01, ***p < 0.001. *p*-values were calculated for panels (**c**, **d**, **e**, **i**) using an unpaired two-tailed Student's *t* test, **j** using a Kruskal–Wallis test, **h** a one-way ANOVA, or **b**, **g** a two-way ANOVA with Sidak correction. *p < 0.05, **p < 0.01, ***p < 0.001. ns = not statistically significant. Source data are provided in the source data file.

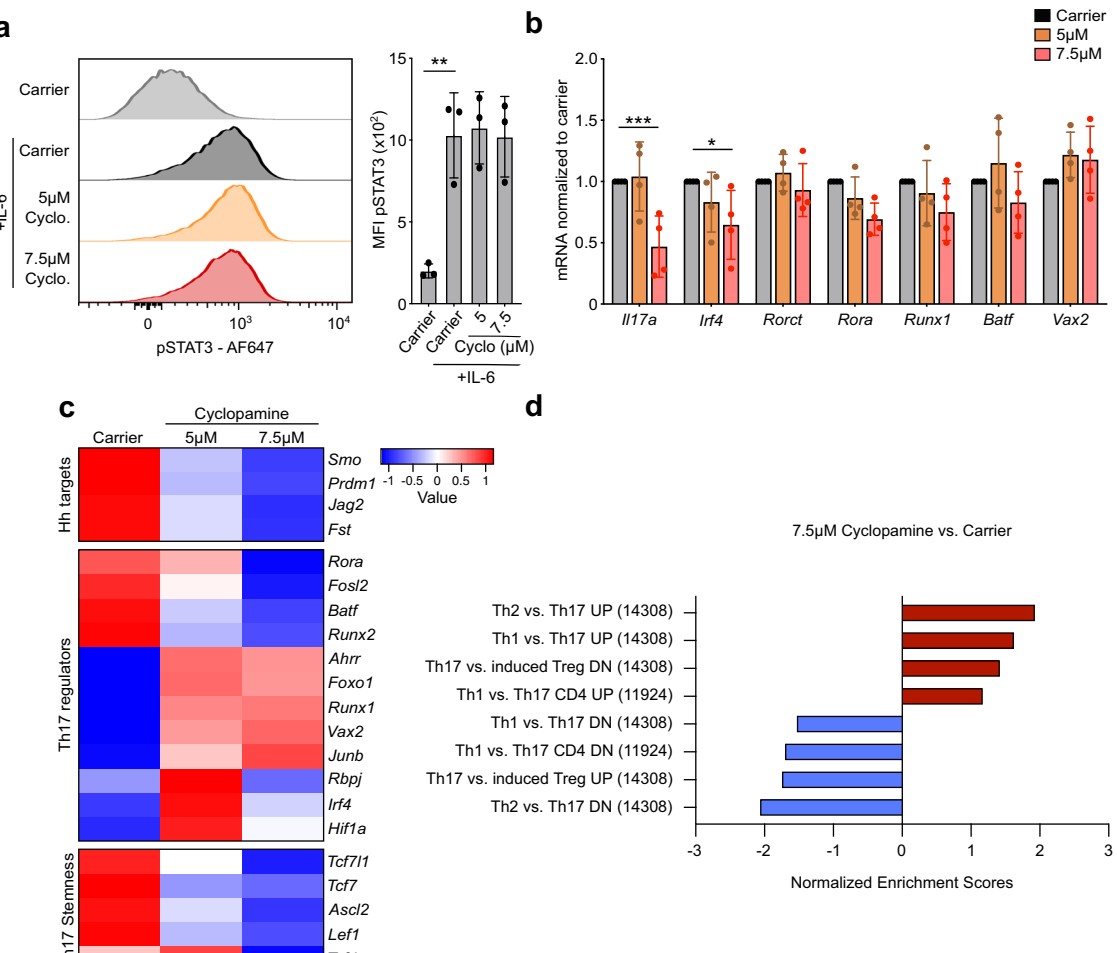

**Fig. 7 Mechanistic analysis of Th17-related signaling nodes.** Naïve CD4+ T cells were purified from spleen and peripheral lymph nodes of C57BL/6 mice and stimulated under Th17 polarizing conditions in the presence of the indicated doses of cyclopamine or carrier control for three days. **a** On day 5, cells were treated with 100 ng/ml IL-6 for 15 min and harvested for analysis of pSTAT3 levels by flow cytometry. Quantitation of pSTAT3 staining is shown on the right. *n* = 3 independent experiments. **b** Cells were harvested for analysis at day 3. The expression of *Rora, Rorct, Irf4, Runx1, Batf, Vax2,* and *Il17a* was analyzed by qRT-PCR. Data is normalized to *Tbp* as a reference gene. Similar results were obtained when *CD3ε* was used as a reference gene. *n* = 4 independent experiments. **c**, **d** Th17 cells were polarized as described above stimulated in the presence of the indicated doses of cyclopamine or carrier control for three days, and harvested at day 3 for RNA-Seq analysis. Six samples/group. **c** RNA-Seq analysis of Hh target gene, Th17 regulator, and Th17 stemness gene expression. **d** Normalized Enrichment Scores (NES) from GSEA of RNA-Seq data demonstrating a loss of Th17 identity upon Hh inhibitor treatment. Data are means +/− SD. *p*-values were calculated using a one-way (**a**) or two-way (**b**) ANOVA with Tukey's multiple comparisons test. *p < 0.05, **p < 0.01, ***p < 0.001. Source data are provided in the source data file.

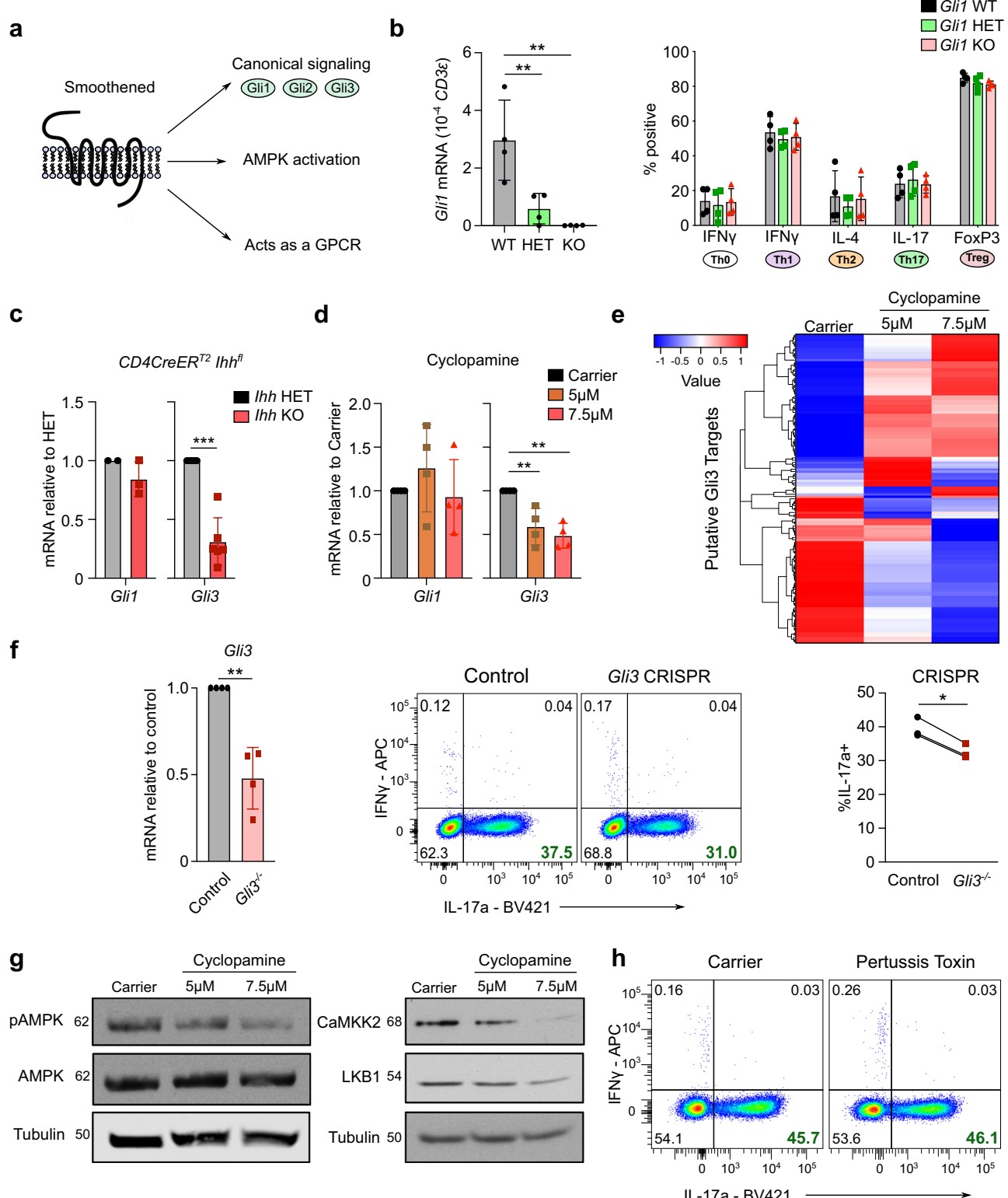

target genes were downregulated in Hh inhibitor-treated cells (Fig. 7c) suggesting that Hh signaling may be critical to endow Th17 cells with stem cell properties.

Overall, Gene Set Enrichment Analysis (GSEA) certified that Hh inhibitor-treated cells moved away from a Th17 transcriptional profile (Fig. 7d and Supplementary Fig. 11). This demonstrates that the global downstream Th17 transcriptional profile is lost upon Hh inhibition, even though the levels of key

Th17 TFs remain largely unchanged. We show that this effect of Hh signaling blockade is not due to any effects on proximal cytokine signaling or expression levels of key Th17 TFs.

**Hh signaling controls Th17 polarization via Gli3 and AMPK.** Smo is the key signal transducer of the Hh pathway and can function in both a canonical and non-canonical fashion (Fig. 8a).

**Fig. 8 Canonical and non-canonical Hh signaling contribute to Th17 polarization. a** Schematic overview of canonical/non-canonical Hedgehog signaling. **b** Naïve CD4+ T cells were purified from spleen/lymph nodes of *Gli1eGFP*+/+ (WT), *Gli1eGFP*+/− (HET) or *Gli1eGFP*−/− (KO) mice. Cells were polarized to Th0, Th1, Th2, Th17 or iTregs. Cells were assessed for of *Gli1* mRNA expression on day 3 (Th17, left) or harvested for flow cytometry on day 5 (right). n = 4 mice per genotype. **c** Th17 cells were polarized from tamoxifen-treated *CD4CreER*T2+ *Ihh*+/fl (HET) or *CD4CreER*T2+ *Ihh*fl/fl (KO) mice and analysis of *Gli1* and *Gli3* expression was performed by qRT-PCR at day 3. n = 2–6 mice. **d** Th17 cells were polarized from C57BL/6 mice in the presence of the indicated dose of cyclopamine or carrier control for three days. Data are normalized to *Tbp* as a reference gene. n = 4 independent experiments. **e** Expression of all putative Gli3 target genes predicted by Miraldi et al.[20] in RNA-Seq analysis from Fig. 7c. Th17 cells were polarized in the presence of the indicated doses of cyclopamine or carrier control for three days and harvested at day 3. Six samples/group. **f–h** Naïve CD4+ T cells were purified from spleen/lymph nodes of C57BL/6 mice and stimulated under Th17 polarizing conditions. **f** After 24 h CRISPR/Cas9-RNP complexes targeting *Gli3* were electroporated. Cells were harvested for qRT-PCR analysis of *Gli3* at day 3 (left panel) and for flow cytometry analysis on day 5 (middle panel). Right: Quantitation of IL-17a expression. n = 3 independent experiments. **g** Naïve CD4+ T cells were stimulated in the presence of the indicated doses of cyclopamine or carrier control for three days. Immunoblot analysis of Th17 cells on day 3 for pAMPK, AMPK, CaMKK2, LKB1 and Tubulin is shown. n = 3 independent experiments. **h** Naïve CD4+ T cells were stimulated in the presence of 100 ng/ml pertussis toxin or carrier control for five days. Cells were analyzed by flow cytometry on day 5. n = 3 independent experiments. Data are means +/− SD. *p*-values were calculated in (**b**, **d**) using a one-way/two-way ANOVA with Tukey's multiple comparison test or an unpaired two-tailed Student's *t* test (**c**, **f**). \**p* < 0.05, \*\**p* < 0.01, \*\*\**p* < 0.001. Source data are provided in the source data file.

Smo activates Gli transcription factors as part of the canonical pathway, but has also been shown to activate AMPK[24] or act as a GPCR through activation of Gi proteins when associated with non-canonical Hh signaling[25].

Three transcription factors are associated with Hh signaling, but only Gli1 and Gli3 are expressed in CD4+ T cells (Fig. 1e and Supplementary Fig. 2a). To dissect their relevance, we first investigated the role of Gli1 in CD4+ T cell polarization. We isolated naïve CD4+ T cells from *Gli1* knockout mice as well as heterozygous and wildtype littermates (validation, Supplementary Fig. 7c and Fig. 8b left panel) and polarized the cells into Th0, Th1, Th2, Th17, and iTreg cells. Surprisingly, no differences in Th polarization were observed between *Gli1* WT, HET, and KO mice showing that Gli1 is functionally not important for Th17 polarization (Fig. 8b, right panel).

Thus, we asked whether Gli3 downstream of canonical Hh signaling was important for Th17 polarization. We polarized Th17 cells from conditional *Ihh* KO or HET control mice as well as Hh inhibitor or carrier control treated Th17 cells and assessed expression levels of *Gli1* and *Gli3* mRNA. While *Ihh* KO and inhibitor treated Th17 cells showed no difference in *Gli1* levels compared to their controls, *Gli3* mRNA was 69% and 52% reduced, respectively (Fig. 8c, d). Although *Gli1* mRNA is a reporter of active Hh signaling in many other tissues[14], we show here that in Th17 cells this is not the case and *Gli3* transcript levels instead serve as a faithful reporter of active Hh signaling.

A pioneering bioinformatic analysis by Miraldi et al. using ATACSeq datasets of Th17 cells to analyze Th17 transcriptional regulatory networks identified the top 30 "core" Th17 TFs[20]. Among these "core" TFs were the well-established Th17-inducing TFs RORγt, RORα, STAT3, Maf, and HIF-1α. Strikingly, the authors identified Gli3 among these top 30 "core" Th17 TFs and predicted putative gene targets of Gli3. We went back to our RNA-Seq analysis of Hh inhibitor-treated Th17 cells and assessed the expression of the predicted Gli3 targets. Gli3 can act in two opposing functions at target gene promoters: as a transcriptional repressor when truncated (GliR) or as a transcriptional activator (GliA)[26]. The balance between GliR and GliA in the nucleus shapes the Hh response. We found that upon Hh inhibitor treatment the majority of the predicted Gli3 target genes increase or decrease in a dose-dependent manner (Fig. 8e and Supplementary Fig. 12) indicating that indeed Hh-induced Gli3 regulates the transcription of these target genes to control Th17 polarization. A notable target is *Cd5l* which is upregulated upon Hh inhibition as well as upon genetic knockout of Ihh and has been shown to be one of the major negative regulators of Th17 pathogenicity[27] (Supplementary Fig. 12).

In order to interrogate the functional relevance of Gli3 we used CRISPR to delete *Gli3* in primary CD4+ T cells and achieved a

52% reduction of *Gli3* RNA by day 3 across the total CD4+ T cell pool (Fig. 8f). Importantly, knockout of *Gli3* in a proportion of primary Th17 cells led to a statistically significant reduction in IL-17a producing cells. Taken together, we have shown that Gli3 is the only Hh transcription factor that is expressed and functionally important in Th17 cells.

Apart from its central role in canonical Hh signaling, non-canonical roles of Smo have recently emerged and Smo has been implicated as a regulator of AMPK phosphorylation in brown adipose tissue and neurons[24,28]. Since AMPK has been shown to be important for promoting Th17 polarization and pathogenicity in both mouse and human[29,30], we investigated whether Smo would regulate activating AMPK phosphorylation marks in Th17 cells. To this end, we analyzed Hh-inhibited and control CD4+ T cells by western blot for pAMPK and observed a dose-dependent decrease of pAMPK upon Hh signaling inhibition (Fig. 8g). Furthermore, the expression of the upstream AMPK regulating kinases CaMKK2 and LKB1 was also reduced as had been previously described in adipose tissue[24].

Another non-canonical signaling mode of Smo is its direct function as a GPCR[25], which is highly sensitive to pertussis toxin inhibition. To investigate whether this signaling mode is important for Th17 polarization we treated naïve CD4+ T cells for 5 days with pertussis toxin under Th17 polarizing conditions. Pertussis toxin treatment did not affect viability or proliferation and had no effect on IL-17a production (Fig. 8h).

Taken together, we have uncovered Gli3 as a fundamental TF implicated in Th17 polarization that works together with non-canonical Hh signaling via pAMPK.

**Hh components are upregulated in human ulcerative colitis.** Next, we sought to determine whether Hh signaling components are expressed in human T cells, specifically in human intestinal T cell populations. For this, we analyzed an RNA-Seq dataset of FACS-sorted CD4+ T cell populations from healthy donor blood, intestinal epithelial lymphocytes (IELs) and lamina propria lymphocytes (LPLs)[31]. This dataset showed that the major components of the Hh pathway—endogenous ligand (IHH), signal transducer (SMO) and key transcription factor (GLI3)—are expressed in human intestinal CD4+ T cells isolated from healthy individuals (Fig. 9a, b).

Given our finding that Hedgehog signaling is critically important in driving Th17 responses in mouse models of intestinal inflammation, we next aimed to analyze Hh component expression in rectal biopsies from patients with ulcerative colitis and matched healthy controls (Fig. 9a, b). We chose two large cohorts of patients with ulcerative colitis and found significant upregulation of *SMO* and *GLI3* (Fig. 9c and Supplementary

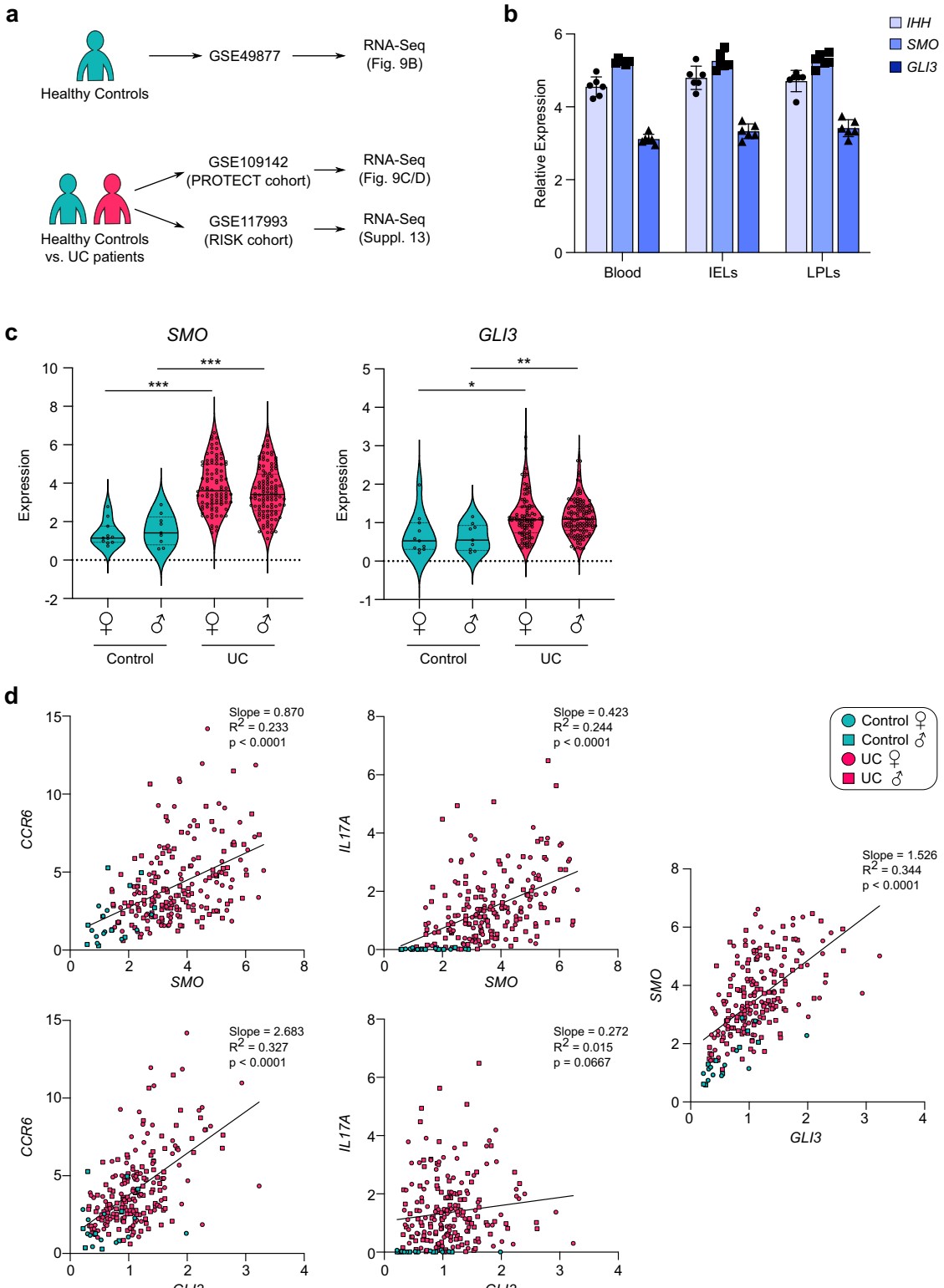

**Fig. 9 Hedgehog signaling components are expressed in human CD4$^+$ T cells in the intestine and are upregulated in ulcerative colitis patients.**
Analysis of human RNA-Seq datasets summarized in panel (**a**). **b** Expression of *IHH, SMO* and *GLI3* mRNA from FACS-sorted CD4$^+$ T cells isolated from blood as well as intraepithelial lymphocytes (IELs) and lamina propria lymphocytes (LPLs) isolated from the terminal ileum from healthy human donors in a study from Raine et al. (2015)[31]. *n* = 6. Error bars show SD. **c, d** Expression of *SMO* and *GLI3* mRNA from human rectal biopsy samples from patients with male (♂) or female (♀) ulcerative colitis (UC) or healthy controls recruited as part of the PROTECT study (GSE109142). Each datapoint represents an individual patient. **c** Shows expression of *SMO* and *GLI3* across all groups. *p*-values were calculated using a limma based moderated *t* test. *\*p* < 0.05, *\*\*p* < 0.01, *\*\*\*p* < 0.001. **d** Shows correlation analysis of *SMO, GLI3* as well as Th17-related transcripts *CCR6* and *IL17A* respectively with simple linear regression analysis. Source data are provided in the source data file.

Fig. 13a) in ulcerative colitis patients compared to healthy controls[32,33]. Interestingly, in the setting of human ulcerative colitis, expression levels of Hedgehog components positively correlated with the expression levels of key Th17 markers such as *IL17A* and *CCR6* (Fig. 9d and Supplementary Fig. 13b). These results are consistent with Hedgehog signaling playing an important role in human Th17 polarization in the context of IBD.

## Discussion

Our work uncovers that Hh signaling selectively controls Th17 polarization, but not polarization into Th1, Th2, and Treg lineages. We show that the critical Hh pathway components Smo and Gli3 are both selectively upregulated in Th17 cells at the time of lineage determination and are induced by Th17-polarizing cytokines IL-6 and TGFβ, respectively. The TGFβ-mediated induction of Gli3 is in line with previous findings in the literature that Gli2 is a direct target of TGFβ signaling[34]. Ihh is the only Hh ligand expressed in Th17 cells. Furthermore, utilizing multiple strategies, namely CD4+ T cells from a conditional CD4-specific *Ihh* KO mouse model as well as treatment with recombinant active Ihh or Hh blocking antibody 5E1, we show that Hh signaling in CD4+ T cells is solely driven by endogenous Ihh—likely acting intracellularly—and does not depend on exogenous Hh ligands. Even though, interestingly, Th17 cells appear capable of secreting Ihh when Ihh is overexpressed in Th17 cells under the control of a retroviral promoter (Supplementary Fig. 2c). Our validation of a cell-autonomous mode of Hh signaling in CD4+ T cells functionally confirms and extends the previous suggestion that Hh signaling may operate intracellularly in CD8+ T cells[11]. This mode of signaling might present a unique characteristic of the Hh pathway in lymphocytes to ensure that signaling is independent of fluctuating exogenous ligand gradients.

We identify Gli3 as a key transcription factor that controls Th17 lineage polarization. Our observation functionally validates an ATACSeq-based bioinformatic model that suggested Gli3 is part of the top 30 "core" TFs of the Th17 transcriptional regulatory network[20]. We show that Hh-induced Gli3 regulates the transcription of previously predicted Gli3 target genes (Supplementary Fig. 12) known to control Th17 polarization[20]. This crucial role for Gli3 seems to be restricted to CD4+ T cells, since Gli3 expression has not been reported in CD8+ T cells[11].

Furthermore, we show that Hh signaling influences the metabolic regulation of Th17 cells through a known non-canonical Hh signaling axis via CaMKK2/LKB1 and pAMPK in adipose tissue[24]. This is in line with published data indicating a role of pAMPK in human and murine Th17 polarization[29,30] and in the pathogenicity of Th17 cells in a model of adoptive T cell transfer colitis[29]. Our model is summarized in Suppl. Fig. 14.

Hh signaling has been implicated in CD4+ T cell polarization of Th2 but not Th1 cells[12]. Furmanski et al. used mice expressing transgenic Gli2A (activator) or Gli2R (repressor) and showed an effect of exogenous Shh on Th2 polarization. Our findings that Gli2 transcripts are absent in murine Th cultures and that Th17 cells do not respond to exogenous Ihh ligand might resolve these differences. In addition, it has been shown that exogenous recombinant Shh is unable to induce Th17 polarization in human CD4+ T cells[35] supporting our model of a unique cell-autonomous Hh signaling pathway in T lymphocytes.

Th17 cells show remarkable plasticity compared to other Th subsets due to their stem cell potential. Depending on the microenvironment, Th17 cells have the ability to convert into nearly all other CD4+ Th lineages[22]. This phenomenon is particularly well studied in autoimmune disease, cancer and infection where Th17 cell conversion into Th1 cells is critical for the overall CD4 effector response[23,36]. Hh signaling is functionally important

for the maintenance of various stem cell compartments throughout the body, as well as cancer stem cells[37–41] and it remains to be seen whether Hh signaling can provide Th17 cells with stem cell potential in addition to being crucial for Th17 polarization. Intriguingly, we see that cells treated with Hh inhibitors lose expression of stem-cell-associated Wnt target genes *Lef1, Tcf4, Tcf7l1, Ascl2*, as well as *Tcf7* which was identified as the main marker of Th17 stemness (Fig. 7c)[23]. In our in vivo adoptive transfer colitis experiment we observed a decrease not only in IL-17a+ cells but also in IL-17a− IFNγ+ cells (Fig. 6i). We speculate that the reduction in IL-17a− IFNγ+ cells may be either due to an impairment of Th17-to-Th1-conversion or be due to a reduced number of Th17 cells in the colon providing a smaller "source" of Th1 cells via Th17-to-Th1-conversion.

Therefore, the defect in Th17 polarization observed in vivo upon *Ihh* ablation may ameliorate intestinal inflammation by a number of mechanisms including reduced IL-17a levels, reduced CCR6 levels affecting T cell recruitment and/or reduced Th1 levels due to reduced/defective Th17-to-Th1 conversion.

Due to the central role that pathogenic Th17 cells play in numerous autoimmune diseases, including IBD, rheumatoid arthritis, multiple sclerosis and psoriasis[42], there has been great interest in developing therapeutic interventions that specifically target Th17 polarization and pathogenicity. The importance of Th17 cells in COVID-19 pathogenesis has further increased this interest[43,44].

There is a shortage of small molecule inhibitors that target Th17 polarization and effector function. A seminal drug screen for RORγt antagonists identified digoxin as a potent inhibitor able to suppress Th17 responses in vitro and in vivo[45]. More recently, bromodomain inhibitors JQ1 and MS402 were found to block Th17 polarization in pre-clinical models of intestinal inflammation[46,47]. However, the narrow therapeutic window of digoxin and the broad-ranging effects of targeting epigenetic regulators with bromodomain inhibitors make their use in the clinic for the treatment of Th17-driven diseases challenging. Here, we not only show that central Hh signaling components are expressed in human intestinal CD4+ T cells and that expression of *SMO* and *GLI3* is upregulated in patients with IBD correlating with Th17 infiltration, but also demonstrate that clinically approved Hh inhibitor vismodegib specifically and potently inhibits Th17 polarization in vivo. Thus, we open up the real possibility of treating autoimmune diseases such as IBD with Hh inhibitors.

## Methods

**Mice.** All housing and procedures were performed in strict accordance with the United Kingdom Home Office Regulations, the Danish Council for Animal Testing/The Supervisory Authority on Animal Testing, and the Cancer Research UK Cambridge Institute Animal Welfare and Ethical Body (AWERB). RAG2KO were a generous gift from Suzanne Turner (University of Cambridge) and OTI mice were purchased from Charles Rivers Inc., UK (C57BL/6-Tg(TcraTcrb)1100Mjb/j, Stock no. 003831). OTI RAG2KO mice were generated from these. *Gli1-eGFP* mice were a generous gift from Alexandra Joyner (Sloan Kettering Institute)[48] and were backcrossed onto the C57BL/6J background (Charles Rivers Inc., UK) for more than 11 generations. *dLckCre* and *ROSA26 loxP-STOP-loxP-tdTom* mice were a generous gift from Randall Johnson and Douglas Winton (University of Cambridge), respectively. *Smo^{f/f}* (Stock no. Smo^{tm2Amc}/J, 004526), *Ihh^{f/+}* (*Ihh^{tm1Blan}*/J, Stock no. 024327) and *CD4CreER^{T2}* (Tg(Cd4-cre/ERT2)11Gnri, Stock no. 022356) mice were purchased from Charles Rivers Inc., UK. *Smo^{f/f}* mice were back-crossed to the C57BL/6J background (Charles Rivers Inc., UK)) for more than 10 generations and crossed to *ROSA26*tdTom and *CD4CreER^{T2}* mice to generate *CD4CreER^{T2}/ROSA26*tdTom/*Smo^{f/f}* mice or with *dLckCre* to generate *dLckCre/ROSA26*tdTom/Smo^{f/f} mice. *Ihh^{f/f}* mice (C57BL/6J background) were crossed onto *CD4CreER^{T2}/ROSA26*tdTom mice (C57BL/6J background). Mice were genotyped using Transnetyx, maintained at the CRUK Cambridge Institute/University of Cambridge. Mice were harvested from 6 weeks' of age up to 6 months of age; both male and female mice were used. Different genotypes of the same sex and from the same litter were compared where possible. Mice were housed under specific-pathogen free conditions at the University of Cambridge, Cancer Research

UK Cambridge Institute. Mouse housing conditions were in a 12 h light-dark cycle at room temperature (20–23 °C) with 40–70% humidity with *ad libitum* access to food and filtered water. Environmental enrichment, such as mouse houses or fun tunnels and nesting material were provided to all animals.

**Murine CD4$^+$ T cell in vitro polarization.** Spleens and lymph nodes were harvested and placed into ice-cold sterile DPBS (Gibco, cat no. 14190094, hereafter referred to as PBS). Spleens were dissociated through a 70 μm filter (Greiner, cat no. 542070) using the top of the plunger from a 2 ml syringe. Filters were washed with ice-cold, sterile MACS Buffer, made with PBS + 2% fetal calf serum (FCS, Biosera, cat no. 1001, 500 ml) + 2 mM EDTA and then centrifuged at $300 \times g$ for 10 min at 4 °C. Naïve murine CD4$^+$ T cells were isolated using a negative-selection MACS approach (Naïve CD4$^+$ T Cell Isolation Kit, Miltenyi Biotec, cat no. 130-104-453). MACS separation was performed by loading the labeled cell suspension through a 50 μm sterile CellTrics Partec filter (Wolflabs, cat no. 04-004-2327) onto the LS column (Miltenyi Biotec, cat no. 130-042-401). The purity of the sorted populations was above 95%.

CD4$^+$ T cells were plated onto a 96-well plate pre-coated overnight at 4 °C with 2.5 μg/ml anti-CD3ε (Clone: 500A2, eBioscience, cat no. 16-0033) and 2 μg/ml anti-CD28 antibody (Clone: 37.51, eBioscience, cat no. 16-0281). For Th polarization experiments CD4$^+$ T cells were plated at a final concentration of $1 \times 10^6$ cells/ml in complete IMDM supplemented with cytokines/blocking antibodies to induce lineage polarization (see Supplementary Table 1 for details).

**Cell culture.** Purified naïve CD4$^+$ T cells were resuspended in IMDM (Gibco, cat no. 12440-053) supplemented with 5% heat-inactivated batch-tested FCS (Biosera, cat no. 1001-500 ml), 10 μM β-Mercaptoethanol (50 mM, Gibco, cat no. 31350-010) and 100 U/ml Penicillin/Streptomycin (10,000 U/ml, Gibco, cat no.15140-122).

*Smo* KO MEFs were generated by James Chen (Stanford University) and kindly provided by Natalia A Riobo-Del Galdo (University of Leeds). WT MEFs were a generous gift from Jane Goodall (University of Cambridge). MEFs were cultured in DMEM medium (DMEM, Gibco, cat no. 10566-016; 500 ml) supplemented with 10% heat-inactivated, batch-tested FCS (Biosera, cat no. 1001), 10 mM HEPES (Sigma, cat no. H0887), 1:100 MEM Non-essential amino acids solution (Thermo, cat no. 11140050) and 100 U/ml penicillin/streptomycin (Gibco, cat no. 15140-122; 10,000 U/ml). All cell lines tested mycoplasma negative (MycoProbe® Mycoplasma Detection Kit, R&D systems). HEK 293T cells were a generous gift from James Brenton (University of Cambridge). HEK 293T cells cultured in DMEM medium (DMEM, Gibco, cat no. 10566-016; 500 ml) supplemented with 10% heat-inactivated, batch-tested FCS (Biosera, cat no. 1001). Where indicated cells were cultured in the presence of the recombinant N-terminal active fragment of Indian Hedgehog (Bio-Techne, cat no. 1705-HH/CF) or carrier control.

Cells were grown in a humidified incubator at 37 °C and 5% CO$_2$. Cell lines were authenticated by in-house STR analysis. All cell lines tested mycoplasma negative (MycoProbe Mycoplasma Detection Kit, R&D Systems).

**CRISPR of primary naïve CD4$^+$ T cells.** Alt-R® CRISPR-Cas9 crRNA (IDT) and tracrRNA (IDT, cat no. 1072534) were reconstituted at 100 μM in nuclease-free duplex buffer (IDT). 1.9 μl of each stock solution was mixed with 8.7 μl duplex buffer. Samples were heated in a thermal cycler at 95 °C for 5 min and left for 10 min at room temperature. 10.5 μl Buffer T (Thermo, cat no. MPK10096) was added as well as 2 μl TrueCut™ Cas9 Protein v2 (5 mg/ml, Thermo, cat no. A36499) which was pipetted very slowly in a circular motion to ensure optimal solubility. The mixture was incubated at 37 °C for 10 min to assemble the ribonucleoprotein (RNP) complex.

Naïve CD4 + T cells were isolated from C57BL/6 spleens and stimulated with plate bound anti-CD3ε/CD28 antibodies in the presence of Th17 polarizing cytokines. After 24 h of stimulation, cells were washed twice in pre-warmed PBS (Gibco) prior to resuspension in 80 μl Buffer T (1 million cells/electroporation reaction). The suspension was briefly mixed with the RNP complex solution prior to electroporation with the Neon™ electroporation system in 100 μl electroporation tips (Thermo, cat no. MPK10096) with three pulses of 1600 V each with a pulse width of 10 ms. Cells were left to recover in complete IMDM in the absence of antibiotics for 20 min. Cells were then centrifuged at 480 g for 5 min and returned into culture to an anti-CD3ε/CD28 antibody coated plate in complete IMDM (without antibiotics) supplemented with polarizing cytokines at the concentrations mentioned previously. Cells were centrifuged at $200 \times g$ for 1 min for quick adherence to the coated plate. Antibiotics were re-added after six hours at the indicated concentrations.

*Gli3* was targeted with two crRNA sequences:
Guide 1: 5′-GCATATGAGAAGACACACTG-3′
Guide 2: 5′-CTCTCATCACTAGACGTCGA-3′
Two non-targeting crRNA sequences were used for the negative control (IDT, cat no. 1072544/10725455). Indels were validated using the Alt-R Genome Editing Detection Kit (IDT, cat no. 1075932) per the manufacturer's instructions.

**In vivo tamoxifen treatment.** 75 mg/kg/day of tamoxifen (Sigma, cat no. T5648) prepared in 100 μl of a 10% ethanol (Sigma, cat no. E7023), 90% corn oil (Sigma,

cat no. C8267, autoclaved) solution was given to the mice once a day for 4 consecutive days as an intraperitoneal injection. Tamoxifen was prepared by adding ethanol, then corn oil and sonicating for 30 min in a 37 °C water bath.

**Surface staining.** Cells were stained in 96-well round-bottom plates (Corning) or 5 ml polystyrene round bottom tubes (Fisher Scientific/Falcon, cat no. 352003). Samples were washed twice with ice-cold PBS and incubated for 10 min light protected at room temperature with Fixable Viability dye eFlour780, prepared at 1:1000 in PBS (eBioscience, cat no. 65-0865-18; 500 tests). Cells were then washed once with flow cytometry buffer, made with PBS (Gibco, cat no. 14190094) + 3% FCS (Biosera) + 0.05% Sodium Azide (Sigma, cat no. 71289), +2 mM EDTA (prepared in-house). Cells were then incubated with Fc block (1:100; Biolegend TruStain fcX anti-mouse CD16/32, cat no. 101320) for 5 min at room temperature protected from light. Next, cells were incubated with 50 μl of fluorophore-conjugated antibodies at the appropriate dilution (see Supplementary Table 2) for 20 min at 4 °C protected from light. Cells were washed twice with flow cytometry buffer and moved to 5 ml polystyrene round bottom tubes prior to immediate analysis (Fisher Scientific/Falcon) or fixation for intracellular staining.

**Intracellular staining.** For the staining of intracellular cytokines, cells were incubated at 37 °C for 4 h in the presence of 1 μg/ml Ionomycin (Sigma, cat no. I9657) and 50 ng/ml PMA (Sigma, cat no. P1585) and Golgistop (BD, cat no. 554724) added for the duration of stimulation per the manufacturer's instructions. Following surface marker staining (as above), cells were fixed with BD Cytofix/Cytoperm Plus Fixation Buffer (BD Biosciences, cat no. 554715) for 25 min at 4 °C protected from light. Samples were then spun down in a table-top centrifuge and washed once with permeabilization buffer (10×; BD Biosciences, cat no. 554715), prepared to a final concentration of 1× with Milli-Q Water (in-house). Cells were resuspended in 50 μl of permeabilization buffer containing fluorophore-conjugated antibodies at the appropriate concentration and incubated light protected at 4 °C for 30 min. Prior to analysis, cells were washed once in permeabilization buffer and once in flow cytometry buffer.

Flow cytometric analyses were conducted on a BD LSRII or BD LSR Symphony cell analyzer in the presence of AccuCheck counting beads (Thermo Fisher, cat no. PCB1000) where indicated, and data was analyzed with FlowJo software (Tree Star Inc., version 10.4).

**In vitro inhibitor and blocking antibody treatment.** CD4$^+$ T cells were treated with the Smo inhibitors cyclopamine (Alfa Aesar, cat no. J61528; 25 mg) or vismodegib (LC Laboratories, cat no. V-4050) at the concentrations indicated. Stock solutions were prepared in DMSO (Life Technologies) for vismodegib and ethanol for cyclopamine at 10 mM and diluted to working concentration in complete IMDM (Gibco). Vismodegib stocks were prepared from fresh vials and used within a week.

CD4$^+$ T cells were treated with the selective STAT3 inhibitor STATTIC (Bio-Techne, cat no. 2798) and TGFβ signaling inhibitor SB-505124 (Sigma, cat no. S4696) during Th17 polarization. In order to block extracellular Ihh ligand the monoclonal Hh blocking antibody (clone: 5E1, 2BScientific, cat no. Ab01175) or mouse IgG control antibody (Biolegend, cat no. 400370) was used. To block TGFβ signaling, 10 μg/ml anti-TGFβ blocking antibody (clone: 1D11, R&D Systems, cat no. MAB1835) was used.

**qRT-PCR.** Cells harvested for RNA extraction were washed twice in ice-cold PBS, snap-frozen as dry pellets, and stored at −80 °C. RNA was extracted using the RNAqeous™-Micro Total RNA Isolation Kit or with the Ambion PureLink RNA Isolation Kit according to the manufacturer's instructions. Thymus and testes were harvested from mice and homogenized with Precellys 1.4 mm ceramic beads in 2 ml tubes (KT03961-1-003.2, Bertin Instruments) using Precellys 24 lysis and homogenization unit (Bertin Instruments). Total RNA was extracted from homogenized samples using Ambion PureLink RNA Kit (12183025, Invitrogen) according to the manufacturer's instructions. RNA concentration was measured with a Nanodrop spectrophotometer (Labtech ND-1000) and samples were stored at −80 °C if not used immediately.

Reactions for qRT-PCR were set up in 384 well plate format on ice at a final volume of 10 μl using the One-Step qRT-PCR Kit (Thermo Fisher SuperScript III Platinum, cat no. 11732088). The reaction contained 5 μl 2× Reaction mix, 2–3 μl total RNA, 0.5 μl 20× Taqman probe (Thermo Fisher, see Supplementary Table 3), 0.2 μl Superscript III RT/Platinum Taq enzyme, and 1.3–2.3 μl nuclease-free ddH$_2$0 (total volume 10 μl).

Each sample was run in triplicate with *Tbp* and *CD3ε* used as housekeeping genes. In addition, each experiment included a non-template control and each probe was validated by a non-RT control where Platinum Taq DNA Polymerase (Life Tech, cat no. 10966018) was used instead of SuperScript Platinum III RT. Samples were run on a QuantStudio 6 Flex Real-Time PCR System (Thermo Fisher). Reverse transcription thermal cycling was set at 50 °C for 15 min, followed by 2 min at 95 °C and 45–50 cycles of PCR with 15 s at 95 °C, then 1 min at 60 °C.

Expression of the gene transcript of interest was calculated with the ΔCt method[49]. The cycle threshold (Ct) value from the gene of interest was subtracted from the

housekeeping gene and transformed with a factor of 2^(−ΔCt) to give the fold expression relative to the housekeeping gene.

**Monoclonal antibody generation.** PCR was used to amplify cDNA encoding the C-terminal 248 amino acids of *mus musculus* Smo (aa 545−793) followed by a 6xHis tag. The PCR product was cloned into the BamH1 and EcoR1 sites of the pGEX-4T1 vector (GE Healthcare) by Gibson assembly (New England Biolabs). Expression of the GST-Smo C-terminal-6xHis tag fusion protein (GST-CSmoHis) was carried out in the host *E. coli* strain SHuffle® T7 *E.coli*/K12 overnight at 16 °C. The cell culture (4 L) was harvested by centrifugation at 4000 × g for 10 min and the pellet was resuspended in 50 mM phosphate buffer pH 7.5 containing 150 mM NaCl (PBS) and lysed in an Avestin EmulsiFlex C5 homogenizer. The lysate was clarified by centrifugation at 48,000 × g for 30 min and the supernatant applied to a column containing 2.5 ml packed volume of glutathione-Sepharose (GE Healthcare). The column was washed extensively with PBS and GST-CSmoHis was eluted with 20 mM glutathione. The eluate was then treated with 20 U of Thrombin (Sigma-Aldrich) for 5 h at room temperature and loaded onto a column containing 0.5 ml packed volume of Ni NTA affinity resin. After extensive column washing, the CSmoHis protein was eluted with PBS containing 350 mM imidazole, desalted and concentrated to a concentration of 1 mg/ml protein.

The fusion was made from splenocytes of NMRI mice or SPRD rats (Taconic), which were SC immunized twice at a 14-day interval with 30 µg of the murine Smo C-terminal part aa544-793 containing a 6xHIS tag antigen glutaraldehyde coupled to diphtheria toxoid. The antigen was administered with the GERBU P™ adjuvant according to the manufacturer's recommendations. Four days prior to the fusion, the animals received an IV injection boost of 15 µg antigen administered with adrenaline.

Fusions and screenings were performed with the mouse SP2 myeloma cell line as the fusion partner[50].

**Immunofluorescence.** WT and Smo KO MEFs were grown on coverslips to confluency. Primary murine cells were diluted to 1 × 10^6 cells/ml, plated onto glass slides and allowed to adhere to the glass at 37 °C for 10 min. Cells were fixed with 4% PFA (16% PFA solution, CN Technical Services, cat no. 15710-s), 1× PBS (from 10× PBS, in-house), and Milli-Q H₂O) for 10 min at room temperature. Slides were washed 5 times with PBS (in-house) and blocked with blocking buffer: PBS + 1% bovine serum albumin (BSA, Sigma, cat. no. A3912; 50 g lyophilized powder) + 0.1% TritonX-100 (Alfa Aesar) for 30 min at room temperature. Blocking solution was aspirated and mouse (Clone 18-2-3) Smo hybridoma supernatant + 0.1% TritonX-100 was added. Slides were incubated for 1.5 h at room temperature and then washed 5 times with PBS + 0.1% TritonX-100. Anti-mouse AlexaFlour 488 secondary (Thermo, cat no. A21202, 1:500 dilution) was added in blocking buffer (PBS + 1% BSA + 0.1% TritonX-100) and slides were incubated for 30 min, at room temperature and protected from light. After incubation, slides were washed 5 times with PBS + 0.1% TritonX-100 and stained with Hoechst (Hoechst 33342, trihydrochloride, trihydrate Invitrogen/Fisher, cat no. H3570, 1:30,000 dilution) prepared in PBS for 5 min, light protected at room temperature. Slides were washed 5 times with PBS and mounted with ProLong Diamond Antifade Mountant (Fisher, cat no. P36961). Excess mounting fluid was wiped off and slides were allowed to set at room temperature and light protected overnight before imaging.

**Image acquisition and analysis.** Confocal spinning disc microscopy was performed on an Andor Dragonfly 500 (Oxford Instruments). Images were processed using Imaris software (Bitplane/Oxford Instruments).

**Retroviral transduction.** A pMig vector was cloned containing a murine Ihh construct, flagged with an N-terminal HA tag, seven codons downstream of the N-terminal lipidation site in order to not disrupt the signal sequence or any N-terminal palmitoylation. HEK 293T cells were seeded in a six well plate resulting in 75% confluency the next day. Media was replaced with 2 ml fresh DMEM per well prior to transfection with retroviral plasmids. HEK 293T cells were transfected with 1.5 µg of packaging plasmid pCL-Eco (generous gift from Gillian Griffiths, Cambridge UK) and 1.9 µg pMig vector prepared in Opti-MEM™ media (Gibco, cat no. 31985070). Concurrently, a 1:25 dilution of Lipofectamine™ 2000 (Invitrogen, cat no. 11668019) was prepared in Opti-MEM™ media. The DNA-Opti-MEM™ solution was mixed with the Lipofectamine™2000-Opti-MEM™ solution at a ratio of 1:1 and incubated for 5 min at room temperature after which 400 µl of the mixture was added drop-wise to the HEK-293T cells. Media was replaced 18 h after transfection and collected 48 h post transfection. The retroviral supernatant was passed through a 0.45 µm PVDF membrane filter (Merck Millipore, cat no. SLHVM33RS) for immediate use.

Naïve CD4⁺ T cells were isolated from C57BL/6 spleens and stimulated with plate bound anti-CD3ε/CD28 antibodies. After 24 h of stimulation retroviral supernatant was added to the culture in a 1:1 ratio and the cultures supplemented with protamine sulfate (Sigma, cat no. 1101230005) at a final concentration of 6 µg/ml. Cells were then centrifuged at 680 × g for 10 min at 32 °C and subsequently placed in a humidified cell culture incubator. 48 h after transduction cells and supernatants were harvested for western blot.

**Western blot.** Cells were harvested at 4 °C, washed twice in ice-cold PBS and lysed in ice-cold RIPA buffer (150 mM NaCl, 50 mM Tris pH 7.4, 1 mM MgCl₂, 2% NP40, 0.25% Na deoxycholate, 1 mM DTT) with protease inhibitor (Pierce, cat. no. 88666) at a concentration of 40 × 10^6 cells/ml for 15 min on ice with intermittent vortexing. PhosStop phosphatase inhibitor (Sigma, cat no. 4906845001) was added in the lysis buffer for samples where phosphoepitopes were to be detected. Lysates were centrifuged at 1600 × g for 10 min at 4 °C and the supernatant was transferred to a fresh tube and stored at −80 °C. Samples were boiled (100 °C) for 3 min or incubated at 37 °C for 15 min where Smo was to be probed, and loaded, together with a protein standard (Bio-Rad, cat no. 161-0394), onto a NuPAGE 4–12% gradient Bis/Tris Acrylamide gel (Thermo Fisher, cat no. NP0335BOX). PAGE was run in Nu-PAGE MOPS running buffer (Thermo Fisher, cat no. NP0001). Western blotting was performed using wet transfer in Nu-PAGE Transfer Buffer (Thermo Fisher, cat no. NP0006-01) + 10% Methanol (Honeywell, cat no. 32213-2.5 L) for 90 min at room temperature at 300 mA constant onto a 0.45 µm nitrocellulose membrane (Thermo Fischer, cat no LC2001). Membranes were blocked with 5% (w/v) nonfat dry milk (Marvel Original, Dried Skimmed Milk) in TBST or in 5% BSA in TBST for 1 h at room temperature prior to overnight incubation with primary antibody at 4 °C. The membrane was developed with SuperSignal West Pico Plus Chemiluminescent Substrate (Thermo Fisher, cat no. 34580) or Super-Signal West Dura Extended Duration Substrate (Thermo Fisher, cat no. 34075) or imaged with LICOR Odyssey CLx. A list of primary and secondary antibodies used for Western Blot is shown in Supplementary Table 4.

**RNA sequencing.** Samples were generated and RNA was extracted as detailed in the "methods" section. Six biological replicates were used per condition. RNA quality was assessed using a capillary electrophoresis system (4200 Tapestation, Agilent) using RNA ScreenTape (Agilent, cat no. 5067-5576) as per the manufacturer's instructions. RNA concentrations were quantified using the Qubit™ RNA BR Assay kit (Thermo, cat no. 10210) as instructed by the manufacturer. Libraries were generated using the TruSeq stranded mRNA kit (Illumina) following the manufacturer's instructions and sequenced using single-read sequencing with the HiSeq4000 platform (Illumina).

Reads were aligned to the mouse genome version GRCm38 using STAR v2.5.3a[51]. Read counts were obtained using feature Counts function in Subread v1.5.267[52] and read counts were normalized and tested for differential gene expression using the DESeq2 workflow[53]. Multiple testing correction was applied using the Benjamini–Hochberg method.

GSEA (http://www.broad.mit.edu/gsea) was carried out using the gene sets from Molecular Signatures Database (MSigDB)[54]. Genes were ranked based on the log2 Fold change multiplied by −log10(padj).

**CD3 injection model of small intestinal inflammation.** 8–10-week-old female C57BL/6 mice were injected with 20 µg anti-CD3 monoclonal antibody (Clone: 145-2C11) at 0 and 48 h[17]. Mice were dosed every 12 h by oral gavage with 100 mg/kg vismodegib (LC laboratories), prepared in MCT from a fresh vial for each experiment (0.5% methylcellulose, 0.2% Tween 80), or carrier control, with four to five mice per group. Mice were harvested at 52 h, at which point serum was collected and small intestines were harvested in ice-cold PBS. Small intestines were cleaned using ice-cold PBS, cut into roughly 5 mm pieces and collected in complete IMDM. Tissue pieces were rotated for 30 min at room temperature to release intraepithelial lymphocytes (IELs). Cells were strained on a 70 µm filter and IELs were collected from the interface of a 40–80% Percoll gradient (GE Healthcare, cat no. 17089101). IELs were restimulated for 4 h with PMA and Ionomycin in the presence of Monensin and subjected to viability, surface, and intracellular flow cytometric staining (for details see section on flow cytometric methods). Serum IL-17a concentrations were determined using the ELISA MAX™ Deluxe Set Mouse IL-17A (Biolegend, cat no. 432504) according to the manufacturer's instructions.

**Adoptive T cell transfer colitis model.** CD4⁺ T cells were isolated from spleen and peripheral lymph nodes of donor mice using MACS isolation. The CD4⁺ T cells were stained in MACS Buffer for 15 min at 4 °C for CD4, CD25, CD45RB, and DAPI was added prior to FACS sorting which was performed. Pure CD4⁺ CD25⁻ CD45RB^hi tdTomato⁺ cells were sorted to high purity. Cells were washed twice in PBS prior to intraperitoneal injection into Rag2⁻/⁻ recipient mice. A cell suspension of 200 µl at 2 × 10^6 cells/ml PBS were injected intraperitoneally per mouse. Mice were weighed at least twice weekly thereafter and harvested between 5 and 6 weeks after injection.

Colons, mesenteric lymph nodes, and spleens were harvested in sterile ice-cold PBS. Colons were flushed through with ice-cold PBS to remove faeces and a 5–10 mm representative sample was collected in 10% Neutral Buffered Formalin (NBF). These samples were incubated overnight in NBF prior to being moved to a 70% Ethanol solution for H&E staining (in-house histopathology core). After collection of histology samples, the colon was cut into roughly 5 mm pieces collected in a 50 ml falcon with complete IMDM + 58 µg/ml DNAse (Stem Cell Technologies, cat no. 7469) + 58 µg/ml Liberase, Thermolysin low (Roche, cat no. 5401020001) and incubated in a shaker at 37 °C for 25 min at 225 rotations per minute to release colonic lLPLs. Cells were strained twice through a 70 µm filter and LPLs were collected from the interface of a 40–80% Percoll gradient (GE

Healthcare, cat no. 17089101). Spleens were mashed through a 70 μm filter with sterile PBS and collected from the interface of a Lympholyte gradient (Cedarlane Laboratories, cat no. CL5035). Gradient centrifugations were set up in 15 ml falcons and centrifuged for $680 \times g$ for 20 min (acceleration 1/9, deceleration 0/9). Mesenteric lymph nodes were strained twice on a 40 μm filter.

LPLs, splenocytes, and lymph node suspensions were restimulated for 4 h with PMA and Ionomycin in the presence of Monensin and subjected to viability, surface, and intracellular flow cytometric staining (for details see section on flow cytometric methods). Statistical analysis was performed using Prism 7 software (GraphPad Inc.). Statistical analysis was performed using an unpaired two-tailed Student's $t$ test or Mann–Whitney U test, respectively. For each data set, the presence of a statistically significant outlier was assessed using Grubbs' test ($\alpha = 0.05$) and where present was excluded from subsequent analysis.

Semi-quantitative analysis of colitis severity was determined on formalin-fixed, paraffin-embedded, and hematoxylin & eosin stained sections as established in ref. [55]. Sections were analyzed in a blinded fashion by an independent expert. The sum of each subscore of mononuclear infiltration (0–3), crypt hyperplasia (0–3), epithelial injury (0–3), neutrophil infiltration (0–3), and inflammatory penetration (0–2) depicted colitis severity.

**Human data sets**. The human datasets are from publicly available data were obtained from GEO with the following accession numbers: GSE49877, GSE109142, and GSE117993. The cited references provide details of participant characteristics and relevant ethics/consent guidelines applied[31–33]. Number and sex of participants are indicated in the Source Data.

**Statistics & reproducibility**. Statistical analysis was performed using Prism 7 software (GraphPad Inc.). Details of the respective statistical tests used are noted in the figure legends. No statistical method was used to predetermine sample size. Sample size was based on typical standards in the field. No animals were excluded from analysis in any experiment. For the analysis of the adoptive transfer colitis model only, data points from all three independent experiments were pooled and the presence of a single statistically significant outlier was assessed using Grubbs' test ($\alpha = 0.05$), and if present that single data point was excluded from further analysis. Every experiment has been independently repeated at least three times within the main manuscript as detailed in the figure legends. Within the supplemental information three independent repeats were not always performed due to breeding problems in the genetically-modified mouse lines (Supplementary Figs. 4b, 6, 7d). Regarding randomization for genetic mouse experiments: Different genotypes of the same sex and from the same litter were compared where possible. Wild-type and Rag2KO recipient mice were randomly divided into control or treatment groups. Other experiments are not relevant to the application of randomization. Pathology scores in the adoptive transfer colitis model were determined by a gastroenterologist (TA) who was blinded to experimental groups. Further details can be found in the Reporting Summary.

**Reporting summary**. Further information on research design is available in the Nature Research Reporting Summary linked to this article.

## Data availability

The RNA-Seq data generated in this study have been deposited in NCBI's Gene Expression Omnibus under GEO Series accession number GSE205848. Reads were aligned to the mouse genome version GRCm38 using STAR v2.5.3a. Publicly available data were obtained from GEO with the following accession numbers: GSE49877, GSE109142, and GSE117993. Source data are provided with this paper.

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

## Acknowledgements

Special thanks and gratitude goes to Jing Su from the Bioinformatics core at the CRUK Cambridge Institute for her support in analyzing RNA-Seq data; Ellie Pryor, Nicky Jacobs, and Gemma Cronshaw from the BRU for expert animal care; the Histology core for processing and staining of colon samples; the flow cytometry core for assistance with cell sorting and the Genomics core for sequencing. We thank Gitta Stockinger and Gillian Griffiths for advice and support on the manuscript. This work was supported by Cancer Research UK (MdlR (A22257), F.B., L.M.O.B., C.K., H.-C.C., V.C.); Sir Henry Dale Fellowship jointly funded by the Wellcome Trust and the Royal Society (MdlR (WT107609); LMOB); Gates Cambridge Trust (A.K.); J.H. is undertaking a PhD funded by the Cambridge School of Clinical Medicine, Frank Edward Elmore Fund and the Medical Research Council's Doctoral Training Partnership (award reference: 1954837). T.A. is grateful for the support from the Austrian Science Fund (FWF P33070) and the European Research Council (ERC – STG: 101039320).

## Author contributions

M.d.l.R. and J.H. conceived the project and designed the experiments. J.H. performed most experiments and was helped by F.B. L.M.O.B., C.K., and H.-C.C. helped with experiments and technical expertise. K.K. analyzed RNA-Seq datasets. Monoclonal antibodies against murine Smo were generated by K.S., M.S., Marc d.l.R., and M.d.l.R. and validated by L.M.O.B., V.C., and A.K. T.A. performed the histological severity scoring of the adoptive transfer colitis tissue samples. J.H. and M.d.l.R. analyzed the results and wrote the manuscript.

## Competing interests

The authors declare no competing interests.
