## [Peer Review File · Nature Communications]

Cell-autonomous Hedgehog signaling controls Th17 polarization and pathogenicityReviewers' comments:

Reviewer #1 (Remarks to the Author):

The article of Hanna et al. describes a novel role of the canonical Hh pathway in Th17 differentiation. This pathway is clearly dependent on *Ihh*, *Smo* and *Gli3* and, unexpectedly, appears to take place intracellularly. The study confirms the relevancy of the three Hh pathway components with the use of tissue-specific genetic models, and a combination of pharmacological inhibitors of *Smo* in vivo, in a novel Th17 polarisation protocol in vitro and two mouse models of intestinal inflammation. The data is mostly solid and supportive of the conclusions; however, I have a few issues (below) that would like to see addressed experimentally:

1- The data indicate that *Ihh* has important contributions for Th17 polarisation in a cell-autonomous manner (*CD4ERT2Cre+ Ihhfl/fl*). However, that does not show that signalling takes place intracellularly vs an autocrine effect. The authors conclude that signalling is intracellular due to the inability of recombinant *Ihh* to stimulate Th17 polarisation on wild type cells that express *Ihh*, which could be already maximally active in an autocrine manner. A possible way of demonstrating the inability of exogenous *Shh* to mediate Hh signalling in Th17 differentiation would be to repeat the in vitro test of lineage specification using *Ihhfl/fl* *CD4+* naive cells exposed to recombinant *Ihh*-N peptide. In addition, the effect of a blocking *Shh/Ihh* antibody (such as the 5E1 monoclonal Ab) in the in vitro polarisation assay of wild type cells will provide clear evidence of the role of any autocrinally secreted *Ihh* (since *Shh* is not expressed). It would also be interesting to look at processing of *Ihh* in these cells compared to *Ihh* transfected in for example HEK293 cells by WB, to determine if processing of the 45kDa precursor into the active 19 kDa N-peptide takes place in *CD4+* naive cells.

2- The lack of *Gli2* expression in all the *CD4+* lineages is surprising, since it had previously thought to be the main constitutively expressed Gli transcription factor to mediate *Smo*'s activation. To confirm the findings, the authors should include a positive control of their amplification strategy (any other mouse cell type that expresses *Gli2*).

3- While the use of *Smo* KO cells and cyclopamine clearly demonstrate the role of *Smo* in Th17 differentiation, the concentrations of vismodegib required to observe an inhibitory effect are too high and could be the result of unspecific effects. Vismodegib is effective in cell culture models in nM concentrations; however, its mechanism of action is different of cyclopamine: cyclopamine induces accumulation of inactive *Smo* in the primary cilium (absent in these cells) while Vismodegib in most cell types prevents *Smo* accumulation in cilia. Could the authors take advantage of the high quality *Smo* mAb to investigate if there are any change in *Smo* localisation during Th17 polarisation and if that is altered by both inhibitors?

4- The finding of reduced P-AMPK in naive *CD4+* T cells after incubation with cyclopamine suggests that *Smo* activity is necessary to increase AMPK activity. However, it does not indicate in its own that AMPK mediates the effect of *Smo* in Th17 differentiation. To conclude that non-canonical Hh signalling through AMPK partly mediates the polarisation of Th17 cells, first the authors need to demonstrate that AMPK is necessary in that process, for example with a specific AMPK inhibitor.

5- I would also advise to re-analyse the *GLI3*-target genes modulated by 5 microM cyclopamine (*Smo*-specific) without including the 7.5 microM cyclopamine results

(concentration at which non-specific effects start to occur). The possible off target effects of 10 microM cyclo are particularly evident from those clusters that show a change in the opposite direction between the 2 drug concentrations, like the top cluster and middle clusters in Fig 7D.

6- The text "Smo activates Gli transcription factors as part of the canonical pathway, but has also been shown to activate AMPK 23 or act through its GTPase activity when associated with non-canonical Hh signaling" and the diagram of Fig 7A are somewhat incorrect. Smo does not have intrinsic GTPase activity, it acts as a GEF for heterotrimeric Gi proteins and, downstream, small GTPases like Rac1 and RhoA. Thus, the sentence should be rephrased as "or act as a GPCR through activation of Gi proteins" and the diagram updated accordingly. Indeed, the experiment with PTX inhibits Smo/Gi coupling.

7- Can the authors analyse changes in expression of a subset of the top Gli3-regulated genes identified by RNA-seq in the CRISPR Gli3 cells?

Reviewer #2 (Remarks to the Author):

The study by Hanna et al. provides new evidence for the involvement of Hedgehog signaling in the polarization of naïve Th0 to Th17 cells. The authors show that during Th17 polarization, there is activation of the expression of Hedgehog effectors such as Smo, Gli1, and Gli3. Genetic and pharmacological inhibition of Smo function in vitro and in T-cell specific conditional mouse models reduces the in vitro polarization of Th17 cells. Of note, genetic inhibition of Ihh or pharmacological perturbation of Smo function in vivo reduces the inflammatory phenotype in two IBD mouse models with a demonstrated pathological role of Th17 cells.

This study extends our current understanding of the function of the Hedgehog signaling pathway in the immune system and in pathological processes such as inflammatory bowel disease. However, the data presented also allow other conclusions and therefore the following main points would need to be addressed by the authors:

1) It is unclear why the authors speak of intracellular Hedgehog signaling. Ihh has a signal peptide and is secreted by producing cells. Thus, autocrine Hedgehog signaling and a cell-autonomous role of Ihh/Gli signaling is likely. It would therefore be important to investigate in vitro polarization in the presence of Ihh neutralizing antibodies (mAb 5E10) (e.g. supplemental Figure 2).

2) The use of cyclopamine is very problematic due to its many non-specific effects. FDA-approved vismodegib is unfortunately only used once in vitro in Figure 3C. Here, at 5µM, there is minimal, if any, effect on Th17 polarization. This is problematic as the IC50 values of vismodegib are in the range of 10-20nM. Vismodegib is effectively used in vitro at concentrations of 100-500nM and at these levels vismodegib de facto completely inhibits Smo signaling. Thus, the role of Ihh and Smo in the in vitro polarization of Th17 cells remains unclear.

3) Why did the authors use different Cre-deleter strains for Ihh and Smo knockouts?

4) The autocrine role of Ihh-Smo-Gli3 signaling in vivo remains unclear. In Figure 4, it would be necessary to show the total number of T cells as well as the number of Th2 and iTregs. Similarly, it would be important to quantify the number of tdTOM-positive cells in the HET and KO transplanted mice in the whole organism as well as in the affected tissue. An alternative explanation for the decreased pathologic score could be that Ihh-deficient T cells generally show a reduced infiltration behavior in the Rag-/- recipient mice than HET cells. It

would also be important to compare Smo-Wt and Smo-deficient cells, as the effect of Ihh could also be via paracrine signaling.

5) Figure 4 C and H: why does vismodegib not affect the number of Il17a-/Ifng+ cells, while loss of Ihh function does?

6) Figure 4H: do the cell numbers change for iTregs and Th2?

7) The role of Gli3 in Th17 polarization is not fully understood. Gli3 activity is primarily regulated by proteolytic processing. It is therefore essential to investigate this at least by Western blotting (Gli3R vs Gli3 full-length) in the course of T cell polarization. In Fig. 2A, Gli3 mRNA levels increase sharply and Gli1 levels decrease. This could indicate Gli3 repressor function. Similarly, it would support the authors' conclusions if Gli3-KO cells were examined in the adoptive transfer model.

8) Figure 4: why do CD4CreER;Ihhfl/fl mice differ from dLckCre;Smo^{-/-} mice in terms of the number of Ifngamma pos. Th1 cells (increase by Ihh KO, no change by Smo-KO)? Similarly, for iTreg cells. This suggests a non-cell autonomous role of Ihh in Th cell polarization and somewhat contradicts the conclusions of the authors.

9) Smo Western blots of T cells show two bands in contrast to MEFs. Why, and which one is specific? It would be important to study Smo protein expression in T cells from wild-type and Smo-deficient mice, preferably also by confocal microscopy.

10) Please add a positive control for Gli2 to support the absence of Gli2 mRNA expression.

11) The relevance to human pathology remains undefined. At least Ihh-Smo-Gli3 expression should be quantitatively investigated in T cells (at least in Th17) from IBD patient tissue.

12) The apparent anti-inflammatory effect of vismodegib is in contradiction to previous publications in which, for example, a clear pro-inflammatory effect has been shown in colitis models (e.g. Lee, et al. Control of Inflammation by Stromal Hedgehog Pathway Activation Restrains Colitis. Proceedings of the National Academy of Sciences of the United States of America 2016, 113 (47), E7545-E7553. <https://doi.org/10.1073/pnas.1616447113>). The current study would benefit from experiments using the Smo Agonist (SAG) for in vivo treatment. Thus, it would be important to investigate whether SAG treatment affects the phenotype of Rag^{-/-} mice transplanted with Ihh-deficient T cells.

Minor points:

1) Western blot (not western)

2) Tfh cells should also be mentioned in the introduction

3) Line 99 is not totally correct; a suggestion could be: ... where it activates the Gli transcription factors Gli2 and Gli3. These translocate to the nucleus and initiate the transcription of Hh target genes including Gli1.

4) Figure 2A: The amount of tubulin loading control varies relatively strongly. Densitometric quantification of Smo/tubulin amounts would be helpful.

5) Figure 2A: Why is there such a significant difference in Smo protein levels between TGFB-block and TGF-beta untreated samples?

Reviewer #3 (Remarks to the Author):

The manuscript by Hanna et al., investigates the role of the Hh pathway in Th cells differentiation. The authors claim that Hh signalling cascade, via Smo and Gli3 is critical for the generation of Th17 cells.

Major concerns:

1) This reviewer would like to see a more systematic presentation of the results: especially

showing the repeats as individual symbols in bar graphs and showing all subsets in critical experiments. E.g. Hh inhibitors were only tested on Th17 and Treg. Often Treg are robust, this needs to be tested on Th1 cells in addition as a minimum (concentration range to 10uM, not just one). The effect of the absence of Gli1 is assessed for all subsets, but absence of Gli3 only for Th17 cells, while this is critical.

2) The data do not support the conclusions (or the title). The authors make claims regarding Th17 cells, but till figure 7, only analyse IL17A production. This leaves the reader unsure if the regulation is at the level of the IL17A cytokine or if indeed Th17 differentiation is affected. In final figure 7, the authors finally show that Th17 generation is not affected, with the typical transcriptional programme (RORgt, RORa, etc) intact. Yet, the authors keep referring to and draw conclusions regarding Th17 cells in general.

At end of page 10 (and 14), the authors conclude "Taken together, we found that Hh signalling is crucial for Th17 polarization but not Th1, Th2, or iTreg differentiation." This is a very strong statement. Especially when they show in Figure 4, that the genetic absence of Ihh reduced IL17, but certainly does not ablate it, nor is there info on Th17 cells. Especially in light of the very good deletion shown in Suppl Fig 6.

It seems Hh signalling may contribute to IL17A production, but is not crucial. Later experiments indicate Th17 cells are fine and the observations at best are limited to a subset of the Th17 transcriptional programme.

3) Absence of molecular mechanism.

Expression of Gli1 is found in Th1, 17 and Treg, expression of Gli3 in Th17 and Treg, Smo mRNA expression in all, but Smo protein mainly in Th17. Why is this? Is there cross-reactivity (validation of antibody only on KO and overexpression)? Smo is assessed by WB only, while other factors are analysed by qPCR. The running pattern in Figure 2A shows 2 bands for Smo of distinctly different weights, Figure 2B, two bands are close in molecular weight. Condition in Fig2A, IL-6, -23 and 1b does not show Smo protein, which undermines the authors' conclusion that IL-6 is required for Smo induction. How does IL-6 induce Smo protein induction, is this STAT3 mediated? All this undermines the confidence in the data. The authors establish in Figure 7 that there is no effect on Th17 cell differentiation since the main transcriptional programme is present. i.e. the cells still express the required transcriptional programme, but one (or more) aspects of the programme, namely the translation/transcription of IL17A is affected. Yet, the authors keep referring to Th17 differentiation thereafter.

Gli1 is excluded (checked on all Th subsets), and Gli3 may have a supportive role. But figure 7E does not show a large difference that explains previously reported marked effects on IL17A. Yet, the authors conclude "we have uncovered Gli3 as a fundamental new TF implicated in Th17 polarization" I do not find the data supporting this conclusion sufficiently. Again, critical controls (other subsets) are missing here, e.g. in the pAMPK analysis.

4) Physiological relevance:

The colitis models seem to show a reduced infiltration/recruitment of T cells, also Th1 cells are diminished in the second model, and not shown in the first model (as under point 1, a more systematic analysis is required).

Although similar T cell numbers are present in the periphery, the expression of Smo etc during thymic selection may alter TCR selection. Transfer colitis depends on high-affinity TCRs responding to the environment in lymphopenic mice, giving some doubts to the use of this model.

The models used do not distinguish between a more general role in T cell biology vs a

specific role in Th17 cell differentiation and function or need much more robust analysis.

Other concerns:

The authors should include a positive control for Hh components (Suppl Fig2, Fig 1, etc) for the context of expression levels. It is not clear what the qPCR expression data means without context.

This reviewer is concerned that the authors claim Th17 cells can be generated with IL-6 alone. The authors have (Suppl Fig4) a substantial background of IL-17 staining (~10%). Possibly due to use of MACS separated cells only.

Line 67: Helminth infections, not parasites (i.e. Plasmodium or Toxo are parasites inducing a Th1 response)

Line 183: 1 uM = 10 nM?

Figure 2: IL1b symbols should be "+"?

The authors mention several time they "prove" something. I like to see that changed to data supporting or substantiating a hypothesis.

Reviewers' comments:

Reviewer #1 (Remarks to the Author):

We thank the reviewer for the excellent and constructive comments which we have all addressed with new data.

The article of Hanna et al. describes a novel role of the canonical Hh pathway in Th17 differentiation. This pathway is clearly dependent on *Ihh*, *Smo* and *Gli3* and, unexpectedly, appears to take place intracellularly. The study confirms the relevancy of the three Hh pathway components with the use of tissue-specific genetic models, and a combination of pharmacological inhibitors of *Smo* in vivo, in a novel Th17 polarisation protocol in vitro and two mouse models of intestinal inflammation. The data is mostly solid and supportive of the conclusions; however, I have a few issues (below) that would like to see addressed experimentally:

1- The data indicate that *Ihh* has important contributions for Th17 polarisation in a cell-autonomous manner (*CD4ERT2Cre+ Ihhf1/fl*). However, that does not that signalling takes place intracellularly vs an autocrine effect. The authors conclude that signalling is intracellular due to the inability of recombinant *Ihh* to stimulate Th17 polarisation on wild type cells that express *Ihh*, which could be already maximally active in an autocrine manner. A possible way of demonstrating the inability of exogenous *Shh* to mediate Hh signalling in Th17 differentiation would be to repeat the in vitro test of lineage specification using *Ihhfl/fl CD4+* naive cells exposed to recombinant *Ihh-N* peptide. In addition, the effect of a blocking *Shh/Ihh* antibody (such as the 5E1 monoclonal Ab) in the in vitro polarisation assay of wild type cells will provide clear evidence of the role of any autocrinally secreted *Ihh* (since *Shh* is not expressed).

These are excellent suggestions and we have performed both experiments.

We have polarised naïve $CD4^+$ T cells from excised *CD4CreER^{T2} Ihh^{ff}* (KO) mice as well as *CD4CreER^{T2} Ihh^{+f}* (HET) and *CD4CreER^{T2} Ihh^{+/+}* (WT) controls with increasing doses of recombinant *Ihh-N* peptide and found that exogenous active *Ihh-N* peptide cannot rescue the defect observed in the *Ihh* KO cells (**new Fig. 5 C**).

Figure 5: Conditional knockout of *Ihh* in CD4⁺ T cells leads to diminished Th17 polarization but does not affect other Th lineages. (C) Th17 cells were polarized from HET, KO mice in the presence/absence of recombinant N-terminal murine *Ihh* fragment at the indicated concentrations. Cells were harvested for analysis by flow cytometry on day 5. Representative flow cytometry plots of are shown on the left with cells treated with 200ng/ml rlhh with a summary on the right. n = 3-4 mice. Data are means +/- SD. p-values were calculated using an unpaired two-tailed Student's t test. * p<0.05, ** p<0.01, *** p<0.001.

In addition, we have cultured naïve CD4⁺ T cells under different strength of Th17-polarizing conditions, in the presence of Hh blocking antibody 5E1 or isotype control, and have shown that blocking of extracellular Hh ligands does not affect Th17 polarisation (**new Fig. 3 B**).

Figure 3: Exogenous Hedgehog ligands do not affect Th17 polarization. (B) Naïve CD4⁺ T cells were stimulated with the indicated polarizing cytokines in the presence or absence of Hh ligand blocking antibody 5E1 or isotype control antibody at 10μg/ml. n = 4. Data are means +/- SD. p-values were calculated using an unpaired two-tailed Student's t test. * p<0.05, ** p<0.01, *** p<0.001.

Taken together, we firmly establish that (1) cell-endogenous Ihh is sufficient and required for optimal Th17 polarisation (*Ihh* KO rescue experiments) and that (2) neither exogenous Hh ligand, potentially present in the serum, or endogenous Ihh secreted from the CD4⁺ T cells themselves can promote Th17 polarisation (recombinant Ihh administration experiments + 5E1 experiments).

It would also be interesting to look at processing of Ihh in these cells compared to Ihh transfected in for example HEK293 cells by WB, to determine if processing of the 45kDa precursor into the active 19 kDa N-peptide takes place in CD4⁺ naive cells.

We agree with the reviewer that Hh processing in T cells is a very interesting question to investigate and we are in fact actively exploring Ihh processing in CD8⁺ T cells as part of another manuscript. Commercial antibodies faithfully detecting endogenous Ihh in primary murine T cells are not available at the moment. We thus designed a retroviral Ihh-construct for transduction which contains an N-terminal HA tag located seven codons downstream of the N-terminal lipidation site (in order to not interrupt the signal sequence nor palmitoylation). In addition, we have developed a protocol to retrovirally transduce primary Th17 cells (see new Materials and Methods) and assessed processing of overexpressed Ihh in Th17 cells (**new Suppl. Fig. 2C**). Th17 cells process full-length Ihh precursor (45kDa) successfully into N-peptide (19kDa) and secrete N-peptide in the culture supernatant. (Based on published literature, we speculate that the upper band between 20 and 25kDa could be the palmitoylated N-Peptide, that is preferentially secreted.)

Supplemental Figure 2: Expression of Hh signaling components during CD4⁺ T helper cell polarization. (C) Th17 cells were transduced with pMIG retrovirus encoding HA-(N-terminal)-tagged Ihh or empty vector (EV) control on day 1 post stimulation. Cells and culture supernatants were harvested separately on day 3 for immunoblot analysis of HA and Actin. Representative blot of 2 independent experiments with 2 separate mice each is shown.

Whether T cells process endogenous, physiological levels of Ihh precursor, or the processing observed is an artefact of the overexpression remains to be seen but requires us to make better antibodies against Ihh first.

Since both, Ihh-N and the full length Ihh are biologically active ¹, the processing of Ihh ligand in Th17 cells might have little impact on Th17 polarisation especially since we have shown that Ihh does not function in an autocrine manner. However, secretion of Ihh by Th17 cells might impact exogenous Ihh gradients and thereby affecting Hh signalling in other cell types.

2- The lack of Gli2 expression in all the CD4⁺ lineages is surprising, since it had previously thought to be the main constitutively expressed Gli transcription factor to mediate Smo's activation. To confirm the findings, the authors should include a positive control of their amplification strategy (any other mouse cell type that expresses Gli2).

Mouse Embryonic Fibroblasts (MEFs) and murine thymus have been reported to be positive for Gli2. We have thus included qRT-PCR analysis of MEFs and thymic tissue as additional controls and shown that our Gli2 probe faithfully reports Gli2 expression (**new Suppl. Figure 2A, B**).

Supplemental Figure 2: Expression of Hh signaling components during CD4⁺T helper cell polarization. (A) Naïve CD4⁺T cells were purified from spleen and peripheral lymph nodes of C57BL/6 mice and stimulated with plate-bound anti-CD3 ϵ /CD28 antibodies in the presence of polarizing cytokines to generate Th0, Th1, Th2, Th17 and Treg subsets. Expression of *Ptch1*, *Ptch2*, *Smo*, *Shh* and *Dhh* were assessed by qRT-PCR in Th subsets at the indicated timepoints after TCR stimulation in the presence of polarizing cytokines. Data is normalized to *Tbp* as a reference gene. Similar results were obtained when *CD3 ϵ* was used as a reference gene. n = 3 independent experiments. *Gli2/Dhh* mRNA was undetectable across all conditions tested. Data are means +/- SD. nd = not detected. As a positive control, expression levels of Hh components in Mouse Embryonic Fibroblasts (MEFs) were assessed (right column). n = 3. Data is normalized to *Tbp* as a reference gene. n = 3 independent experiments. (B) Left: Mouse Embryonic Fibroblasts (MEFs) were assessed for *Gli1* and *Gli3* expression by RT-qPCR. n = 3. Right: Testes and thymi were dissected from C57BL/6 mice (n=4), homogenized and lysed for qRT-PCR analysis of *Dhh* and *Gli2* mRNA expression, respectively.

3- While the use of Smo KO cells and cyclopamine clearly demonstrate the role of Smo in Th17 differentiation, the concentrations of vismodegib required to observe an inhibitory effect are too high and could be the result of unspecific effects. Vismodegib is effective in cell culture models in nM concentrations; however, its mechanism of action is different of cyclopamine: cyclopamine induces accumulation of inactive Smo in the primary cilium (absent in these cells) while Vismodegib in most cell types prevents Smo accumulation in cilia.

Regarding the concentrations of Vismodegib used, we agree with the reviewer. Although we used higher concentrations of Vismodegib than those reported in the literature for cell lines, the concentrations we used are in line with those used in primary murine CD8⁺ T cells without off-target effects² and other primary cells (murine rhabdomyosarcoma cells and human BCC explants)^{3,4}.

We have carefully monitored for potential off-target effects by assessing viability and cell numbers of the delicate primary CD4⁺ Th cell lineages when treating with Hh inhibitors (shown in **Fig.4 B**, right panels and **Suppl. Fig.5 A**, right panels – see below).

Most importantly, however, we have generated and validated unique conditional and inducible Smo and Ihh KO models for CD4⁺ T cells confirming the results of the inhibitor experiments, thus showing that the effect with both Hh inhibitors is reproducible across two independent novel mouse models, which do not suffer from potential off-target effects as small molecule inhibitors do.

Note: We find that commercially available Vismodegib is very unstable. This also might set our concentrations apart from cell line studies by Genentech done “in house”.

Could the authors take advantage of the high quality Smo mAb to investigate if there are any change in Smo localisation during Th17 polarisation and if that is altered by both inhibitors?

This is an excellent question. The immune synapse (the surrogate of a primary cilium in CD8⁺ T cells) is virtually unstudied in Th17 cells. In preliminary experiments we found that if we perform synapse formation on anti-TCR coated coverslips using polarised Th17 cells, actin clears and in very few cells we see Smo at the synapse (see below). In-depth characterisation of Th17 synapses with regards to timing of actin dynamics, centrosome polarisation/docking and vesicle secretion is first needed to obtain meaningful data on Smo localisation with regards to the synapse as a surrogate cilium. We consider this beyond the scope of the manuscript.

Figure: Imaging of Smo localization in Th17 cell synapses. Immuno-fluorescence imaging (single x-y confocal section) of Th17 cells forming synapses on anti-TCR-coated coverslips at day 3 labelled with antibodies against Smo (green). Nuclei were stained with Hoechst (blue). Phalloidin stain is shown in orange. Scale bars: 10µm.

4- The finding of reduced P-AMPK in naive CD4+ T cells after incubation with cyclopamine suggests that Smo activity is necessary to increase AMPK activity. However, it does not indicate in its own that AMPK mediates the effect of Smo in Th17 differentiation. To conclude that non-canonical Hh signalling through AMPK partly mediates the polarisation of Th17 cells, first the authors need to demonstrate that AMPK is necessary in that process, for example with a specific AMPK inhibitor.

In this part of our manuscript we are connecting two separate previously published observations: (1) AMPK signalling is important for promoting Th17 polarization and pathogenicity in both mice and humans^{5,6}, and (2) Smo, the key signal transducer of the Hh pathway, is a regulator of AMPK phosphorylation in brown adipose tissue and neurons^{7,8}. The Blagih *et al.* paper is of particular significance since it demonstrates the importance of AMPK in Th17 effector function in the adoptive transfer colitis model (that we have used), demonstrating that AMPK is a key regulator of Th17 identity and effector function *in vivo*. We have now modified the text to clarify what was meant.

5- I would also advise to re-analyse the Gli3-target genes modulated by 5 microM cyclopamine (Smo-specific) without including the 7.5 microM cyclopamine results (concentration at which non-specific effects start to occur). The possible off target effects of 10 microM cycle are particularly evident from those clusters that show a change in the opposite direction between the 2 drug concentrations, like the top cluster and middle clusters in Fig 7D.

We have re-plotted the Gli3 target genes for 5µM cyclopamine only. Importantly, we have also validated a Gli3 target gene assessed in the “cyclopamine RNASeq” dataset by qRT-PCR analysis of Th17 cells from our *CD4CreER^{T2} Ihh^{fl/fl}* (KO) and *HET* control mice. We had already shown that Th17 cells from the KO mice have greatly reduced Gli3 levels (Fig. 8C) and now confirm that transcript levels of *Cd5l*, a predicted Gli3 target gene, are differentially regulated in Gli3-depleted cells (new Suppl. Fig. 12A). The crucial role of CD5l in Th17 biology has been shown in a recent high-impact publication⁹.

Supplemental Figure 12: Hh inhibitor treatment affects predicted, putative Gli3 target genes in Th17 cells.

RNASeq analysis of all putative Gli3 target genes as predicted by *Miraldi et al. (2019)*. Th17 cells were polarized as described above, stimulated in the presence of the indicated doses of cyclopamine or carrier control for three days and harvested at day 3. Six samples/group. **(A)** Analysis of putative Gli3 target gene expression between carrier and 5 μ M cyclopamine. **(B)** Analysis of putative Gli3 target gene expression between carrier, 5 μ M and 7.5 μ M cyclopamine. Inlay shows *Cd5l* mRNA expression as assessed by qRT-PCR from *in vitro* polarized Th17 cells harvested at day 3 isolated from tamoxifen-treated *CD4ER^{T2}Cre Ihh^{fl/+}* (HET) and *CD4ER^{T2}Cre Ihh^{fl/fl}* (KO) mice. Data is normalized to *Tbp* as a reference gene. Similar results were obtained when *CD3 ϵ* was used as a reference gene. p-values were calculated using an unpaired one-tailed Student's t test. * = p < 0.05. n = 7 mice per condition.

6- The text "Smo activates Gli transcription factors as part of the canonical pathway, but has also been shown to activate AMPK 23 or act through its GTPase activity when associated with non-canonical Hh signaling" and the diagram of Fig 7A are somewhat incorrect. Smo does not have intrinsic GTPase activity, it acts as a GEF for heterotrimeric Gi proteins and, downstream, small GTPases lie Rac1 and RhoA. Thus, the sentence should be rephrased as "or act as a GPCR through activation of Gi proteins" and the diagram updated accordingly. Indeed, the experiment with PTX inhibits Smo/Gi coupling.

We thank the reviewer for picking up on the mistake and have amended the figure and text accordingly (new Fig. 8 A).

7- Can the authors analyse changes in expression of a subset of the top Gli3-regulated genes identified by RNA-seq in the CRISPR Gli3 cells?

CRISPR is very hard to do in primary naïve CD4⁺ T cells before lineage polarization and we have only achieved a 50% reduction of *Gli3* mRNA (**Fig. 8 F**) while Th17 cells from our *CD4CreER^{T2} Ihh^{fl/fl}* (KO) mice have a 70% reduction in *Gli3* levels (**Fig. 8C**).

Figure 8F: Canonical and non-canonical Hh signaling contribute to Th17 polarization. Naïve CD4⁺ T cells were purified from spleen and peripheral lymph nodes of C57BL/6 mice and stimulated under Th17 polarizing conditions. After 24h CRISPR/Cas9 RNP complexes targeting *Gli3* were electroporated. Cells were harvested for qRT-PCR analysis of *Gli3* at day 3 (left panel) and for analysis by flow cytometry on day 5 (middle panel). Quantitation of IL-17a expression is shown on the right. n = 3 independent experiments. Data are means +/- SD. p-values were calculated using an unpaired two-tailed Student's t test. * p<0.05, ** p<0.01, *** p<0.001.

Figure 8C: Canonical and non-canonical Hh signaling contribute to Th17 polarization. (C) Th17 cells were polarized from *CD4ER^{T2}Cre⁺Ihh^{+/fl}* (HET) or *CD4ER^{T2}Cre⁺Ihh^{fl/fl}* (KO) mice and analysis of *Gli1* and *Gli3* expression was performed by qRT-PCR at day 3. n=3-6 mice. Data are means +/- SD. p-values were calculated using an unpaired two-tailed Student's t test. * p<0.05, ** p<0.01, *** p<0.001.

Thus, we have used Th17 cells from our *CD4CreER^{T2} Ihh^{fl/fl}* (KO) and HET control mice to validate *Cd5l*, a key Th17-associated gene⁹ from the list of predicted *Gli3* target genes (new Suppl. Fig. 12, see also answer to reviewer's point 5).

Supplemental Figure 12: Hh inhibitor treatment affects predicted, putative Gli3 target genes in Th17 cells.

RNASeq analysis of all putative Gli3 target genes as predicted by *Miraldi et al. (2019)*. Th17 cells were polarized as described above, stimulated in the presence of the indicated doses of cyclopamine or carrier control for three days and harvested at day 3. Six samples/group. **(A)** Analysis of putative Gli3 target gene expression between carrier and 5µM cyclopamine. **(B)** Analysis of putative Gli3 target gene expression between carrier, 5µM and 7.5µM cyclopamine. Inlay shows *Cd5l* mRNA expression as assessed by qRT-PCR from *in vitro* polarized Th17 cells harvested at day 3 isolated from tamoxifen-treated *CD4ERT2Cre Ihh^{fl/+}* (HET) and *CD4ERT2Cre Ihh^{fl/fl}* (KO) mice. Data is normalized to *Tbp* as a reference gene. Similar results were obtained when *CD3ε* was used as a reference gene. p-values were calculated using an unpaired one-tailed Student's t test. * = p < 0.05. n = 7 mice per condition.

Reviewer #2 (Remarks to the Author):

The study by Hanna et al. provides new evidence for the involvement of Hedgehog signaling in the polarization of naïve Th0 to Th17 cells. The authors show that during Th17 polarization, there is activation of the expression of Hedgehog effectors such as *Smo*, *Gli1*, and *Gli3*. Genetic and pharmacological inhibition of *Smo* function in vitro and in T-cell specific conditional mouse models reduces the in vitro polarization of Th17 cells. Of note, genetic inhibition of *Ihh* or pharmacological perturbation of *Smo* function in vivo reduces the inflammatory phenotype in two IBD mouse models with a demonstrated pathological role of Th17 cells.

This study extends our current understanding of the function of the Hedgehog signaling pathway in the immune system and in pathological processes such as inflammatory bowel disease. However, the data presented also allow other conclusions and therefore the following main points would need to be addressed by the authors:

1) It is unclear why the authors speak of intracellular Hedgehog signaling. *Ihh* has a signal peptide and is secreted by producing cells. Thus, autocrine Hedgehog signaling and a cell-autonomous role of *Ihh*/*Gli* signaling is likely. It would therefore be important to investigate in vitro polarization in the presence of *Ihh* neutralizing antibodies (mAb 5E10) (e.g. supplemental Figure 2).

This is an excellent suggestion. We have performed the experiments. Polarizing Th17 cells using different strength of Th17-polarizing conditions, in the presence of Hh blocking antibody 5E1 or isotype control does not affect Th17 polarisation (new Fig. 3 B).

This suggests that – like in CD8⁺ T cells² – *Ihh* is likely not secreted and meets its receptor Patched intracellularly.

Figure 3: Exogenous Hedgehog ligands do not affect Th17 polarization. (B) Naïve CD4⁺ T cells were stimulated with the indicated polarizing cytokines in the presence or absence of Hh ligand blocking antibody 5E1 or isotype control antibody at 10μg/ml. n = 4. Data are means +/- SD. p-values were calculated using an unpaired two-tailed Student's t test. * p<0.05, ** p<0.01, *** p<0.001.

In addition, we have polarised naïve CD4⁺ T cells from excised *CD4CreER^{T2} Ihh^{ff}* (KO) mice as well as *CD4CreER^{T2} Ihh^{+ff}* (HET) and *CD4CreER^{T2} Ihh^{+/+}* (WT) controls with increasing doses of recombinant *Ihh*-N peptide and found that exogenous active *Ihh*-N peptide cannot rescue the defect observed in the *Ihh* KO cells (new Fig. 5 C) and supporting our hypothesis

that CD4⁺ T cells cannot respond to extracellular *Ihh* ligand presumably because of the lack of Patched at the plasma membrane².

Figure 5: Conditional knockout of *Ihh* in CD4⁺ T cells leads to diminished Th17 polarization but does not affect other Th lineages. (C) Th17 cells were polarized from HET, KO mice in the presence/absence of recombinant N-terminal murine *Ihh* fragment at the indicated concentrations. Cells were harvested for analysis by flow cytometry on day 5. Representative flow cytometry plots of are shown on the left with cells treated with 200ng/ml rllhh with a summary on the right. n = 3-4 mice. Data are means +/- SD. p-values were calculated using an unpaired two-tailed Student's t test. * p<0.05, ** p<0.01, *** p<0.001.

Taken together, these data support the idea that (1) cell-endogenous *Ihh* is sufficient and required for optimal Th17 polarisation (*Ihh* KO rescue experiments) and that (2) neither exogenous Hh ligand, potentially present in the serum, or endogenous *Ihh* secreted from the CD4⁺ T cells themselves can promote Th17 polarisation (recombinant *Ihh* administration experiments + 5E1 experiments).

2) The use of cyclopamine is very problematic due to its many non-specific effects. FDA-approved vismodegib is unfortunately only used once in vitro in Figure 3C. Here, at 5µM, there is minimal, if any, effect on Th17 polarization. This is problematic as the IC50 values of vismodegib are in the range of 10-20nM. Vismodegib is effectively used in vitro at concentrations of 100-500nM and at these levels vismodegib de facto completely inhibits Smo signaling.

Regarding the concentrations of Vismodegib used, we agree with the reviewer. We used concentrations of Vismodegib that are higher than what has been used for cell lines in the literature. However, our concentrations of cyclopamine and Vismodegib are in line with concentrations used in primary murine CD8⁺ T cells without off-target effects² and other primary cells (murine rhabdomyosarcoma cells and human BCC explants)^{3, 4, 10}. Throughout the experiments we have carefully monitored for potential off-target effects by assessing viability and cell numbers of the delicate primary CD4⁺ T cell lineages when treating with Hh inhibitors (shown in Fig.4 B, right panels and Suppl. Fig.5 A, right panels – see below).

Most importantly, however we have generated and validated unique conditional and inducible *Smo* and *Ihh* KO models for CD4⁺ T cells confirming the results of the inhibitor experiments, thus showing that the effect with both Hh inhibitors is reproducible across two independent

novel mouse models, which do not suffer from potential off-target effects as small molecule inhibitors do.

Note: We find that commercially available Vismodegib is very unstable. This also might set our concentrations apart from cell line studies by Genentech done “in house”.

Supplemental Figure 5: Clinically-approved small molecule Hedgehog inhibitor vismodegib selectively blocks Th17 polarization *in vitro*.

Naïve CD4⁺T cells were stimulated under Th17 polarizing conditions in the presence of the indicated doses of vismodegib or carrier control for five days. Cells were harvested for analysis by flow cytometry on day 5. (A) Panel on the left shows representative flow cytometry plots. Quantitation of IL-17a/FoxP3 expression, viability measured by absence of live/dead staining, and cell numbers are shown on the right. n = 3-4 independent experiments

Figure 4: Small molecule Hh inhibitors selectively block Th17 polarization *in vitro*. (B) Naïve CD4⁺ T cells were stimulated under Th17 or Treg polarizing conditions in the presence of the indicated doses of cyclopamine or carrier control for three days. Cells were harvested for analysis by flow cytometry on day 5. Quantitation of IL-17a/FoxP3 expression, viability measured by absence of live/dead staining, and cell numbers are shown on the right. n = 4 independent experiments.

Continued point (2) of reviewer #2:

Thus, the role of Ihh and Smo in the in vitro polarization of Th17 cells remains unclear.

We strongly disagree.

We have designed, generated, backcrossed, and validated two independent, novel, inducible and conditional knockout models of *Ihh* and *Smo*, respectively, for CD4⁺ T cells (see **Suppl. Fig. 7 A, B** and **Suppl. Fig. 7**). Most importantly, we have shown that our models have normal T cell development in the thymus with unaltered CD4⁺ and CD8⁺ ratios and unaltered naïve (CD44^{neg}, CD62L⁺), central memory (CD44⁺, CD62L⁺), and effector memory (CD44⁺, CD62L^{neg}) subsets in the periphery. These new animal models are the cleanest models for CD4⁺ T cells so far, and are by nature devoid of off-target effects. We have used these mouse models to demonstrate an important role of Hh signalling in Th17 polarisation both *in vitro* (Fig. 5, Suppl. Fig. 6) and *in vivo* (Fig. 7F-J) - see below.

Figure 5: Conditional knockout of *Ihh* in CD4⁺ T cells leads to diminished Th17 polarization but does not affect other Th lineages. Naïve CD4⁺ T cells were purified from spleen and peripheral lymph nodes of either *CD4ERT²Cre⁺ Ihh^{fl/fl}* (HET) and *Ihh^{fl/fl}* (KO) C57BL/6 mice (Cells were stimulated under Th1, Th2, Th17 or Treg polarizing conditions and harvested for analysis by flow cytometry on day 5. n=4-6 mice per genotype. Flow cytometry plots are shown in (A) and quantification is shown in (B). Data are means +/- SD. (B,C) p-values were calculated using an unpaired two-tailed Student's t test. * p<0.05, ** p<0.01, *** p<0.001.

Supplemental Figure 6: Conditional knockout of *Smo* in CD4⁺ T cells leads to diminished Th17 polarization but does not affect other Th lineages. Naïve CD4⁺ T cells were purified from spleen and peripheral lymph nodes of either *dLckCre⁺ Smo^{+/+}* (WT) or *dLckCre⁺ Smo^{fl/fl}* (KO) C57BL/6 mice. Cells were stimulated under Th1, Th2, Th17 or Treg polarizing conditions and harvested for analysis by flow cytometry on day 5. n=2-3 mice per genotype.

Figure 6: Hh signaling is critical for Th17 responses *in vivo*. (F) *Rag2*^{-/-} mice were injected *i.p.* with 4×10^5 CD45RB^{hi} CD25^{neg} tdTom⁺ CD4⁺ T cells isolated from the spleens and peripheral lymph nodes of *CD4*^{ERT2}*Cre Ihh*^{fl/fl} (HET) or *CD4*^{ERT2}*Cre Ihh*^{ffl} (KO) mice. Mice were harvested at 6-7 weeks. (G-J) pooled data from three independent experiments shown. (G) Weight loss during the course of the experiment. (H) Colon weight/length ratios. (I) Numbers of IL-17a⁺, IL-22⁺, IL-17a⁺ IL-22⁺, IFN γ ⁺ IL-17a^{neg} T cells isolated from colonic lamina propria. Gating strategy is shown in Suppl. Fig. 9. (J) Panels on the top show representative H&E staining of the recipient mouse colons. Panel on the bottom shows quantification of histological severity scored blindly by a gastroenterologist. Scale bars: 200 μ m. Data are means \pm SEM (G). Rest are means \pm SD. p-values were calculated using an unpaired two-tailed Student's t test. * p<0.05, ** p<0.01, *** p<0.001. (J) p-values were calculated using an unpaired two-tailed Student's t test (I), a Kruskal-Wallis test (J), a one-way ANOVA (H) or a two-way ANOVA with Sidak correction (G). * p<0.05, ** p<0.01, *** p<0.001.

3) Why did the authors use different Cre-deleter strains for *Ihh* and *Smo* knockouts?

Hedgehog signalling has important roles during multiple steps of T cell development in the thymus¹¹. Thus, we took utmost care to avoid any Cre activity in the thymus.

The *dLckCre* is the best conditional Cre in T cells for this purpose, as it is expressed during the final stage of T cell development¹² and has been widely used. However, *dLckCre* deletes in both, CD4⁺ and CD8⁺ T cells. Furthermore, the *Smo*^{ffl} strain shows considerable toxicity from the Cre resulting in a much-reduced frequency of *Smo* KO mice being born. Thus, when the *CD4*^{ERT2}*Cre* mice became available from Jackson, we moved over to this Cre-deleter strain since it gives us the unique opportunity to delete in mature, naïve CD4⁺ T cells in the periphery upon tamoxifen treatment which gives us another level of control to prevent off-target toxicity.

4) The autocrine role of *Ihh-Smo-Gli3* signaling in vivo remains unclear.

In Figure 4, it would be necessary to show the total number of T cells as well as the number of *Th2* and *iTregs*.

With regards to the enumeration of total T cells and *Th2*/*Tregs*, kindly see comment/data in response to the paragraph below.

Similarly, it would be important to quantify the number of *tdTom*-positive cells in the *HET* and *KO* transplanted mice in the whole organism as well as in the affected tissue. An alternative explanation for the decreased pathologic score could be that *Ihh*-deficient T cells generally show a reduced infiltration behavior in the *Rag*^{-/-} recipient mice than *HET* cells. It would also be important to compare *Smo*-Wt and *Smo*-deficient cells, as the effect of *Ihh* could also be via paracrine signaling.

We thank the reviewer for raising this important point and giving us the opportunity to clarify. We include further data to address this point below and have amended the main text of the manuscript to address this.

For all our experiments we have assessed the total number of adoptively transferred T cells in the compartments where T cells have been shown to localise in the adoptive transfer colitis model¹³: mesenteric lymph nodes (new Suppl. Fig. 10A), spleen (new Suppl. Fig. 10B) and colon (Figure 6 I). We did not observe any significant difference in the number of adoptively transferred cells in mesenteric lymph nodes and spleen, supporting our hypothesis that the effect we observe in the colon is not the result of a general failure of the *KO* cells to homeostatically expand in the lymphopenic host.

Enumeration of T cell subsets in the main and supplemental figures is done on *tdTom*⁺ cells only, as indicated in the figure legend. The gating strategy used can be found in Suppl. Fig. 9 (renamed for clarity: Gating strategy for *CD4*⁺*tdTom*⁺ cells in the adoptive transfer colitis model).

Supplemental Figure 10: Homeostasis and maintenance of CD4⁺ T cells is unaffected by loss of *Ihh* in the adoptive colitis model.

(A-D) *Rag2*^{-/-} mice were injected *i.p.* with 4x10⁵ CD45RB^{hi} CD25⁻ tdTom⁺ CD4⁺ T cells isolated from the spleens and peripheral lymph nodes of tamoxifen-treated *CD4ER^{T2}Cre Ihh^{fl/fl}* (HET) and *CD4ER^{T2}Cre Ihh^{fl/fl}* (KO) mice. Numbers of IL-17a⁺, IL-22⁺, IL-17a⁺/IL-22⁺, IFN γ ⁺/IL-17a⁻, IFN γ ⁻/IL-17a⁻, ROR γ t⁺, FoxP3⁺CD4⁺ T cells isolated from lamina propria (LPL), mesenteric lymph node (mLN) and spleen are shown. (A-C) Gating strategy as shown in Suppl. Fig. 9. All cells are gated on tdTom⁺ CD4⁺ T cells. Data are means \pm SD. p-values were calculated using an unpaired two-tailed Student's t test. n.s. = not significant.

We have now also included total numbers of tdTom⁺ CD4⁺ T cells in the colon (LPL), mLN, spleen, and all organs combined, as well as total numbers of IL-17a^{neg}/IFN γ ^{neg} (new Suppl. Fig. 10C) in all compartments, in order to make the point very clear that we are not looking at a general maintenance or migration phenotype in the KO mice.

By contrast, we observe a specific lack of Th17 and exTh17 cells in the colon (Figure 6 I). Th17 polarisation and gut homing go hand-in-hand, since CCR6 is a hallmark of Th17 cells and the critical homing chemokine receptor to the gut. We have shown that *Ihh*-inhibited CD4⁺ T cells under Th17-polarising conditions lose CCR6 protein expression (see Fig. 4C). We have already shown that *Ihh* KO cells show diminished Th17 polarisation (see Fig. 5A/B) and CCR6 protein expression (see Fig. 5B) *in vitro* and hypothesise that the same occurs *in vivo*: meaning less well polarised Th17 cells from the KO mice express lower CCR6 levels and home less well to the gut. We have amended the manuscript text to clarify this point.

We also enumerated the total number of iTregs that developed in the adoptive transfer colitis model, which were only few in number and did not show a significant difference (new Suppl. Fig. 10D) again supporting a very specific Th17-polarisation defect in the *Ihh* KO cells.

We did not assess the number of Th2 cells by IL-4 expression in our model since it is well documented in the literature that Th2 cells do not polarise at all in the adoptive transfer colitis model¹⁴ (please see Sup. 1c of this paper).

Supplemental Figure 10: Homeostasis and maintenance of CD4⁺ T cells is unaffected by loss of *Ihh* in the adoptive colitis model.

(A-D) *Rag2*^{-/-} mice were injected *i.p.* with 4x10⁵ CD45RB^{hi} CD25⁻ tdTom⁺ CD4⁺ T cells isolated from the spleens and peripheral lymph nodes of tamoxifen-treated *CD4ER^{T2}Cre Ihh^{fl/fl}* (HET) and *CD4ER^{T2}Cre Ihh^{fl/fl}* (KO) mice. Numbers of IL-17a⁺, IL-22⁺, IL-17a⁺/IL-22⁺, IFN γ ⁺/IL-17a⁻, IFN γ ⁻/IL-17a⁻, ROR γ t⁺, FoxP3⁺ CD4⁺ T cells isolated from lamina propria (LPL), mesenteric lymph node (mLN) and spleen are shown. (A-C) Gating strategy as shown in Suppl. Fig. 9. All cells are gated on tdTom⁺ CD4⁺ T cells. Data are means +/- SD. p-values were calculated using an unpaired two-tailed Student's t test. n.s. = not significant.

Regarding the possibility of paracrine signaling occurring in our model, we have conducted further experiments to rule this out based on the following observations:

Firstly, we have shown that – like in CD8⁺ T cells² – CD4⁺ T cells do not respond to extracellular active *Ihh* ligand (Fig. 3A and Suppl. Fig. 4C). (Most likely because the receptor Patched is not localised on the plasma membrane and meets the *Ihh* ligand intracellularly on vesicles².) Secondly, our new data indicates that blocking of all extracellular *Ihh* ligand using 5E1 is not interfering with Th17 polarisation (new Fig. 3B).

And thirdly, our new data shows Th17 polarisation in *Ihh* KO CD4⁺ T cells cannot be rescued by addition of exogenous active *Ihh* (new Fig. 5C and Suppl. Fig. 4C).

Figure 3: Exogenous Hedgehog ligands do not affect Th17 polarization. (B) Naïve CD4⁺ T cells were stimulated with the indicated polarizing cytokines in the presence or absence of Hh ligand blocking antibody 5E1 or isotype control antibody at 10 μ g/ml. n = 4. Data are means +/- SD. p-values were calculated using an unpaired two-tailed Student's t test. * p<0.05, ** p<0.01, *** p<0.001.

Figure 5: Conditional knockout of *Ihh* in CD4⁺ T cells leads to diminished Th17 polarization but does not affect other Th lineages. (C) Th17 cells were polarized from HET, KO mice in the presence/absence of recombinant N-terminal murine *Ihh* fragment at the indicated concentrations. Cells were harvested for analysis by flow cytometry on day 5. Representative flow cytometry plots of are shown on the left with cells treated with 200ng/ml rlhh with a summary on the right. n = 3-4 mice. Data are means +/- SD. p-values were calculated using an unpaired two-tailed Student's t test. * p<0.05, ** p<0.01, *** p<0.001.

5) Figure 4 C and H: why does vismodegib not affect the number of IL17a-/Ifng⁺ cells, while loss of *Ihh* function does?

Since there is no Fig. 4H, we assume the reviewer is referring to **Figure 6 C and H**.

Th17 cells display great plasticity and can acquire Th1 characteristics (Ifn γ -production) *in vivo*¹⁵. Th17 cells giving rise to Th1 cells is in fact required for the pathogenesis of adoptive transfer colitis over the course of 4 weeks¹⁶. However, using the Flavell colitis model that lasts only two days¹⁷, does not give enough time to induce sufficient numbers of IL17a-/Ifn γ + (exTh17) cells (see **Fig. 6D**) for Vismodegib to have a statistically significant effect. We have amended both the results and discussion to clarify this point.

6) Figure 4H: do the cell numbers change for iTregs and Th2?

Since there is no Fig. 4H, we assume the reviewer is referring to **Figure 6H** (colitis model *in vivo*).

We have enumerated the total number of iTregs that developed in the adoptive transfer colitis model, which were only few in number and did not show a significant difference (new Suppl. **Fig. 10D**) again supporting a very specific Th17-polarisation defect in the *Ihh* KO cells. We did not assess the number of Th2 cells by IL-4 expression in our model since it is well documented in the literature that Th2 cells do not polarise at all in the adoptive transfer colitis model we have used¹⁴(please see Sup. 1c of this paper).

Supplemental Figure 10: Homeostasis and maintenance of CD4⁺ T cells is unaffected by loss of *Ihh* in the adoptive colitis model.

(A-D) *Rag2*^{-/-} mice were injected *i.p.* with 4×10^5 CD45RB^{hi} CD25⁻ tdTom⁺ CD4⁺ T cells isolated from the spleens and peripheral lymph nodes of tamoxifen-treated *CD4ER*^{T2}*Cre Ihh*^{fl/+} (HET) and *CD4ER*^{T2}*Cre Ihh*^{fl/fl} (KO) mice. Numbers of IL-17a⁺, IL-22⁺, IL-17a⁺/IL-22⁺, IFN γ ⁺/IL-17a⁻, IFN γ ⁻/IL-17a⁻, ROR γ t⁺, FoxP3⁺ CD4⁺ T cells isolated from lamina propria (LPL), mesenteric lymph node (mLN) and spleen are shown. Data are means \pm SD. p-values were calculated using an unpaired two-tailed Student's t test. n.s. = not significant.

7) The role of *Gli3* in Th17 polarization is not fully understood. *Gli3* activity is primarily regulated by proteolytic processing. It is therefore essential to investigate this at least by Western blotting (*Gli3R* vs *Gli3* full-length) in the course of T cell polarization. In Fig. 2A, *Gli3* mRNA levels increase sharply and *Gli1* levels decrease. This could indicate *Gli3* repressor function. Similarly, it would support the authors' conclusions if *Gli3*-KO cells were examined in the adoptive transfer model.

We agree with the reviewer that characterisation of *Gli3* processing in Th17 cells would be very interesting. We have tried extensively to blot for *Gli3* using a well-used antibody in the Hh fibroblast field (R&D AF3690) as well as other commercial anti-*Gli3* antibodies. *Gli3* levels in T cells might be very low compared to MEFs (see Fig. 1E and new Suppl. Fig. 2 B) and unfortunately anti-*Gli3* antibodies generated many non-specific bands, so that the Western blot experiments cannot be clearly interpreted.

We have however been able to contribute substantially to *Gli3* biology in Th17 cells by showing that TGF β is specifically responsible for the induction of *Gli3* (new Fig. 2C) in this cell type.

Unfortunately, it is not possible to perform the adoptive transfer model using *Gli3*-KO CD4⁺ T cells for the following reasons:

- 1) *In vivo* adoptive transfer experiments require that all mice (donors and recipients) are genetically matched on the same C57BL/6 background. As Hh-component floxed mice, including the available *Gli3*^{fl/fl} mice, are historically kept on a mixed background for developmental studies, it would take us over 11 generations of backcrossing (a minimum of 2.3 years) to generate appropriate mice for these studies.
- 2) CRISPR of primary naive CD4⁺ T cells is extremely hard to do and after extensive optimisation we have only been able to deplete *Gli3* in a subset of cells (Fig. 8F), ca. 50%. Since in live cell suspensions we cannot distinguish between *Gli3*⁺ from *Gli3*^{neg} cells, the injection of an uncharacterised mixed population is likely not to yield meaningful data.

Figure 2: IL-6 and TGF β are the primary inducers of Hh signaling components in Th17 cells. (C) Cells were polarized *in vitro* under full Th17-polarising conditions and harvested at day 3 post stimulation for qRT-PCR of *Gli3* and *Smo*. Cells were treated for three days with either 2 μ M STATTC, 1 μ M SB 505124 or carrier control; naïve CD4⁺ T cell RNA levels (“0h”) are shown as controls. Data is normalized to *Tbp* as a reference gene. Similar results were obtained when *CD3 ϵ* was used as a reference gene. n = 4. Data are means \pm SD. p-values were calculated using an unpaired two-tailed Student’s t test. * p<0.05, ** p<0.01, *** p<0.001.

8) Figure 4: why do *CD4CreER;Ihhfl/fl* mice differ from *dLckCre;Smo-/-* mice in terms of the number of *Ifngamma* pos. Th1 cells (increase by *Ihh* KO, no change by *Smo*-KO)? Similarly, for *iTreg* cells. This suggests a non-cell autonomous role of *Ihh* in Th cell polarization and somewhat contradicts the conclusions of the authors.

Due to COVID19 we had to drastically reduce our mouse colony. We have now analyzed more *CD4CreER^{T2} Ihh^{fl/fl}* (KO) and *CD4CreER^{T2} Ihh^{fl/+}* (HET) mice and show that IFN γ -production in the KO cells is not significantly different from control cells (new Fig. 5B).

Figure 5: Conditional knockout of *Ihh* in CD4⁺ T cells leads to diminished Th17 polarization but does not affect other Th lineages. Naïve CD4⁺ T cells were purified from spleen and peripheral lymph nodes of either *CD4ER^{T2}Cre⁺ Ihh^{+fl/fl}* (HET) and *Ihh^{fl/fl}* (KO) C57BL/6 mice (Cells were stimulated under Th1, Th2, Th17 or Treg polarizing conditions and harvested for analysis by flow cytometry on day 5. n=4-6 mice per genotype. Flow cytometry plots are shown in (A) and quantification is shown in (B). Data are means +/- SD. (B,C) p-values were calculated using an unpaired two-tailed Student's t test. * p<0.05, ** p<0.01, *** p<0.001.

9) *Smo* Western blots of T cells show two bands in contrast to MEFs. Why, and which one is specific? It would be important to study *Smo* protein expression in T cells from wild-type and *Smo*-deficient mice, preferably also by confocal microscopy.

We have included western blot analysis of Th17 cells from our *Smo* KO and WT mice showing that all three bands are specific for *Smo* in T cells (new Suppl. Fig. 3 C).

Supplemental Figure 3: Validation of novel monoclonal antibodies raised against the C-terminus of murine *Smo* (mcSmo). (C) Immunoblot analysis of *Smo* expression in Th17 cells polarized from spleens of tamoxifen-treated *CD4ER^{T2}Cre Smo^{+/+}* (WT) and *CD4ER^{T2}Cre Smo^{fl/fl}* (KO) mice. n = 2 independent experiments with 2-3 mice per condition.

We agree with the reviewer that it is extremely interesting that T cells express multiple specific bands of *Smo* and we are actively investigating the reason for this. We hypothesize that post-translational modifications of *Smo* especially differential glycosylation status - as observed by Ogden and co-workers for *Drosophila Smo*¹⁸ - are responsible for the multiple bands. We consider this work beyond the remit of the current manuscript.

10) Please add a positive control for *Gli2* to support the absence of *Gli2* mRNA expression.

Mouse Embryonic Fibroblasts (MEFs) and murine thymus have been reported to be positive for *Gli2*. We have thus included qRT-PCR analysis of MEFs and thymic tissue as additional controls and shown that our *Gli2* probe faithfully reports *Gli2* expression (new Suppl. Figure 2A, B).

Supplemental Figure 2: Expression of Hh signaling components during CD4⁺T helper cell polarization. (A) Naïve CD4⁺T cells were purified from spleen and peripheral lymph nodes of C57BL/6 mice and stimulated with plate-bound anti-CD3 ϵ /CD28 antibodies in the presence of polarizing cytokines to generate Th0, Th1, Th2, Th17 and Treg subsets. Expression of *Ptch1*, *Ptch2*, *Smo*, *Shh* and *Dhh* were assessed by qRT-PCR in Th subsets at the indicated timepoints after TCR stimulation in the presence of polarizing cytokines. Data is normalized to *Tbp* as a reference gene. Similar results were obtained when *CD3 ϵ* was used as a reference gene. $n = 3$ independent experiments. *Gli2/Dhh* mRNA was undetectable across all conditions tested. Data are means \pm SD. nd = not detected. As a positive control, expression levels of Hh components in Mouse Embryonic Fibroblasts (MEFs) were assessed (right column). $n = 3$. Data is normalized to *Tbp* as a reference gene. $n = 3$ independent experiments. (B) Left: Mouse Embryonic Fibroblasts (MEFs) were assessed for *Gli1* and *Gli3* expression by RT-qPCR. $n = 3$. Right: Testes and thymi were dissected from C57BL/6 mice ($n=4$), homogenized and lysed for qRT-PCR analysis of *Dhh* and *Gli2* mRNA expression, respectively.

11) The relevance to human pathology remains undefined. At least *Ihh-Smo-Gli3* expression should be quantitatively investigated in T cells (at least in Th17) from IBD patient tissue.

We thank the reviewer for this excellent suggestion and have performed the experiments.

We have analysed published datasets of >99% pure, CD45RO⁺ (activated and memory cells) sorted human CD4⁺ T cells from blood, gut epithelium and gut lamina propria of healthy controls¹⁹. Importantly, we found that Hedgehog components IHH, SMO, and GLI3 are all expressed in human effector CD4⁺ T cells from the blood and the gut mucosa (new Fig. 9 A, B).

Next, we wanted to investigate whether expression levels of Hh components would change in patients with IBD compared to matched healthy controls. For this we analysed datasets of rectal biopsies from patients with ulcerative colitis (UC) – a paediatric cohort (discovery cohort)^{20, 21} and an adult cohort (validation cohort)²². Reassuringly, we found significantly higher expression of *SMO* and *GLI3* in gut biopsies of patient with UC compared to healthy controls. Most importantly however is our finding that *SMO* and *GLI3* expression highly correlated with hallmarks of Th17 cells – the Th17 marker CCR6 and the Th17 effector cytokine IL-17A (new Fig. 9 C, D and new Suppl. Fig. 13).

Figure 9: Hedgehog signaling components are expressed in human immune cells in the gut and are upregulated in ulcerative colitis. Analysis of human RNA-Seq datasets summarized in panel (A). (B) Expression of *Ihh*, *Smo* and *Gli3* mRNA from FACS-sorted CD4+ T cells isolated from blood as well as intraepithelial lymphocytes (IELs) and lamina propria lymphocytes (LPLs) isolated from the terminal ileum from healthy human donors in a study from *Raine et al.* (2015). n = 6. Error bars show SD. (C, D) Expression of *Smo* and *Gli3* mRNA from human rectal biopsy samples from patients with male (♂) or female (♀) ulcerative colitis (UC) or healthy controls recruited as part of the PROTECT study (GSE109142). Each datapoint represents an individual patient. (C) shows expression of *SMO* and *GLI3* across all groups. p-values were calculated using a limma based moderated t test. * p<0.05, ** p<0.01, *** p<0.001. (D) shows correlation analysis of *SMO*, *GLI3* as well as Th17-related transcripts *CCR6* and *IL17a*, respectively, with simple linear regression analysis.

Supplemental Figure 13: Hedgehog signaling components are upregulated in an adult ulcerative colitis cohort and expression of SMO and GLI3 correlate with markers of Th17 cell infiltration. (A, B) Expression of *SMO* and *GLI3* mRNA in human rectal biopsy samples from patients with active ulcerative colitis (UC) or healthy controls recruited as part of the RISK study (GSE117993). Each datapoint represents an individual patient. **(A)** shows expression of *SMO* and *GLI3* across all groups. p-values were calculated using a limma based moderated t test. ** p<0.01, *** p<0.001. **(B)** shows correlation analysis between *SMO* and *GLI3* and Th17-related transcripts *CCR6* and *IL17a* using simple linear regression analysis.

12) The apparent anti-inflammatory effect of vismodegib is in contradiction to previous publications in which, for example, a clear pro-inflammatory effect has been shown in colitis models (e.g. Lee, et al. Control of Inflammation by Stromal Hedgehog Pathway Activation Restrains Colitis. Proceedings of the National Academy of Sciences of the United States of America 2016, 113 (47), E7545-E7553. <https://doi.org/10.1073/pnas.1616447113>). The current study would benefit from experiments using the Smo Agonist (SAG) for in vivo treatment. Thus, it would be important to investigate whether SAG treatment affects the phenotype of Rag-/- mice transplanted with *Ihh*-deficient T cells.

Our data complements the findings of Lee, et al. in characterising the role of Hh in gut inflammation. The different results obtained with regard to Hh inhibitor treatment are likely due to differences in the models used:

Lee, et al. uses a DSS colitis model. In this model, acute chemical injury leads to death and shedding of the majority of the gut epithelium (especially at the high concentrations of DSS

used by Lee, et al.). As a result, the lamina propria is exposed and penetrating microbiota as well as the injury-induced cell death leads to a dramatic activation of mainly the innate immune system but not so much Th17 cells. After the injury, DSS colitis is resolved by wound healing and tissue repair in a matter of days. The lack of Th17 involvement and the self-resolving nature of the DSS colitis make this model a very poor model of human IBD. However, we agree with Lee, et al. that Hedgehog signalling in the stroma is likely contributing beneficially to the wound healing response.

By contrast, we have used a colitis model from the Flavell lab¹⁷. In this model, injection of CD3-specific antibody leads to activation-induced cell death of T cells and results in systemic upregulation of IL-6 and TGF β induced by phagocyte engulfment of apoptotic T cells. The combination of these cytokines leads to the development of Th17 cells that consequently home to the small intestine (high steady state levels of Ccl20, the ligand for CCR6) via CCR6 causing colitis.

Taken together, systemic Hh inhibitor treatment has different primary targets – gut stroma (Lee, et al.) and pathogenic Th17 (current manuscript).

Systemic SAG treatments of mice during our 6-week adoptive colitis model are impossible to perform due to Home Office regulations in the UK.

Minor points:

1) Western blot (not western)

We have amended this in the revised manuscript.

2) Tfh cells should also be mentioned in the introduction

We have amended this in the revised manuscript.

3) Line 99 is not totally correct; a suggestion could be: ... where it activates the Gli transcription factors Gli2 and Gli3. These translocate to the nucleus and initiate the transcription of Hh target genes including Gli1.

We have amended this in the revised manuscript.

4) Figure 2A: The amount of tubulin loading control varies relatively strongly. Densitometric quantification of Smo/tubulin amounts would be helpful.

We have included the analysis in the revised manuscript (**new Suppl. Fig. 4 B**).

Supplemental Figure 4: Expression of IL-17a expression in different Th17-polarizing conditions, densitometric quantification of Smo Western Blots and validation of recombinant Ihh.

(A,B) Naïve CD4⁺T cells were purified from spleen and peripheral lymph nodes of C57BL/6 mice and stimulated with plate-bound anti-CD3 ϵ /CD28 antibodies in the presence of the indicated polarizing cytokines. (B) Cells were assessed for Smo and α -tubulin protein levels by Western Blot at day 3. Data shows densitometric expression of Smo protein levels normalized to α -tubulin expression relative to the TGF-blocking antibody-treated condition. Data are means +/- SD.

5) Figure 2A: Why is there such a significant difference in Smo protein levels between TGFB-block and TGF-beta untreated samples?

T cell receptor-activated CD4⁺ T cells produce TGF β ²³. The high amount of blocking antibody used eliminates this endogenous signalling.

Reviewer #3 (Remarks to the Author):

The manuscript by Hanna et al., investigates the role of the Hh pathway in Th cells differentiation. The authors claim that Hh signalling cascade, via Smo and Gli3 is critical for the generation of Th17 cells.

Major concerns:

1) This reviewer would like to see a more systematic presentation of the results: especially showing the repeats as individual symbols in bar graphs and showing all subsets in critical experiments.

We thank the reviewer for this suggestion and have adjusted all graphs /figures throughout the manuscript.

E.g. Hh inhibitors were only tested on Th17 and Treg. Often Treg are robust, this needs to be tested on Th1 cells in addition as a minimum (concentration range to 10uM, not just one).

We have performed the experiments and shown no effect of Hh inhibitors on Th1 polarisation, even at high concentrations (new Suppl. Fig. 5D).

Supplemental Figure 5D. Naïve CD4⁺ T cells were stimulated under Th1 polarizing conditions in the presence of the indicated doses of carrier control as previously described. Cells were harvested for analysis by flow cytometry on day 5. Full dose titration was performed as n = 2.

The effect of the absence of Gli1 is assessed for all subsets, but absence of Gli3 only for Th17 cells, while this is critical.

We have tested the effect of Smo inhibition, genetic Smo loss, and genetic Ihh loss on all subsets (Fig. 5, Suppl. Fig. 6). Ihh and Smo are key upstream signalling components of Gli3 and inhibition of Ihh and Smo led to Gli3 loss (Fig. 8C/D). Since CRISPR of Gli3 has been extremely challenging, we did not perform the experiments for all CD4⁺ T cell subsets.

2) The data do not support the conclusions (or the title).

We have changed the original manuscript title to “Cell-autonomous Hedgehog signaling controls Th17 polarization and pathogenicity”. This change is guided by new experiments showing that blocking extracellular Ihh ligand with 5E1 does not affect Th17 polarisation (new Fig.3B) and that addition of exogenous Ihh cannot rescue the Th17 polarisation defect observed in the Ihh KO cells (new Fig.5C).

The authors make claims regarding Th17 cells, but till figure 7, only analyse IL17A production. This leaves the reader unsure if the regulation is at the level of the IL17A cytokine or if indeed Th17 differentiation is affected. In final figure 7, the authors finally show that Th17 generation is not affected, with the typical transcriptional

programme (ROR γ t, ROR α , etc) intact. Yet, the authors keep referring to and draw conclusions regarding Th17 cells in general.

We have not stopped at one but have examined the following crucial markers of Th17 identity:

- 1) IL-17a production (Fig. 3-8)
- 2) Upregulation of the Th17 hallmark chemokine receptor CCR6 (Fig. 4, Fig. 5, Suppl. Fig. 5)
- 3) IL17f production (Suppl. Fig. 5)
- 4) IL-22 production (Fig. 6, Suppl. Fig. 10)

Taken together, our data supports the idea that it is Th17 polarisation (rather than IL-17a production only) that is affected based on our analysis of (1-4), and we have modified the text to clarify this.

In addition, we show that the Th17 transcriptional programme as a whole is profoundly perturbed when Hh signaling is inhibited (RNASeq, GSEA analysis, Fig. 7D and Suppl. Fig. 11). The data shown in Fig. 7B merely serves to show that the perturbation in Th17 transcriptional programme is not due to known transcription factors changing. This is in line with our model that it is Gli3 and AMPK that regulate Th17 identity in a ROR γ t/ROR α -independent manner.

Supplemental Figure 5: Clinically-approved small molecule Hedgehog inhibitor vismodegib selectively blocks Th17 polarization *in vitro*.

Naïve CD4⁺ T cells were stimulated under Th17 polarizing conditions in the presence of the indicated doses of vismodegib or carrier control for five days. Cells were harvested for analysis by flow cytometry on day 5. (B) Panel on the left shows representative flow cytometry plots of IL-17f expression. Quantitation shown on the right. n = 3-4 independent experiments. (C) Panel on the left shows representative flow cytometry plots of CCR6 expression. Quantitation shown on the right. n = 3-4 independent experiments. (D) Naïve CD4⁺ T cells were stimulated under Th1 polarizing conditions in the presence of the indicated doses of cyclopamine or carrier control as previously described. Cells were harvested for analysis by flow cytometry on day 5. Full dose titration was performed as n = 2. Data are means +/- SD. p-values were calculated using an unpaired two-tailed Student's t test. * p<0.05, ** p<0.01, *** p<0.001.

Figure 5: Conditional knockout of *Ihh* in CD4⁺ T cells leads to diminished Th17 polarization but does not affect other Th lineages. Naïve CD4⁺ T cells were purified from spleen and peripheral lymph nodes of either *CD4ER^{T2}Cre⁺ Ihh^{fl/fl}* (HET) and *Ihh^{fl/fl}* (KO) C57BL/6 mice (Cells were stimulated under Th1, Th2, Th17 or Treg polarizing conditions and harvested for analysis by flow cytometry on day 5. n=4-6 mice per genotype). Flow cytometry plots are shown in (A) and quantification is shown in (B). Data are means \pm SD. (B,C) p-values were calculated using an unpaired two-tailed Student's t test. * p<0.05, ** p<0.01, *** p<0.001.

At end of page 10 (and 14), the authors conclude “Taken together, we found that *Hh* signalling is crucial for Th17 polarization but not Th1, Th2, or iTreg differentiation.” This is a very strong statement. We have modified the text to say that our (*in vitro*) data is “consistent with *Hh* signalling having important roles...”. Especially when they show in Figure 4, that the genetic absence of *Ihh* reduced IL17, but certainly does not ablate it, nor is there info on Th17 cells. Especially in light of the very good deletion shown in Suppl Fig 6. It seems *Hh* signalling may contribute to IL17A production, but is not crucial. Later experiments indicate Th17 cells are fine and the observations at best are limited to a subset of the Th17 transcriptional programme.

The Th17 transcriptional network is very robust, particularly in the context of very high cytokine concentrations used in all *in vitro* polarisation systems. This means that even in the absence of major regulators of Th17 identity, substantial IL-17a production still occurs *in vitro*. Good examples are AHR and RORa (both undisputed Th17 master regulators): see publications ²⁴(Fig.1b), ²⁵(Fig.3A).

Loss of either led to “only” a ca. 50% reduction in the proportion of IL-17a+ cells, which is very comparable with our data from *Ihh* and *Smo* KO mice (Fig. 5, Suppl. Fig. 6). The reason why RORa and AHR are nonetheless master Th17 regulators comes from *in vivo* data which show that loss-of-function profoundly impairs Th17 inflammatory responses, exactly as we have shown for *Ihh* KO mice *in vivo* (see Fig. 6).

3) Absence of molecular mechanism.

Expression of *Gli1* is found in Th1, 17 and Treg, expression of *Gli3* in Th17 and Treg, *Smo* mRNA expression in all, but *Smo* protein mainly in Th17. Why is this?

Gli1 is very lowly expressed across all Th subsets analysed and expression is likely to be induced in naïve CD4⁺ T cells via the T cell receptor as has been described for naïve CD8⁺ T cells².

Treg and Th17 cells need TGFβ in their polarisation media. We now provide new data showing that expression of Gli3 is dependent on active TGFβ signalling using a potent TGFβ-blocker (new Fig. 2C). Thus, only Th subsets with TGFβ in the culture medium (Treg and Th17) express Gli3.

Smo mRNA is induced across all Th subsets analysed and expression is likely to be induced in naïve CD4⁺ T cells via the T cell receptor as has been described for naïve CD8⁺ T cells². However, Smo mRNA expression is highest in Th17 cells (Suppl. Fig. 2) and we have generated new data showing that IL6 - only present in Th17 polarising conditions - is critical to maintain Smo mRNA expression via Stat3 signalling (new Fig. 2C). Consequently, loss of IL6 from Th17 cultures leads to loss of Smo protein expression (Fig. 2B, new Suppl. Fig. 4B)

Figure 2: IL-6 and TGFβ are the primary inducers of Hh signaling components in Th17 cells. (A-C) Naïve CD4⁺ T cells were purified from spleen and peripheral lymph nodes of C57BL/6 mice and polarized with the indicated polarizing cytokines. **(A,B)** Cells were harvested at day 3 post stimulation for immunoblot analysis of Smo and qRT-PCR analysis of *Gli1* and *Gli3*. TGFβ blocking antibody (clone: 1D11) was added in the indicated condition for the duration of polarization at 10μg/ml. Data is normalized to *Tbp* as a reference gene. Similar results were obtained when *CD3ε* was used as a reference gene. n = 3 independent experiments. **(C)** Cells were polarized *in vitro* under full Th17-polarising conditions and harvested at day 3 post stimulation for qRT-PCR of *Gli3* and *Smo*. Cells were treated for three days with either 2μM STAT3IC, 1μM SB 505124 or carrier control; naïve CD4⁺ T cell RNA levels ("0h") are shown as controls. Data is normalized to *Tbp* as a reference gene. Similar results were obtained when *CD3ε* was used as a reference gene. n = 4. Data are means +/- SD. p-values were calculated using an unpaired two-tailed Student's t test. * p<0.05, ** p<0.01, *** p<0.001.

Supplemental Figure 4: Expression of IL-17a expression in different Th17-polarizing conditions, densitometric quantification of Smo Western Blots and validation of recombinant lhh.

(A,B) Naïve CD4⁺T cells were purified from spleen and peripheral lymph nodes of C57BL/6 mice and stimulated with plate-bound anti-CD3 ϵ /CD28 antibodies in the presence of the indicated polarizing cytokines. (B) Cells were assessed for Smo and α -tubulin protein levels by Western Blot at day 3. Data shows densitometric expression of Smo protein levels normalized to α -tubulin expression relative to the TGF-blocking antibody-treated condition. Data are means +/- SD.

Is there cross-reactivity (validation of antibody only on KO and overexpression)?

We have included western blot analysis of Th17 cells from our *Smo* KO and WT mice showing that all three bands are specific for Smo in T cells (new Suppl. Fig. 3 C).

Supplemental Figure 3: Validation of novel monoclonal antibodies raised against the C-terminus of murine Smo (mcSmo). (C) Immunoblot analysis of Smo expression in Th17 cells polarized from spleens of *CD4ER^{T2}Cre Smo^{+/+}* (WT) and *CD4ER^{T2}Cre Smo^{fl/fl}* (KO) mice. n = 2 independent experiments with 2-3 mice per condition.

***Smo* is assessed by WB only, while other factors are analysed by qPCR.**

Smo and *Ptch* expression is analysed by WB (Fig.1C) and qRT-PCR (Suppl. Fig. 2A).

The running pattern in Figure 2A shows 2 bands for Smo of distinctly different weights, Figure 2B, two bands are close in molecular weight.

We consider that the band at ca. 150kDa in Figure 2A is caused by the TGF-beta blocking antibody. Our anti-mouse secondary antibody picks up the TGF β -blocking antibody which was also raised in mouse (Clone 1D11).

(*Smo* is very unstable and in order to detect it we do not boil the lysates (see materials and methods), thus whole IgG is running on the gel)

Condition in Fig2A, IL-6, -23 and 1b does not show Smo protein, which undermines the authors' conclusion that IL-6 is required for Smo induction. How does IL-6 induce Smo protein induction, is this STAT3 mediated? All this undermines the confidence in the data.

We have now included densitometric analysis of Smo protein (new Suppl. Fig. 4B) and show that only in the absence of IL-6, Smo protein expression is lost.

Supplemental Figure 4: Expression of IL-17a expression in different Th17-polarizing conditions, densitometric quantification of Smo Western Blots and validation of recombinant Ihh.

(A,B) Naïve CD4⁺T cells were purified from spleen and peripheral lymph nodes of C57BL/6 mice and stimulated with plate-bound anti-CD3 ϵ /CD28 antibodies in the presence of the indicated polarizing cytokines. (B) Cells were assessed for Smo and α -tubulin protein levels by Western Blot at day 3. Data shows densitometric expression of Smo protein levels normalized to α -tubulin expression relative to the TGF-blocking antibody-treated condition. Data are means \pm SD.

The authors establish in Figure 7 that there is no effect on Th17 cell differentiation since the main transcriptional programme is present. i.e. the cells still express the required transcriptional programme, but one (or more) aspects of the programme, namely the translation/transcription of Il17A is affected. Yet, the authors keep referring to Th17 differentiation thereafter.

In our RNAseq dataset we have shown that the Th17 transcriptional programme dramatically changes upon loss of Hh signalling. We have performed Gene Set Enrichment Analyses (hundreds of genes) and shown that Hh-inhibited Th17 cells lose Th17 identity (Fig. 7D and Suppl. Fig. 11).

In addition, we kindly refer the reviewer to our earlier comments (point 2 of this reviewer, page 29-30).

Figure 7: Mechanistic analysis of Th17-related signaling nodes. (D) Normalized Enrichment Scores (NES) from GSEA of (C) demonstrating a loss of Th17 identity upon Hh inhibitor treatment. Data are means \pm SD. p-values were calculated using an unpaired two-tailed Student's t test. ** indicates $p < 0.01$.

Supplemental Figure 11: Gene Set Enrichment Analysis of RNASeq data demonstrates loss of Th17 identity upon Hh inhibitor treatment.

(A) Gene set enrichment plots (GSE11924 and GSE14308) using RNASeq data shown in Fig. 7D. Th17 cells were polarized as described, stimulated in the presence of the indicated dose of cyclopamine or carrier control for three days, and harvested on day 3. Six samples/group.

Gli1 is excluded (checked on all Th subsets), and Gli3 may have a supportive role. But figure 7E does not show a large difference that explains previously reported marked effects on IL17A. Yet, the authors conclude “we have uncovered Gli3 as a fundamental new TF implicated in Th17 polarization” I do not find the data supporting this conclusion sufficiently. Again, critical controls (other subsets) are missing here, e.g. in the pAMPK analysis.

CRISPR knockdown in undifferentiated primary CD4⁺ T cells is extremely challenging and in the context of Gli3 required extensive optimisation. The fact that a partial ablation (in ca. half of electroporated T cells) of Gli3 in a subset of cells led to a 20% reduction in IL-17a production is astonishing, given the results seen in CD4⁺ T cells from AHR and RORa (both undisputed Th17 master regulators) KO mice: see publications ²⁴(Fig.1b), ²⁵(Fig.3A). Loss of either led to “only” a ca. 50% reduction in the proportion of IL-17a+ cells, which is very comparable with our data from *Ihh* and *Smo* KO mice (**Fig. 5, Suppl. Fig. 6**). The reason why RORa and AHR are nonetheless master Th17 regulators comes from *in vivo* data which show that loss-of-function profoundly impairs Th17 inflammatory responses, exactly as we have shown for *Ihh* KO mice *in vivo* (**see Fig. 6**) where loss of Hh signalling profoundly impairs Th17 inflammatory responses. Furthermore we propose that the Hh pathway is acting not only via Gli3 but also via pAMPK (Fig.8G), which would explain why Gli3 loss only partially recapitulates the phenotype of *Ihh*/*Smo* loss (since they act to regulate Th17 function via AMPK as well).

For a complete picture of the mechanistic detail of Hh-mediated Th17 polarisation, we are investigating pAMPK. Here, we are connecting two separate published observations: (1) AMPK signalling has been shown to be important for promoting Th17 polarization and pathogenicity in both mice and humans ^{5, 6}, (hence the focus in our analysis on only Th17 cells) and (2) *Smo*, the key signal transducer of the Hh pathway, is a regulator of AMPK phosphorylation in brown adipose tissue and neurons ^{7, 8}.

The Blagih *et al.* paper is of particular significance since it demonstrates the importance of AMPK in Th17 effector function in the adoptive transfer colitis model (that we have used), demonstrating that AMPK is a key regulator of Th17 identity and effector function *in vivo*. By showing that protein levels of pAMPK as well as upstream regulating kinases CaMKK2 and LKB1 are reduced in Th17 cultures in the presence of Hh inhibitors (**Fig. 8G**) we expand our thorough mechanistic analysis.

4) Physiological relevance:

The colitis models seem to show a reduced infiltration/recruitment of T cells, also Th1 cells are diminished in the second model, and not shown in the first model (as under point 1, a more systematic analysis is required).

Th17 cells have been shown to display great plasticity and can acquire Th1 characteristics (Ifn γ -production) *in vivo* ¹⁵. Importantly, Th17 cells giving rise to Th1 cells is required for the pathogenesis of adoptive transfer colitis over the course of 4 weeks ¹⁶.

The short-term colitis model (**Fig. 6A-E**) established by the Flavell laboratory¹⁷ lasts only two days and as a result does not give enough time to induce transdifferentiation/sufficient numbers of IL17a⁻/Ifn γ ⁺ (“exTh17”) cells for Vismodegib to have a statistically significant effect (**see Fig. 6D**).

We have amended both the results and discussion to clarify this point.

Although similar T cell numbers are present in the periphery, the expression of Smo etc during thymic selection may alter TCR selection. Transfer colitis depends on high-affinity TCRs responding to the environment in lymphopenic mice, giving some doubts to the use of this model.

We thank the reviewer for the opportunity to explain the unique advantages of our novel inducible, traceable, CD4⁺ T cell-specific *Ihh* KO model (*CD4ER^{T2}Cre Ihh^{fl/fl} ROSA26-tdTom*) we have developed in the laboratory.

Hedgehog signalling has important roles during multiple steps of T cell development in the thymus¹¹. Thus, we took utmost care to avoid any Cre activity during thymic development. We treat adult *CD4ER^{T2}Cre ROSA26-tdTom* mice (*Ihh^{fl/fl}* and *Ihh^{fl/+}* controls) for 4 days with Tamoxifen and on day 4 harvest mature naïve CD4⁺ T cells from the periphery (spleen). The *CD4ER^{T2}Cre* model is the only model that excises in mature CD4⁺ T cells (gold standard) and our treatment regime excludes any effect of *Ihh*-deletion on thymic maturation as well as TCR selection²⁶.

The models used do not distinguish between a more general role in T cell biology vs a specific role in Th17 cell differentiation and function or need much more robust analysis.

We thank the reviewer for the opportunity to correct this misunderstanding.

The adoptive transfer colitis model is a very well validated model for studying Th17 biology since it recapitulates many key features of Th17-induced inflammation including the phenomenon of Th17 plasticity and transdifferentiation. It is also considered one of the most physiologically relevant models of inflammatory bowel disease since it recapitulates the *de novo* polarisation of Th17 cells in the gut against commensal microorganisms (reviewed in²⁷, p.161-164). It has been extensively used to study Th17 polarization *in vivo* (see for example^{28, 29, 30}).

Importantly, we have shown that the total number of adoptively transferred T cells in the compartments where T cells have been shown to localise in the adoptive transfer colitis model¹³ - mesenteric lymph nodes (**new Suppl. Fig. 10A**), spleen (**new Suppl. Fig. 10B**) and colon (**Figure 6 I**) – are not different in KO and control mice, supporting our hypothesis that the effect we observe in the colon is not the result of a general failure of the KO cells to migrate and homeostatically expand in the lymphopenic host.

We have now also included total numbers of tdTom⁺ CD4⁺ T cells in the colon (LPL), mLnn, spleen, and combined, as well as total numbers of IL-17a^{neg}/IFNγ^{neg} (**new Suppl. Fig. 10C**), in order to make the point very clear that we are not looking at a general maintenance or migration phenotype in the KO mice.

By contrast, we observe a specific lack of Th17 and “exTh17” cells in the colon (**Figure 6 I**). Th17 polarisation and gut homing go hand-in-hand, since CCR6 is a hallmark of Th17 cells and the critical homing chemokine receptor to the gut. We have shown that Hh-inhibited CD4⁺ T cells under Th17-polarising conditions lose CCR6 protein expression (**see Fig. 4C**). We have already shown that *Ihh* KO cells show diminished Th17 polarisation (**see Fig. 5B**) and CCR6 protein expression (**see Fig. 5B**) *in vitro* and hypothesise that the same occurs *in vivo*: meaning less well polarised Th17 cells from the KO mice express lower CCR6 levels and home less well to the gut. We have amended the manuscript text to clarify this point.

We also enumerated the total number of iTregs that developed in the adoptive transfer colitis model, which were only few in number and did not show a significant difference (**new Suppl. Fig. 10D**) again supporting a very specific Th17-polarisation defect in the *Ihh* KO cells.

We did not assess the number of Th2 cells by IL-4 expression in our model since it is well documented in the literature that Th2 cells do not polarise at all in the adoptive transfer colitis model¹⁴(please see Sup. 1c of this paper).

Supplemental Figure 10: Homeostasis and maintenance of CD4⁺ T cells is unaffected by loss of *Ihh* in the adoptive colitis model.

(A-D) *Rag2*^{-/-} mice were injected *i.p.* with 4×10^5 CD45RB^{hi} CD25⁻ tdTom⁺ CD4⁺ T cells isolated from the spleens and peripheral lymph nodes of tamoxifen-treated *CD4ER^{T2}Cre Ihh^{fl/fl}* (HET) and *CD4ER^{T2}Cre Ihh^{fl/fl}* (KO) mice. Numbers of IL-17a⁺, IL-22⁺, IL-17a⁺/IL-22⁺, IFN γ ⁺/IL-17a⁻, IFN γ ⁻/IL-17a⁻, ROR γ t⁺, FoxP3⁺ CD4⁺ T cells isolated from lamina propria (LPL), mesenteric lymph node (mLN) and spleen are shown. (A-C) Gating strategy as shown in Suppl. Fig. 9. All cells are gated on tdTom⁺ CD4⁺ T cells. Data are means \pm SD. p-values were calculated using an unpaired two-tailed Student's t test. n.s. = not significant.

Other concerns:

The authors should include a positive control for Hh components (Suppl Fig2, Fig 1, etc) for the context of expression levels. It is not clear what the qPCR expression data means without context.

We have included qRT-PCR analysis of Mouse Embryonic Fibroblasts (MEFs) as positive controls for expression of Hh components. In addition, we performed qRT-PCR analysis of murine testis (Dhh-positive) and thymus (Gli2-positive) which serve as additional controls for our Dhh and Gli2 probes, respectively (new Fig. 1B and new Suppl. Figure 2A, B).

Figure 1: Key Hedgehog signaling components are induced in Th17 cells. (B) Naïve CD4⁺ T cells were purified from spleen and peripheral lymph nodes of C57BL/6 mice and stimulated with plate-bound anti-CD3 ϵ /CD28 antibodies in the presence of polarizing cytokines to generate Th0, Th1, Th2, Th17 and Treg subsets. Expression of *lhh* was assessed by qRT-PCR in mouse embryonic fibroblasts or naïve CD4⁺ T cells and Th17 cells at the indicated timepoints after TCR stimulation. Data is normalized to *Tbp* as a reference gene. n = 3 independent experiments

Supplemental Figure 2: Expression of Hh signaling components during CD4⁺T helper cell polarization. (A) Naïve CD4⁺T cells were purified from spleen and peripheral lymph nodes of C57BL/6 mice and stimulated with plate-bound anti-CD3 ϵ /CD28 antibodies in the presence of polarizing cytokines to generate Th0, Th1, Th2, Th17 and Treg subsets. Expression of *Ptch1*, *Ptch2*, *Smo*, *Shh* and *Dhh* were assessed by qRT-PCR in Th subsets at the indicated timepoints after TCR stimulation in the presence of polarizing cytokines. Data is normalized to *Tbp* as a reference gene. Similar results were obtained when *CD3 ϵ* was used as a reference gene. n = 3 independent experiments. *Gli2/Dhh* mRNA was undetectable across all conditions tested. Data are means +/- SD. nd = not detected. As a positive control, expression levels of Hh components in Mouse Embryonic Fibroblasts (MEFs) were assessed (right column). n = 3. Data is normalized to *Tbp* as a reference gene. n = 3 independent experiments. (B) Left: Mouse Embryonic Fibroblasts (MEFs) were assessed for *Gli1* and *Gli3* expression by RT-qPCR. n = 3. Right: Testes and thymi were dissected from C57BL/6 mice (n=4), homogenized and lysed for qRT-PCR analysis of *Dhh* and *Gli2* mRNA expression, respectively.

This reviewer is concerned that the authors claim Th17 cells can be generated with IL-6 alone. The authors have (Suppl Fig4) a substantial background of IL-17 staining (~10%). Possibly due to use of MACS separated cells only.

Using negative selection by MACS we yield unlabelled, naïve CD4⁺ T cells that were always >95% but most times >98% pure. These naïve CD4⁺ T cells do not produce any IL-17a when polarised into Th0, Th1, Th2, or Treg cells (Suppl. Fig. 1). The low level of IL-17a+ staining observed in Suppl. Fig. 4 (<10%) is likely due to a slight polarisation towards the Th17 lineage in the presence of IL-6 and anti-IFN γ /IL-4 antibodies.

We do not claim that we can generate bona-fide Th17 cells with IL-6 alone, but rather aim to titrate down number/doses of Th17-polarising cytokines, in order to interrogate (I) the role of Th17-polarising cytokines in inducing expression of Hh components (Fig. 2), (II) whether exogenous active Ihh ligand can enhance polarisation towards the Th17 lineage in suboptimal Th17-polarisation conditions (Fig.3A), and (III) whether blocking all Hh ligands in the culture medium could affect polarisation towards the Th17 lineage at different polarisation strength (Fig. 3B). We have reworded the relevant section of the text to clarify this.

Line 67: Helminth infections, not parasites (i.e. Plasmodium or Toxo are parasites inducing a Th1 response)

We have revised the manuscript accordingly.

Line 183: 1 μ M = 10 nM?

It is 1ng/ml. We have corrected the error.

Figure 2: IL1b symbols should be “+”?

We have amended the figure so that the labelling of the cytokines added to the polarisation cocktail is more consistent.

The authors mention several time they “prove” something. I like to see that changed to data supporting or substantiating a hypothesis.

We have revised the manuscript accordingly.

We have spotted a mistake in the original **Fig. 1E**. Here, we have used the incorrect y-axis for the mRNA expression of *Gli1* in *Th0*, *Th1* and *Th2* conditions. We have corrected our mistake in the **new Fig.1E**.

1. Tokhunts, R. *et al.* The full-length unprocessed hedgehog protein is an active signaling molecule. *J Biol Chem* **285**, 2562-2568 (2010).
2. de la Roche, M. *et al.* Hedgehog signaling controls T cell killing at the immunological synapse. *Science* **342**, 1247-1250 (2013).
3. Geyer, N. *et al.* Different Response of Ptch Mutant and Ptch Wildtype Rhabdomyosarcoma Toward SMO and PI3K Inhibitors. *Front Oncol* **8**, 396 (2018).
4. Yao, C.D. *et al.* AP-1 and TGF β s cooperativity drives non-canonical Hedgehog signaling in resistant basal cell carcinoma. *Nat Commun* **11**, 5079 (2020).
5. Blagih, J. *et al.* The energy sensor AMPK regulates T cell metabolic adaptation and effector responses in vivo. *Immunity* **42**, 41-54 (2015).
6. Nicholas, D.A. *et al.* Fatty Acid Metabolites Combine with Reduced beta Oxidation to Activate Th17 Inflammation in Human Type 2 Diabetes. *Cell Metab* **30**, 447-461 e445 (2019).
7. Teperino, R. *et al.* Hedgehog partial agonism drives Warburg-like metabolism in muscle and brown fat. *Cell* **151**, 414-426 (2012).
8. D'Amico, D. *et al.* Non-canonical Hedgehog/AMPK-Mediated Control of Polyamine Metabolism Supports Neuronal and Medulloblastoma Cell Growth. *Dev Cell* **35**, 21-35 (2015).
9. Wang, C. *et al.* CD5L/AIM Regulates Lipid Biosynthesis and Restrains Th17 Cell Pathogenicity. *Cell* **163**, 1413-1427 (2015).
10. Yuan, X. *et al.* Ciliary IFT80 balances canonical versus non-canonical hedgehog signalling for osteoblast differentiation. *Nat Commun* **7**, 11024 (2016).
11. Crompton, T., Outram, S.V. & Hager-Theodorides, A.L. Sonic hedgehog signalling in T-cell development and activation. *Nat Rev Immunol* **7**, 726-735 (2007).
12. Zhang, D.J. *et al.* Selective expression of the Cre recombinase in late-stage thymocytes using the distal promoter of the Lck gene. *J Immunol* **174**, 6725-6731 (2005).

13. Ostanin, D.V. et al. T cell transfer model of chronic colitis: concepts, considerations, and tricks of the trade. *Am J Physiol Gastrointest Liver Physiol* **296**, G135-146 (2009).
14. Chen, W. et al. A protocol to develop T helper and Treg cells in vivo. *Cell Mol Immunol* **14**, 1013-1016 (2017).
15. Muranski, P. & Restifo, N.P. Essentials of Th17 cell commitment and plasticity. *Blood* **121**, 2402-2414 (2013).
16. Harbour, S.N., Maynard, C.L., Zindl, C.L., Schoeb, T.R. & Weaver, C.T. Th17 cells give rise to Th1 cells that are required for the pathogenesis of colitis. *Proc Natl Acad Sci U S A* **112**, 7061-7066 (2015).
17. Esplugues, E. et al. Control of TH17 cells occurs in the small intestine. *Nature* **475**, 514-518 (2011).
18. Marada, S. et al. Functional Divergence in the Role of N-Linked Glycosylation in Smoothed Signaling. *PLoS Genet* **11**, e1005473 (2015).
19. Raine, T., Liu, J.Z., Anderson, C.A., Parkes, M. & Kaser, A. Generation of primary human intestinal T cell transcriptomes reveals differential expression at genetic risk loci for immune-mediated disease. *Gut* **64**, 250-259 (2015).
20. Haberman, Y. et al. Ulcerative colitis mucosal transcriptomes reveal mitochondriopathy and personalized mechanisms underlying disease severity and treatment response. *Nat Commun* **10**, 38 (2019).
21. Hyams, J.S. et al. Factors associated with early outcomes following standardised therapy in children with ulcerative colitis (PROTECT): a multicentre inception cohort study. *Lancet Gastroenterol Hepatol* **2**, 855-868 (2017).
22. Vanhove, W. et al. Strong Upregulation of AIM2 and IFI16 Inflammasomes in the Mucosa of Patients with Active Inflammatory Bowel Disease. *Inflamm Bowel Dis* **21**, 2673-2682 (2015).
23. Li, M.O. & Flavell, R.A. TGF-beta: a master of all T cell trades. *Cell* **134**, 392-404 (2008).
24. Veldhoen, M. et al. The aryl hydrocarbon receptor links TH17-cell-mediated autoimmunity to environmental toxins. *Nature* **453**, 106-109 (2008).
25. Yang, X.O. et al. T helper 17 lineage differentiation is programmed by orphan nuclear receptors ROR alpha and ROR gamma. *Immunity* **28**, 29-39 (2008).

26. Aghajani, K., Keerthivasan, S., Yu, Y. & Gounari, F. Generation of CD4CreER(T2) transgenic mice to study development of peripheral CD4-T-cells. *Genesis* **50**, 908-913 (2012).
27. Kiesler, P., Fuss, I.J. & Strober, W. Experimental Models of Inflammatory Bowel Diseases. *Cell Mol Gastroenterol Hepatol* **1**, 154-170 (2015).
28. Durant, L. et al. Diverse targets of the transcription factor STAT3 contribute to T cell pathogenicity and homeostasis. *Immunity* **32**, 605-615 (2010).
29. Kaufmann, U. et al. Calcium Signaling Controls Pathogenic Th17 Cell-Mediated Inflammation by Regulating Mitochondrial Function. *Cell Metab* **29**, 1104-1118 e1106 (2019).
30. Yan, J., Pandey, S.P., Barnes, B.J., Turner, J.R. & Abraham, C. T Cell-Intrinsic IRF5 Regulates T Cell Signaling, Migration, and Differentiation and Promotes Intestinal Inflammation. *Cell Rep* **31**, 107820 (2020).

REVIEWER COMMENTS

Reviewer #1 (Remarks to the Author):

The study of Hanna et al. is now greatly improved following all 3 reviewers' suggestions. The new experimental data and the re-wording of some statements, such as the title now fully supports the authors claim. This is very valuable contribution to both immunology and Hh signalling and I support its publication in this journal.

Reviewer #2 (Remarks to the Author):

The reviewer thanks for the detailed revision and concise answers to the open points of the original version of the manuscript.

The authors were able to support the role of IHH signaling in the polarization of Th17 cells by important additional experiments. Equally welcome is the change in title from intracellular to autocrine HH signaling. HH proteins are known to be lipid-modified and can be anchored to the cell membrane. Such a membrane-bound HH signaling mechanism could also be responsible for pathway activation. These alternative options should be consistently implemented in the text (i.e. line 121, line 320 needs to be toned down to: ...model suggests that Hh signaling may be intracellular in CD4+ T cells; lines 627, 631, 655; However, for the manuscript to be accepted, the following points still need to be addressed and added:

1) Figure 3B: 5E1 treatment clearly does not influence Th17 polarization. However, a positive control is missing to show that secreted IHH is blocked by 5E1. Without this control, it cannot be excluded that the antibody is not functional.

2) Figure 5C: the addition of extracellular rIHH cannot rescue the genetic loss of IHH function. However, the positive control showing the biological activity of the recombinant protein is missing here. The authors could treat MEFs with rIHH and measure the activation of HH target genes. Furthermore, WT data should also be included in the right graph in the new Figure 5C.

Minor points:

1) Figure legend 5C (line 344): please add the missing word(s): Representative flow cytometry plots of ... are shown on the left with cells treated with 200ng/ml rlhh....

Reviewer #3 (Remarks to the Author):

The authors have made substantial clarifications and strengthened their data. This reviewer much appreciated the carefully constructed rebuttal page including the new data.

The authors have substantially improved their data and have added data that is in support of the hypothesis and that strengthens the overall conclusions.

1) The inclusion of the other subsets was critical to ensure the authors could rightfully make statements regarding Th17 and not more general T cell activation. The added data and clarifications suffice.

2) The authors make claims regarding Th17 cells, but primarily their analysis concerns the function or activity of Th17 cells; as the authors reiterate including with gene expression data, the Th17 subset itself is actually not affected, but its activity or activation; the production of IL17A, F and IL-22 and expression of CCR6 are. This remains an important

point. The authors show that *lhh* regulates Th17 cells, but in this reviewer's opinion, not the differentiation of Th17 cells. This should be made clear in the title, abstract and conclusions/discussion.

The arguments regarding ROR α and AhR also do not hold. These are not master regulator genes. ROR γ is. The authors do refer to the literature that indeed shows that for example, the absence of AhR reduces IL17 production, but mainly affects a transcriptional programme resulting in IL-22 production. Indeed, the absence of this AhR factor does result on higher Th17 levels *in vivo*.

3) the expression issue is addressed, mRNA level and differences remain low. Figure 2A shows a titration of TGF β and *Gli3* levels. The differences are appreciated at mRNA level. Protein data shown suggests differences but these are not quantified. This quantification would help the authors but are not critical. The role of IL-6 has been made clear.

The crossreactivity is addressed

4) Here the authors use the same arguments to make the opposite point (Fig9). In this short model, it is surprising that both Th17 and Th1 cells are found diminished. This, as the authors argue, can not be due to Th17 to-Th1 conversion. Hence, this appears to be a broader effect that may not be Th17 cell-related? The authors overestimate the importance of Th17 cells in the transfer model, T cells, in general, are pathogenic and Th1 cells play a very important role. The authors have extended their data (Fig.10) sets and show that the numbers of T cells in the colitis models are similar in LN, and Spleen. This is important but passes the important critique that effector T cells in the target tissue are lower, in both Th1 and Th17 cells. This is not explained by Th17-to-Th1 conversion as the authors themselves also argue. This remains a weak point and suggests that more general recruitment (less CCR6) or T cell activation is at play.

The repertoire is at least similar, I agree with the authors on that.

All other concerns have been addressed appropriately.

We thank the reviewers for their re-review of our manuscript NCOMMS-21-04046A-Z (“Cell-autonomous Hedgehog signaling controls Th17 polarization and pathogenicity”).

A detailed point-by-point response can be found below.

Reviewer #1 (Remarks to the Author):

The study of Hanna et al. is now greatly improved following all 3 reviewers' suggestions. The new experimental data and the re-wording of some statements, such as the title now fully supports the authors claim. This is very valuable contribution to both immunology and Hh signalling and I support its publication in this journal.

We thank the reviewer for the constructive feedback, which has substantially improved our manuscript.

Reviewer #2 (Remarks to the Author):

The reviewer thanks for the detailed revision and concise answers to the open points of the original version of the manuscript.

We thank the reviewer for the constructive feedback, which has substantially improved our manuscript.

The authors were able to support the role of IHH signaling in the polarization of Th17 cells by important additional experiments. Equally welcome is the change in title from intracellular to autocrine HH signaling. HH proteins are known to be lipid-modified and can be anchored to the cell membrane. Such a membrane-bound HH signaling mechanism could also be responsible for pathway activation. These alternative options should be consistently implemented in the text (i.e. line 121, line 320 needs to be toned down to: ...model suggests that Hh signaling may be intracellular in CD4+ T cells; lines 627, 631, 655;

We have amended the manuscript text to reflect the alternative option of plasma membrane-bound Ihh not completely blocked by 5E1:

Line 121 (new 114): we have replaced “intracellular” with “cell-autonomous/autocrine”.

Line 320 (new 315): we have replaced “intracellular” with “cell-autonomous”.

Line 655 (new 647): we have replaced “intracellular” with “cell-autonomous”.

Line 627 (new 619): we have rephrased to “...we show for the first time that Hh signalling in CD4⁺ T cells is solely driven by endogenous Ihh - likely acting intracellularly – independent of exogenous Hh ligands”.

Line 631(new 624): remains unchanged, since it is referring to (*de la Roche et al, Science 2013*).

However, for the manuscript to be accepted, the following points still need to be addressed and added:
1) Figure 3B: 5E1 treatment clearly does not influence Th17 polarization. However, a positive control is missing to show that secreted IHH is blocked by 5E1. Without this control, it cannot be excluded that the antibody is not functional.

We have now included the validation of the 5E1 blocking antibody in the **new Supplemental Figure 4D**:

Supplemental Figure 4D: Validation of the 5E1 blocking antibody.

(C,D) MEFs were cultured to confluency. When confluent, cells were treated with the indicated concentrations of active recombinant Indian Hedgehog N-terminal peptide. 24h and 48 hours after stimulation cells were lysed and RNA was extracted for mRNA levels of *Gli1* and *Tbp*. *Gli1* mRNA is used as a reporter of active Hh signalling. (D) Wildtype MEFs were treated as described above for 24h either in the presence of 10µg/ml Hh ligand blocking antibody 5E1 or isotype control. n = 2 independent experiments. Data are means +/- SD. p-values were calculated using an unpaired two-tailed Student's t test. * p<0.05.

2) Figure 5C: the addition of extracellular rIHH cannot rescue the genetic loss of IHH function. However, the positive control showing the biological activity of the recombinant protein is missing here. The authors could treat MEFs with rIHH and measure the activation of HH target genes.

This data has been provided in the original manuscript in **Suppl. Fig. 4C** and is now also shown in **Suppl. Fig.4D**.

Supplemental Figure 4C: Validation of recombinant Ihh

(C,D) MEFs were cultured to confluency. When confluent, cells were treated with the indicated concentrations of active recombinant Indian Hedgehog N-terminal peptide. 24h and 48 hours after stimulation cells were lysed and RNA was extracted for mRNA levels of *Gli1* and *Tbp*. *Gli1* mRNA is used as a reporter of active Hh signalling. (C) *Smo* wildtype (WT) and *Smo*^{-/-}(KO) MEFs were treated as described above. *Smo* KO MEFs, that are unable to induce canonical Hh signalling, serve as a specificity control. n = 3 independent experiments

Furthermore, WT data should also be included in the right graph in the new Figure 5C.

Due to breeding issues, we did not have enough WT mice at the time of revisions that were littermate controls to draw statistically meaningful conclusions (the FACS plots shown in Fig. 5C are however from littermate WT /HET controls). For the statistical analysis of the experiment in Figure 5C, we have therefore focussed on HET where we had n=4 littermate controls to our n=4 KO mice.

Minor points:

1) **Figure legend 5C (line 344): please add the missing word(s): Representative flow cytometry plots of ... are shown on the left with cells treated with 200ng/ml rhhh....**

We have corrected the figure legend.

Reviewer #3 (Remarks to the Author):

The authors have made substantial clarifications and strengthened their data. This reviewer much appreciated the carefully constructed rebuttal page including the new data.

The authors have substantially improved their data and have added data that is in support of the hypothesis and that strengthens the overall conclusions.

1) **The inclusion of the other subsets was critical to ensure the authors could rightfully make statements regarding Th17 and not more general T cell activation. The added data and clarifications suffice.**

2) **The authors make claims regarding Th17 cells, but primarily their analysis concerns the function or activity of Th17 cells; as the authors reiterate including with gene expression data, the Th17 subset itself is actually not affected, but its activity or activation; the production of IL17A, F and IL-22 and expression of CCR6 are. This remains an important point. The authors show that *Ihh* regulates Th17 cells, but in this reviewer's opinion, not the differentiation of Th17 cells. This should be made clear in the title, abstract and conclusions/discussion.**

We appreciate the ambiguity around our use of the term "differentiation" and have adapted our manuscript throughout to replace "differentiation" with "polarization". In the introduction we have now clarified our use of the term "polarization" to match the way the term is widely used in the CD4⁺ T cell field.

We refer to "polarization" as "***the process by which a naïve non-secretory CD4⁺ T cell acquires the functional ability to secrete Th17 cell effector cytokines and by corollary to exert Th17 function [1]***". We believe this helps resolve any ambiguities around the term "differentiation" while maintaining consistency with the way the term "polarization" is used in the CD4⁺ T cell literature.

Below we summarise our findings showing that the Hedgehog pathway in CD4⁺ T cells controls Th17 polarization (rather than just effector function):

1) **Fig. 4B (polarization) versus Fig. 4E (effector function):**

Here we directly address the question whether the Hedgehog pathway affects Th17 polarization or effector function. By treating Th17 cultures either for the first three days only (Fig. 4B, when Th17 polarization occurs) or the last 24 hours of the 5-day-culture (Fig. 4E, when effector function is established), we find that Hedgehog inhibition affects Th17 polarisation and has no effect on effector function once lineage polarisation has occurred.

Of note: this is very different from e.g. CD8⁺ T cells where the Hh pathway affects T cell effector function rather than polarization itself. In CD8⁺ T cells, Hedgehog pathway inhibition 24h before a functional readout (killing) impairs CD8⁺ T cell function (*de la Roche et al., Science 2013*). As outlined above, this is **not** the case in Th17 cells (Fig. 4E).

Figure 4: Small molecule Hh inhibitors selectively block Th17 polarization *in vitro*. (A) Schematic overview of Hh inhibitor cyclopamine administration schedule. (B) Naïve CD4⁺ T cells were stimulated under Th17 or Treg polarizing conditions in the presence of the indicated doses of cyclopamine or carrier control for three days. Cells were harvested for analysis by flow cytometry on day 5. Quantitation of IL-17a/FoxP3 expression, viability measured by absence of live/dead staining, and cell numbers are shown on the right. n = 4 independent experiments. (E) Naïve CD4⁺ T cells were stimulated under Th17 or Treg polarizing conditions in the presence of the indicated dose of cyclopamine or carrier control for the final 24h of polarization. Cells were harvested for analysis by flow cytometry on day 5. n = 3 independent experiments. Data are means \pm SD. p-values were calculated using an unpaired two-tailed Student's t test. * p<0.05, ** p<0.01, *** p<0.001.

2) Fig. 7D and Suppl. Fig. 11 (global changes to the Th17 transcriptional profile):

We find that the **global** Th17 transcriptional profile, identified by John O'Shea and Keiji Zhao [2], is lost upon inhibition of the Hedgehog pathway in addition to loss/diminishment of every Th17 effector function (IL-17a, IL17f, IL-22, CCR6). It is important to point out that the Th17 transcriptional profile is not defined by the level of ROR γ t (or any other known lineage-promoting TF), but by the expression of Th17 downstream transcripts. By assessing these downstream transcripts, we found that inhibition of Hedgehog signalling not only leads to a profound loss of the Th17 transcriptional profile but also to a dramatic shift towards a Th1/2/Treg transcriptional profile, indicating a global loss of Th17 polarization in Hh-inhibited CD4⁺ T cells.

We have clarified these points with text changes.

Figure 7: Mechanistic analysis of Th17-related signaling nodes. (D) Normalized Enrichment Scores (NES) from GSEA of (C) demonstrating a loss of Th17 identity upon Hh inhibitor treatment. Data are means +/- SD. p-values were calculated using an unpaired two-tailed Student's t test. ** indicates p<0.01.

Supplemental Figure 11: Gene Set Enrichment Analysis of RNASeq data demonstrates loss of Th17 identity upon Hh inhibitor treatment.

(A) Gene set enrichment plots (GSE11924 and GSE14308) using RNASeq data shown in Fig. 6C. Th17 cells were polarized as described, stimulated in the presence of the indicated dose of cyclopamine or carrier control for three days, and harvested on day 3. Six samples/group.

The arguments regarding RORα and AhR also do not hold. These are not master regulator genes. RORγt is. The authors do refer to the literature that indeed shows that for example, the absence of AhR reduces IL17 production, but mainly affects a transcriptional programme resulting in IL-22 production. Indeed, the absence of this AhR factor does result on higher Th17 levels in vivo.

We have refrained from using the term “master regulator” when referring to these other Th17 regulators such as Rora, AHR, and Gli3 in our manuscript.

3) the expression issue is addressed, mRNA level and differences remain low. Figure 2A shows a titration of TGF β and Gli3 levels. The differences are appreciated at mRNA level. Protein data shown suggests differences but these are not quantified. This quantification would help the authors but are not critical.

We have shown the quantitation in Suppl. Fig. 4B.

The role of IL-6 has been made clear.

The crossreactivity is addressed.

4) Here the authors use the same arguments to make the opposite point (Fig9). In this short model, it is surprising that both Th17 and Th1 cells are found diminished. This, as the authors argue, can not be due to Th17 –to-Th1 conversion. Hence, this appears to be a broader effect that may not be Th17 cell-related?

The short-term colitis model (Fig. 6A-E) established by the Flavell laboratory [3] lasts only two days and as a result does not give enough time to induce transdifferentiation/sufficient numbers of IL17a/IFN γ ⁺ (“exTh17”) cells for Vismodegib to have a statistically significant effect (see Fig. 6D).

Although not statistically significant, a low level of transdifferentiation is likely to occur. We have altered the text to explicitly acknowledge the apparent decrease in Th1 cells that we believe the reviewer is referring to in Figure 5D (reworded to “while there is non-significant decrease in IL-17a⁺ IFN γ ⁺ IELs” in Fig. 6D).

The authors overestimate the importance of Th17 cells in the transfer model, T cells, in general, are pathogenic and Th1 cells play a very important role.

We have provided evidence from the literature in our initial rebuttal clarifying that not all Th subsets in the intestine are pathological, and that Th17 cells are the critical pathology-inducing Th subset in mouse models of IBD [4]. IFN γ ⁺ CD4⁺ T cells are found in murine adoptive transfer colitis. Experiments using the IL-17 fate mapping reporter mouse [5] and other tools have shown that IFN γ ⁺ CD4⁺ T cells are not *de novo*-polarised Th1 cells but transdifferentiated Th1-like IFN γ ⁺ ex-Th17 lymphocytes [6, 7]. This underlies why Th17 cells are the critical disease-driving Th lineage in adoptive transfer colitis.

The authors have extended their data (Fig.10) sets and show that the numbers of T cells in the colitis models are similar in LN, and Spleen. This is important but passes the important critique that effector T cells in the target tissue are lower, in both Th1 and Th17 cells. This is not explained by Th17-to-Th1 conversion as the authors themselves also argue. This remains a weak point and suggests that more general recruitment (less CCR6) or T cell activation is at play.

In the 6 week long adoptive colitis model, in which Th17-to-Th1 conversion is well-reported, we speculate that the reduction in Th1 cells is due to a reduced Th17-to-Th1 conversion (if there are fewer Th17 cells in the colon, there will be fewer Th17 cells converting to Th1 cells and hence fewer Th1 cells). We appreciate the reviewer’s point about CCR6. Diminished Th17 polarization of naïve *Ihh* KO CD4⁺ T cells in the mesenteric lymph nodes leads to less CCR6 expression and thus less homing to the colon. We have rephrased the corresponding section of the discussion to include this aspect.

The repertoire is at least similar, I agree with the authors on that.

All other concerns have been addressed appropriately.

We thank the reviewer for the constructive feedback, which has substantially improved our manuscript.

1. Zhu J, Yamane H, Paul WE: **Differentiation of effector CD4 T cell populations (*)**. *Annu Rev Immunol* 2010, **28**:445-489.
2. Wei G, Wei L, Zhu J, Zang C, Hu-Li J, Yao Z, Cui K, Kanno Y, Roh TY, Watford WT *et al*: **Global mapping of H3K4me3 and H3K27me3 reveals specificity and plasticity in lineage fate determination of differentiating CD4⁺ T cells**. *Immunity* 2009, **30**(1):155-167.
3. Esplugues E, Huber S, Gagliani N, Hauser AE, Town T, Wan YY, O'Connor W, Jr., Rongvaux A, Van Rooijen N, Haberman AM *et al*: **Control of TH17 cells occurs in the small intestine**. *Nature* 2011, **475**(7357):514-518.
4. Durant L, Watford WT, Ramos HL, Laurence A, Vahedi G, Wei L, Takahashi H, Sun HW, Kanno Y, Powrie F *et al*: **Diverse targets of the transcription factor STAT3 contribute to T cell pathogenicity and homeostasis**. *Immunity* 2010, **32**(5):605-615.
5. Hirota K, Duarte JH, Veldhoen M, Hornsby E, Li Y, Cua DJ, Ahlfors H, Wilhelm C, Tolaini M, Menzel U *et al*: **Fate mapping of IL-17-producing T cells in inflammatory responses**. *Nat Immunol* 2011, **12**(3):255-263.

6. Harbour SN, Maynard CL, Zindl CL, Schoeb TR, Weaver CT: **Th17 cells give rise to Th1 cells that are required for the pathogenesis of colitis.** *Proc Natl Acad Sci U S A* 2015, **112**(22):7061-7066.
7. Kiesler P, Fuss IJ, Strober W: **Experimental Models of Inflammatory Bowel Diseases.** *Cell Mol Gastroenterol Hepatol* 2015, **1**(2):154-170.